# Adaptively Grouped Contextual Bandits for Heterogeneous Human-AI Decision Making with Conformal Prediction Sets

**Yanchen Wu** [1]  **Bo Li** [1]

## Abstract

Personalizing AI decision support for heterogeneous human decision-makers remains a key challenge. We study a collaboration workflow where AI provides a reduced prediction set via conformal prediction and the human makes the final decision based on the set. We formulate this personalization problem as a contextual bandit, where individual and task features form the context, candidate significance levels $\alpha$ serve as arms, and the optimal prediction-set size varies across contexts. To address large arm spaces and high-dimensional contexts, we introduce the Adaptively Grouped Contextual Bandit (AGCB) framework, which avoids global function approximation by exploiting two Human-AI structural assumptions: continuity and monotonicity. Continuity enables information sharing across nearby contexts and decisions, and drives a data-driven Zooming Mechanism that balances intra-group estimation error against inter-group approximation bias. Monotonicity converts each observation into directional counterfactual information over the $K$ candidate $\alpha$ values, reducing the arm-dependence factor from polynomial to logarithmic in $K$. Together, these mechanisms yield minimax-optimal dependence on the learning horizon $T$ for both cumulative and simple regret objectives. Empirical results confirm that AGCB achieves the strongest overall performance across most heterogeneous, data-scarce settings.

## 1. Introduction

A prominent paradigm for human–AI collaboration involves an AI decision support system presenting a prediction set of plausible options, from which a human selects the final decision. For instance, a set of possible disease categories in a medical diagnosis task, or a range of plausible asset prices in a portfolio optimization problem. Critically, each element in such a set can correspond to a potential optimal decision; thus, we may view the prediction set equivalently as a recommended decision set. This paradigm is increasingly adopted in high-stakes domains such as medicine, law, and finance (Straitouri et al., 2023; 2024; 2026; Straitouri & Gomez Rodriguez, 2024). Conformal prediction (CP) provides a model-free method for generating prediction sets with guaranteed marginal coverage (Shafer & Vovk, 2008; Angelopoulos et al., 2024a), offering robust uncertainty quantification. Empirical results confirm that providing precise, well-calibrated sets improves decision outcomes (Vishwakarma et al., 2025). The AI-human system achieves a trade-off: smaller prediction sets increase clarity but raise exclusion risks, whereas larger sets enhance safety at the cost of decisiveness. This balance is regulated by the significance level $\alpha$, which directly determines the joint performance of the human–AI team.

However, human decision–makers are inherently *heterogeneous*. Factors such as expertise, cognitive load, and risk tolerance vary significantly across individuals. Tasks are also heterogeneous, with characteristics like type, difficulty, and human–AI compatibility influencing the optimal support level (Steyvers et al., 2022; Bansal et al., 2021; Ma et al., 2023; Aggarwal et al., 2019; Wang et al., 2020; Zahedi & Kambhampati, 2021; Bansal et al., 2019). Consequently, the accuracy-maximizing $\alpha$ may differ across both individuals and task contexts. This raises a critical design question:

*How should the significance parameter $\alpha$ be set per human–AI interaction to optimally facilitate the human's decision, given heterogeneity across both decision–makers and tasks?*

We aim to model this heterogeneous human–AI collaboration as a contextual bandit problem. This formulation offers a general approach, as it does not require modeling the specifics of the downstream decision task (e.g., multi-label classification (Angelopoulos et al., 2024b; Yeh et al., 2025) or robust decision-making (Kiyani et al., 2025; Bao et al., 2025; Zhou & Zhu, 2026)) or the human's internal cognitive process (Straitouri et al., 2023). Instead, it treats

[1]School of Economics and Management, Tsinghua University, Beijing, China. Correspondence to: Bo Li <libo@sem.tsinghua.edu.cn>.

*Proceedings of the $43^{rd}$ International Conference on Machine Learning*, Seoul, South Korea. PMLR 306, 2026. Copyright 2026 by the author(s).

the human as a black box and learns purely from outcome feedback to select the optimal significance parameter $\alpha$.

We build upon but significantly depart from the bandit-based perspective in Straitouri & Gomez Rodriguez (2024). While that work treats each $\alpha$ as an arm in a stochastic bandit (ignoring contextual heterogeneity), we explicitly model variations among humans and tasks via contextual features. Moreover, tuning $\alpha$ is inherently a parameter selection problem. Thus, beyond the online cumulative regret considered in prior work, we argue that the simple regret, measuring the quality of the final recommended $\alpha$ after a fixed budget, is equally crucial for reliable pre-deployment calibration. This yields a challenging contextual bandit instance with a large arm space (fine-grained calibration for candidate $\alpha$ values) and a non-trivial context dimension. Our goal is thus to achieve minimax-optimal rates while tackling this double curse of dimensionality from both actions and features.

To address this dual challenge, we introduce the Adaptively Grouped Contextual Bandits (AGCB) framework, inspired by data-driven grouping in conformal prediction (Kiyani et al., 2024). Unlike generic bandit algorithms, AGCB is a structure-aware design that leverages two key properties of Human–AI collaboration: **continuity** (Yoo et al., 2021; Wispinski et al., 2020) (similar contexts or $\alpha$ values elicit similar responses) and **monotonicity** (Vishwakarma et al., 2025; Straitouri & Gomez Rodriguez, 2024) (smaller prediction sets improve accuracy by reducing cognitive load).

These properties are fundamental to our algorithmic design. Continuity (formalized as a Hölder condition) guides an adaptive zooming split rule and a cross-group information-sharing scheme to alleviate data sparsity. Monotonicity enables efficient counterfactual updates: a single observation can constrain rewards for other $\alpha$ values, drastically reducing dependence on arm-space complexity. Operationally, AGCB decomposes the problem into lightweight per-group bandit instances, each running a simple base algorithm. Each round follows a concise loop: assign context to a group, select an $\alpha$ via the local bandit with counterfactual updates, share reward feedback across similar groups, and refine the partition with the zooming rule. This structure leverages continuity (for safe sharing and splitting) and monotonicity (for sample-efficient updates) to handle heterogeneity and large action spaces with low computational cost. The final output is an interpretable partition with reliable per-group $\alpha$ estimates, ready for deployment.

Existing contextual bandit approaches face fundamental limitations in this complex setting. For cumulative regret, prevailing methods are ill-suited: parametric models (e.g., LinUCB (Li et al., 2010), linear TS (Agrawal & Goyal, 2013)) assume a simple reward form and are prone to misspecification when modeling human decisions; nonparametric function approximators (e.g., Neural- (Zhou et al., 2020),

Kernel- (Valko et al., 2013), or GP-UCB (Grünewälder et al., 2010)) avoid misspecification but their model-based exploration leads to regret bounds inflated by large complexity constants (e.g., effective dimension, maximum information gain), high computational cost and unstable training especially in online setting; other nonparametric partitioning methods (Akhavan et al., 2024; Perchet & Rigollet, 2013) often rely on rigid splits, facing the curse of dimensionality. For contextual simple regret, the literature is sparse, with methods often being computationally expensive (Jörke et al., 2022), reliant on restrictive structures (Deshmukh et al., 2018), or sensitive to priors (Shi et al., 2023).

Our AGCB framework bypasses these pitfalls by avoiding global function approximation entirely. Its core Zooming Mechanism, derived directly from the regret decomposition, explicitly and optimally balances grouping bias against within-group estimation variance for both regret notions. With data-driven splits, information sharing, and monotonicity-based counterfactual inference, AGCB ensures sample efficiency and yields a regret bound free of extraneous complexity constants, offering a principled and scalable alternative for complex human–AI systems, while also providing interpretable groupings that reveal human decision heterogeneity.

In summary, this work advances the design of reliable Human-AI collaboration by providing a framework for learning personalized decision support. We contribute: (1) a novel and interpretable bandit algorithm (AGCB) that learns data-driven, adaptive user groupings to deliver tailored AI assistance; (2) theoretical guarantees ensuring this personalization is both sample-efficient and robust; and (3) empirical validation showing that our approach consistently improves human decision outcomes over strong and established baselines.

## 2. Related Work

Our work bridges research on human–AI decision making with conformal prediction and contextual bandits for personalization.

### 2.1. Human–AI Collaboration and Conformal Prediction

Conformal prediction provides statistically valid prediction sets for decision support (Shafer & Vovk, 2008; Angelopoulos et al., 2024a). Recent work integrates it into decision-support systems and frames set size selection as a bandit problem to optimize human correctness (Angelopoulos et al., 2024b; Straitouri et al., 2023; 2026; De Toni et al., 2024; Kiyani et al., 2025; Yeh et al., 2025; Hullman et al., 2025; Straitouri & Gomez Rodriguez, 2024), but typically assumes homogeneous users (Steyvers et al., 2022). Another line of

work achieves group-wise conditional coverage via feature space partitioning (Kiyani et al., 2024) but focuses on statistical validity. We adapt adaptive grouping to maximize human decision outcomes under a nonparametric reward structure.

## 2.2. Contextual Bandits for Personalization

Contextual bandits are the standard framework for personalization (Li et al., 2010). In our setting, existing approaches face challenges: parametric models (Li et al., 2010; Abbasi-Yadkori et al., 2011; Filippi et al., 2010) are sample-efficient but prone to misspecification; nonparametric function approximators (Valko et al., 2013; Grünewälder et al., 2010; Zhou et al., 2020) avoid misspecification but suffer from high computational complexity and large complexity constants in their regret bounds; partitioning methods with fixed grids (Perchet & Rigollet, 2013) are non-adaptive and suffer from the curse of dimensionality. The literature on contextual simple regret is limited, often relying on strong structural assumptions (Deshmukh et al., 2018) or being computationally intensive (Jörke et al., 2022; Shi et al., 2023; Russo, 2020).

A key distinction of our work is the Zooming Mechanism, which directly balances regret's bias-variance trade-off, unlike complex function approximators that explore based solely on reward-function uncertainty, a strategy not always optimal for regret minimization. Our AGCB framework thus avoids expensive global approximation, learns an adaptive partition via zooming, and leverages continuity (Yoo et al., 2021) and monotonicity (Straitouri & Gomez Rodriguez, 2024) inherent in human–AI collaboration to provide a unified treatment for both simple and cumulative regret.

## 3. Problem Setting

Let $\mathcal{X} \subseteq \mathbb{R}^d$ be the context space representing human characteristics and task features, and $\mathcal{Y}$ be the outcome space (e.g., the label space in multi-class classification, or the set of possible decisions in general decision problems). Let $y \in \mathcal{Y}$ denote the prediction outcome required for optimal human decision-making. A decision support system provides a prediction set $C_\alpha(x) \subseteq \mathcal{Y}$ parameterized by $\alpha \in \mathcal{A}$, where $\alpha$ controls the informativeness of the set. In conformal prediction, $\alpha$ typically corresponds to the significance level, with $\mathcal{A}$ being a finite set derived from a calibration set (Straitouri et al., 2023).

The human-AI interaction proceeds as follows:

1. Given context $x \in \mathcal{X}$, our decision support system selects a parameter $\alpha \in \mathcal{A}$.

2. Given a calibration set $\mathcal{D}_{\text{cal}}$ and any pre-trained model (e.g., a classifier), we construct conformal prediction

sets as follows. For each candidate outcome $y$, let $s$ denote the nonconformity score of the data point computed using the pre-trained model. Then the prediction set is

$$C_\alpha(x) = \big\{ y \in \mathcal{Y} : s \leq q_{1-\alpha} \big\},$$

where $q_{1-\alpha}$ is the $(1-\alpha)$-quantile of the nonconformity scores evaluated on the calibration set $\mathcal{D}_{\text{cal}}$.

3. The human decision maker observes $C_\alpha(x)$ and selects an action $\hat{y} \in C_\alpha(x)$.

4. The system observes $y$ and receives a stochastic reward $R(\hat{y}, y)$.

We define the realized reward for choosing parameter $\alpha$ under context $x$ and outcome $y$ as

$$r_\alpha(x, y) = R\big(h(x, C_\alpha(x)), y\big) \cdot \mathbf{1}\{y \in C_\alpha(x)\},$$

where $h : \mathcal{X} \times 2^{\mathcal{Y}} \to \mathcal{Y}$ represents the human decision-making process, mapping a context and prediction set to a chosen action. Note that $\hat{y} = h(x, C_\alpha(x))$ represents the human's decision. The indicator $\mathbf{1}\{y \in C_\alpha(x)\}$ captures the severe penalty when the prediction set excludes the true outcome: in that case, the realized reward is zero.

The true expected reward function $\mu : \mathcal{A} \times \mathcal{X} \to \mathbb{R}$ is then defined as

$$\mu(\alpha, x) = \mathbb{E}_{y|x} \left[ r_\alpha(x, y) \right].$$

### 3.1. Learning Objective and Heterogeneity

The goal is to learn a policy $\pi : \mathcal{X} \to \mathcal{A}$ that maps contexts to significance level $\alpha$. For any context $x \in \mathcal{X}$, the policy selects $\alpha = \pi(x)$, which determines the prediction set $C_{\pi(x)}(x)$. Given this set, the human selects an action $\hat{y} = h(x, C_{\pi(x)}(x))$, and the system receives reward $R(\hat{y}, y)$. The expected reward under policy $\pi$ is:

$$\mathbb{E}_{(x,y) \sim \mathcal{D}} \left[ r_{\pi(x)}(x, y) \right] = \mathbb{E}_{x \sim \mathcal{D}_\mathcal{X}} \left[ \mu(\pi(x), x) \right],$$

where $\mathcal{D}$ is the joint distribution over contexts and outcomes, and $\mathcal{D}_\mathcal{X}$ is the marginal distribution over contexts. Thus, the learning objective is to find:

$$\max_{\pi : \mathcal{X} \to \mathcal{A}} \mathbb{E}_{x \sim \mathcal{D}_\mathcal{X}} \left[ \mu(\pi(x), x) \right].$$

Due to heterogeneity in the decision-making environment, the optimal $\alpha$ may vary with $x$. Let $\alpha^*(x) = \arg\max_{\alpha \in \mathcal{A}} \mu(\alpha, x)$ be the globally optimal arm for context $x$. The optimal policy $\pi^*$ selects $\alpha^*(x)$ for each $x$, i.e., $\pi^*(x) = \alpha^*(x)$ for all $x \in \mathcal{X}$.

### 3.2. Bandit Formulation

We model $\alpha$ selection as a contextual bandit (arm $\alpha$, context $x$, reward mean $\mu(\alpha, x)$), enabling online learning without

modeling the human decision function. Prior work (Straitouri & Gomez Rodriguez, 2024) relies on binary feedback (correct/incorrect decision) and monotonicity to update multiple arms per round. Our framework extends this to continuous reward structures. Moreover, while (Straitouri & Gomez Rodriguez, 2024) focuses on cumulative regret, we also consider simple regret to ensure high final performance after fine-tuning $\alpha$. In deployment, a new task is assigned to its group and receives the optimal $\alpha$ learned from that group, enabling immediate personalization.

### 3.2.1. SIMPLE REGRET FOR FINAL POLICY QUALITY

After $T$ rounds, the algorithm outputs a group-wise policy $\pi_T : \mathcal{X} \to \mathcal{A}$ that is constant on each group of the final partition $\mathcal{G}_T$: for all $x \in G$, $\pi_T(x) = \alpha_G$. The simple regret measures the expected suboptimality of this final policy:

$$
\begin{aligned}
S(T) &= \mathbb{E}_{x \sim \mathcal{D}_{\mathcal{X}}} \left[ \mu(\pi^*(x), x) - \mu(\pi_T(x), x) \right] \\
&= \sum_{G \in \mathcal{G}_T} \Pr(x \in G) \, \mathbb{E}_{x \in G} \left[ \mu(\alpha^*(x), x) - \mu(\alpha_G, x) \right],
\end{aligned}
\tag{1}
$$

where $\alpha_G$ is the arm chosen for group $G$ after $T$ rounds. The inner expectation captures both the approximation bias due to using a single arm for the whole group and the estimation error of selecting $\alpha_G$. Minimizing $S(T)$ ensures the final deployed policy performs well on future instances.

### 3.2.2. CUMULATIVE REGRET FOR LEARNING EFFICIENCY

During the learning phase, we also care about cumulative regret, which captures the total performance loss incurred while learning. Let $\mathcal{G}_t$ denote the partition at time $t$ (which may evolve over time). Then:

$$
\begin{aligned}
R(T) &= \sum_{t=1}^{T} \left[ \mu(\alpha^*(x_t), x_t) - \mu(\alpha_t, x_t) \right] \\
&= \sum_{G \in \mathcal{G}_T} \sum_{t : x_t \in G} \left[ \mu(\alpha^*(x_t), x_t) - \mu(\alpha_t, x_t) \right], \quad (2)
\end{aligned}
$$

where $\alpha_t$ is the arm selected at round $t$. The second equality decomposes the regret according to the final partition $\mathcal{G}_T$; each inner sum collects the regrets of rounds whose contexts fall into the same final group. Minimizing $R(T)$ is crucial for maintaining satisfactory performance during the learning phase, as it directly controls the total suboptimality experienced while identifying the best arm for each group.

## 4. Methodology

### 4.1. Human-Aligned Structural Assumptions

Before presenting our algorithm, we first introduce the core assumptions regarding human decision behavior, the Hölder

continuity and monotonicity assumptions. Leveraging these assumptions is crucial for our algorithm's effectiveness.

**Assumption 4.1** (Joint Hölder Continuity of the Reward Function). The expected reward function $\mu : \mathcal{A} \times \mathcal{X} \to \mathbb{R}$ is Hölder continuous on the joint action-context space. Formally, there exist constants $L > 0$, $\beta \in (0, 1]$, and a metric $d$ combining action and context distances, such that for any $(\alpha, x), (\alpha', x') \in \mathcal{A} \times \mathcal{X}$,

$$
|\mu(\alpha, x) - \mu(\alpha', x')| \leq L \cdot d\left((\alpha, x), (\alpha', x')\right)^{\beta}.
$$

A canonical choice for the metric is $d\left((\alpha, x), (\alpha', x')\right) = \|x - x'\|_2 + \lambda |\alpha - \alpha'|$, where $\lambda > 0$ balances the scales of context and action distances.

This assumption states that similar contexts and decisions lead to similar outcomes. It enables information sharing across groups and meaningful counterfactual updates for continuous decisions. Assumption 4.1 is not arbitrary but follows from the continuity of the upstream learning system and human feedback; a formal derivation is given in Appendix A.

**Assumption 4.2** (Conditional Monotonicity under Coverage). For any context $x$ and parameters $\alpha' < \alpha$, if the more precise prediction set still covers the true outcome, i.e., $y \in C_\alpha(x)$, then

$$
R\big(h(x, C_\alpha(x)), y\big) \geq R\big(h(x, C_{\alpha'}(x)), y\big).
$$

Equivalently, conditional on coverage being preserved, increasing $\alpha$ shrinks the prediction set and leads to no worse decision quality.

Assumption 4.2 extends the binary feedback monotonicity in Straitouri & Gomez Rodriguez (2024) to continuous rewards: when coverage occurs, a smaller set reduces cognitive load and improves decision quality. Note, however, that the unconditional expected reward $\mu(\alpha, x)$ is not monotonic in $\alpha$ due to the coverage–precision trade-off. This, combined with heterogeneity across $x$, makes the optimal $\alpha^*(x)$ context-dependent and non-trivial to identify.

### 4.2. AGCB Framework

Our Adaptively Grouped Contextual Bandits (AGCB) framework learns personalized policies by grouping similar situations (contexts) and sharing knowledge between groups. It operates in a simple, iterative loop (Algorithm 1).

The AGCB framework operates through a concise four module loop: (1) observe context $x_t$, and determine the group $g_t \in \mathcal{G}_t$ to which $x_t$ belongs, (2) execute a bandit round within that group, selecting $\alpha_t$, obtaining human feedback $r_t$, and counterfactually update the group's statistics, (3) share this observation with other similar groups to boost

**Algorithm 1** Adaptively Grouped Contextual Bandits (AGCB) – Meta-Framework

**Require:** Context space $\mathcal{X}$, parameter set $\mathcal{A}$, horizon $T$. A base stochastic bandit algorithm $Alg$ with counterfactual update.

**Ensure:** Arm $\alpha_t$ to pull in the current round, learned partition $\mathcal{G}_T$ and group-wise policy $\{\hat{\alpha}_g^*\}$.

1: Start with one group covering all contexts: $\mathcal{G}_1 \leftarrow \{g_0\}$.
2: **for** $t = 1$ to $T$ **do**
3:    **Assign to group:** Observe context $x_t$, and determine the group $g_t \in \mathcal{G}_t$ for $x_t$.
4:    **Pull arm and counterfactual update:** Use group $g_t$'s current estimates to select $\alpha_t$ by $Alg$, show $C_{\alpha_t}(x_t)$ to human, and observe reward $r_t$. Counterfactual Update $g_t$'s statistics with $(\alpha_t, r_t)$.
5:    **Share across groups:** Propagate the observation $(\alpha_t, r_t)$ to other groups similar to $g_t$ using kernel $K$.
6:    **Refine grouping:** If group $g_t$'s approximation bias is statistically resolvable relative to its uncertainty, split it (Zooming Mechanism).
7: **end for**
8: **Output:** Final grouping $\mathcal{G}_T$ and the best parameter $\hat{\alpha}_g^*$ for each group $g$.

data efficiency, and (4) adaptively split a group once its local approximation bias becomes statistically resolvable.

Before detailing each component, we highlight the core ideas behind AGCB that exploit the structure of human–AI collaboration by Assumption 4.1 and 4.2. Continuity enables safe information sharing across similar groups (**Share across groups**) and underlies the adaptive Zooming Mechanism (**Refine grouping**) that balances bias and variance. Monotonicity amplifies within-group learning (**Pull arm and counterfactual update**): a single observation for one $\alpha$ gives directional bounds for others, allowing counterfactual updates that reduce the dependence on arm space. This synergy makes AGCB sample-efficient and adaptive.

We now detail Algorithm 1, first outlining its macro-architecture (core modules), then elaborating on the micro-mechanism of counterfactual updates within the base bandit.

**Module 1: Assign to group** Initially the whole context space forms a single group. As the algorithm proceeds, the space is partitioned into a binary tree by recursively splitting along feature dimensions (see Module 4). Each group corresponds to a region described by intervals in each dimension (e.g., all dimensions less than 0.5). For a new context $x_t$, we determine its group by checking which intervals its feature values fall into.

**Module 2: Pull arm and counterfactual update** This

part concerns how to pull arms in the specific base bandit algorithm, how to perform counterfactual inference based on feedback, and how to update other arms accordingly. This will be detailed in Section 4.3.

**Module 3: Share across groups** This module enables cross-group credit assignment. For groups $g$ and $h$ with centers $c_g$ and $c_h$, we define their similarity using a Gaussian kernel $w_{gh} = \exp\left(-\frac{\|c_g - c_h\|^2}{2h_g^2}\right)$, where $h_g \propto d_g$. Reward estimates are pooled across similar groups as

$$\widehat{\mu}_g(\alpha) = \frac{\sum_h w_{gh} n_h \widehat{\mu}_h(\alpha)}{\sum_h w_{gh} n_h},$$

where $n_h$ is the raw number of observations in group $h$. Thus, groups that are both similar to $g$ and supported by more observations contribute more to the estimate of group $g$. For analysis, we write the variance-equivalent effective sample size of this weighted estimator as $n_g^{\text{eff}} = s_g n_g$, where $s_g \geq 1$ is the effective information-sharing gain of group $g$; see Appendix G.1 for the formal definition. This turns Hölder continuity into a concrete mechanism for sample-efficient cross-group sharing.

**Module 4: Refine grouping** The granularity of the partition is controlled by a Zooming Mechanism that formalizes the bias–variance trade-off. For the Sequential Halving and Successive Elimination base learners instantiated below, this balance takes the following form. For a group $g$ with diameter $d_g$ and effective sample size $n_g^{\text{eff}} = s_g \cdot n_g$, splitting occurs once the local statistical uncertainty becomes sufficiently small relative to the inherent approximation bias:

$$C\sigma\sqrt{\frac{\log(KT)}{n_g^{\text{eff}}}} \leq \eta \cdot L \cdot (d_g)^\beta, \tag{3}$$

where $C, \eta > 0$ are constants and $L, \beta$ are the Hölder parameters. The condition refines the partition only when the approximation bias is statistically resolvable. The information-sharing gain $s_g \geq 1$ accelerates uncertainty reduction, allowing AGCB to split heterogeneous regions earlier while keeping coarse groups in regions where refinement is not yet statistically justified. A more general base-learner-dependent zooming analysis is provided in Appendix B, with a comparison to classical zooming bandits (Kleinberg et al., 2008; Slivkins, 2019) in Appendix C.

When a split is triggered for group $g$, we perform a regularized Variance-Midpoint Split (VMS), which restricts splits to non-degenerate dimensions: (1) Select the eligible dimension with the highest empirical reward variance (computed from the group's history), and split the group at the midpoint along that dimension. (2) Inherit weighted sufficient statistics to the child groups, weighted by the estimated fraction of parent data falling into each child region.

Inheritance adds only an absorbable $O(Ld_g^\beta)$ bias and improves post-split sample efficiency (Appendix D). Unlike the static grid splits (BSE) or the predefined $2^d$-ary splits (ABSE) of (Perchet & Rigollet, 2013), our data-driven binary splits refine only where needed, thereby mitigating the practical curse of dimensionality while matching the minimax-optimal $T$-exponents in the regret bounds. A detailed comparison is provided in Appendix E.

### 4.3. Generalizing Counterfactual Learning with Monotonicity and Continuity

Having described the meta-algorithm, we now focus on the within-group counterfactual update used in the **Pull arm and counterfactual update** step of Algorithm 1. We extend the binary-feedback counterfactual idea of Straitouri & Gomez Rodriguez (2024) to continuous rewards by combining conditional monotonicity under coverage (Assumption 4.2) with Hölder continuity (Assumption 4.1).

In the binary-feedback setting of Straitouri & Gomez Rodriguez (2024), monotonicity allows deterministic updates of other arms' estimates (e.g., a success at $\alpha_t$ (reward 1) implies that arms $\alpha' \geq \alpha_t$ would also yield reward 1 when coverage event occurs). For continuous rewards, we adapt this idea using Hölder continuity (Assumption 4.1) to perform soft, local adjustments around the observed $\alpha_t$ without harming the regret bound. Monotonicity still guarantees the adjustment direction is correct: when coverage event occurs, reward at $\alpha_t$ implies larger $\alpha'$ should have no lower estimates, while smaller $\alpha'$ should have no higher estimates:

$$\hat{\mu}_g(\alpha') \leftarrow \frac{\hat{\mu}_g(\alpha') \cdot n_{g,t}^{\text{eff}}(\alpha') + w(\alpha_t, \alpha') \cdot r_t}{n_{g,t}^{\text{eff}}(\alpha') + w(\alpha_t, \alpha')}, \quad (4)$$

$$\hat{\mu}_g(\alpha') \leftarrow \max\left\{\hat{\mu}_g(\alpha'), \hat{\mu}_g(\alpha_t)\right\} \quad \text{for } \alpha^\dagger \geq \alpha' \geq \alpha_t \tag{5}$$

$$\hat{\mu}_g(\alpha') \leftarrow \min\left\{\hat{\mu}_g(\alpha'), \hat{\mu}_g(\alpha_t)\right\} \quad \text{for } 0 < \alpha' \leq \alpha_t \tag{6}$$

where $w(\alpha_t, \alpha) = \exp\left(-\frac{|\alpha_t - \alpha|^2}{2h_g^2}\right)$ and $n_{g,t}^{\text{eff}}(\alpha')$ denotes the current effective sample size of arm $\alpha'$ in group $g$. $\alpha^\dagger = \max\{\alpha \in \mathcal{A}_g : y \in C_\alpha(x)\}$ means the highest significance level ensuring the coverage event occurs. The directional updates (Eqs. (5) and (6)) enforce monotonicity while smoothing via the kernel weights. When $y \in C_{\alpha_t}(x)$, Eq. (5) applies to $\alpha_t \leq \alpha' \leq \alpha^\dagger$, whose prediction sets are smaller and more precise but still cover $y$, so their rewards should be no smaller than that at $\alpha_t$. Eq. (6) applies to $\alpha' \leq \alpha_t$, whose sets are larger and more conservative, so their rewards should be no larger when coverage is already preserved. When $y \notin C_{\alpha_t}(x)$, arms with $\alpha' > \alpha^\dagger$ have even smaller sets that also fail to cover $y$ and can be updated toward zero.

Finally, Algorithms 2 and 3 (Algorithm 3 is shown in Appendix F) leverage sample-level monotonicity and continu-

---

**Algorithm 2** Enhanced Base Learner: COUNTERFACTUAL-AWARE SEQUENTIAL HALVING (Simple Regret)

---
**Require:** Sorted arms $\mathcal{A}_g = \{\alpha_1 < \cdots < \alpha_m\}$, total budget $B$.
**Ensure:** Best arm $\alpha_g^*$.
1: $L \leftarrow 1, U \leftarrow m, B_{\text{phase}} \leftarrow \lfloor B/\lceil \log_2 m \rceil \rfloor$.
2: **while** $U - L > 0$ **do**
3:     $M \leftarrow \lfloor (L+U)/2 \rfloor, \alpha_{\text{mid}} \leftarrow \mathcal{A}_g[M]$.
4:     **for** $b = 1$ to $B_{\text{phase}}$ **do**
5:         Pull $\alpha_{\text{mid}}$; observe $(r, y)$. Update $\hat{\mu}_g(\alpha_{\text{mid}})$.
6:         Compute $\alpha^\dagger \leftarrow \max\{\alpha \in \mathcal{A}_g : y \in C_\alpha(x)\}$.
7:         **if** $y \in C_{\alpha_{\text{mid}}}(x)$ **then**
8:             Update $\hat{\mu}_g(\alpha_j)$ for $\alpha^\dagger \geq \alpha_j > \alpha_{\text{mid}}$ (Eq. (5)).
9:             Update $\hat{\mu}_g(\alpha_j)$ for $0 < \alpha_j \leq \alpha_{\text{mid}}$ (Eq. (6)).
10:        **else**
11:             Update $\hat{\mu}_g(\alpha_j)$ for $\alpha_j > \alpha^\dagger$ with reward 0.
12:        **end if**
13:     **end for**
14:     Compute $\tilde{\mu}_g(\alpha_{\text{mid}}), \tilde{\mu}_g(\mathcal{A}_g[L])$.
15:     **if** $\tilde{\mu}_g(\alpha_{\text{mid}}) \geq \tilde{\mu}_g(\mathcal{A}_g[L])$ **then**
16:         $L \leftarrow M$ {Middle performs better, keep right half}
17:     **else**
18:         $U \leftarrow M - 1$ {Left performs better, keep left half};
19:     **end if**
20: **end while**
21: $\alpha_g^* \leftarrow \mathcal{A}_g[L]$.

---

ity to perform binary-search-style counterfactual updates, thereby accelerating learning by sharing information across a large portion of the action space. Specifically, Algorithm 2 differs from the classic Sequential Halving by exploiting monotonicity: instead of distributing the budget evenly among all arms in each phase, it concentrates the budget on the middle arm of the current interval and propagates the information to the other arms through counterfactual updates. Similarly, Algorithm 3 pulls only the median arm of the active set and updates the remaining arms counterfactually. Although the group-averaged reward need not be exactly monotone or single-peaked, the counterfactual pruning step is only required to be locally safe: any incorrectly pruned arm is within the current local resolution of a retained arm. The Zooming Mechanism makes this local resolution comparable to the bias–variance scale, so the resulting error is absorbed into the regret analysis. Thus, both algorithms integrate binary-search-style updates with monotonicity and continuity, reducing the arm-space dependence from polynomial in $K$ to $\log K$; see Appendix F for details.

*Remark* 4.3. In Algorithms 2 and 3, when only two arms remain ($U = L + 1$), we directly compare their estimates and return the better one.

# 5. Theoretical Guarantees

This section presents regret bounds for our AGCB framework. Let $\mathcal{X} \subset \mathbb{R}^d$ be compact and $\mathcal{A}$ the arm set, $K = |\mathcal{A}|$. The expected reward $\mu(\alpha, x)$ satisfies the $(\beta, L)$-Hölder condition (Assumption 4.1). We treat AGCB as a meta-algorithm. The base bandit algorithm $\mathcal{B}$ has two performance parameters within a homogeneous group: (1) expected simple regret decay $\zeta \in (0, \frac{1}{2}]$ with $\mathbb{E}(r_{\mathcal{B}}(n, K)) \lesssim \Phi(K, \sigma)n^{-\zeta}$, and (2) expected cumulative regret growth $\gamma \in [\frac{1}{2}, 1)$ with $\mathbb{E}(R_{\mathcal{B}}(n, K)) \lesssim \Psi(K, \sigma)n^{\gamma}$. These bounds depend on factors $\Phi(K, \sigma)$ and $\Psi(K, \sigma)$ (e.g., $\sigma\sqrt{K}$, $\sigma\sqrt{\log K}$), and we omit logarithmic factors in $T$.

## 5.1. Regret Bounds for the AGCB Meta-Algorithm

We first present the fundamental performance guarantees of the AGCB meta-algorithm. Throughout this section, $\widetilde{O}$ hides universal constants and logarithmic factors in $T$, while dependence on the number of arms $K$ is displayed.

**Theorem 5.1** (Regret Upper Bounds). *Under Assumption 4.1, suppose the reward is conditionally $\sigma$-sub-Gaussian given the context $x$ and parameter $\alpha$. Let the within-group base learner $\mathcal{B}$ satisfy the local structured-bias conditions in Appendix G.1, with simple-regret exponent $\zeta \in (0, \frac{1}{2}]$, cumulative-regret exponent $\gamma \in [\frac{1}{2}, 1)$, and base-learner factors $\Phi(K, \sigma)$ and $\Psi(K, \sigma)$. Then the AGCB meta-algorithm satisfies*

$$\mathbb{E}[S(T)] \leq \widetilde{O}\left( L^{\frac{d\zeta}{\beta + d\zeta}} \Phi(K, \sigma)^{\frac{\beta}{\beta + d\zeta}} \Gamma_{\min}^{-\frac{\beta\zeta}{\beta + d\zeta}} T^{-\frac{\beta\zeta}{\beta + d\zeta}} \right),$$

$$\mathbb{E}[R(T)] \leq \widetilde{O}\left( L^{\frac{d(1-\gamma)}{\beta + d(1-\gamma)}} \Psi(K, \sigma)^{\frac{\beta}{\beta + d(1-\gamma)}} \Gamma_{\min}^{-\frac{\beta(1-\gamma)}{\beta + d(1-\gamma)}} \right.$$
$$\left. T^{1 - \frac{\beta(1-\gamma)}{\beta + d(1-\gamma)}} \right).$$

$\Gamma_{\min} = \min_{g \in \mathcal{G}_T} s_g$ *is the minimum information-sharing gain of the final partition $\mathcal{G}_T$.*

*Remark* 5.2 (Reward noise). Our $\sigma$-sub-Gaussian condition generalizes the bounded-reward assumption common in practice and in prior work (Straitouri et al., 2026; Straitouri & Gomez Rodriguez, 2024). Since real-world rewards (e.g., accuracy, utility scores) are typically bounded, $\sigma$ can be set directly from known bounds (e.g., $\sigma = (b-a)/2$ for rewards in $[a, b]$), ensuring our theory is both broadly applicable.

We now instantiate Theorem 5.1 with two standard base learners to obtain concrete, interpretable regret bounds. These corollaries highlight how our framework addresses the challenges of many arms and the curse of dimensionality.

**Corollary 5.3** (Simple Regret with Sequential Halving). *When using Sequential Halving as the base learner $\mathcal{B}_{SH}$ within groups, which has $\zeta = 1/2$ and $\Phi(K, \sigma) = \sigma\sqrt{K}$*

*(Zhao et al., 2023), the AGCB framework achieves:*

$$\mathbb{E}(S(T)) \leq \widetilde{O}(L^{\frac{d}{2\beta + d}} \cdot \Gamma_{\min}^{-\frac{\beta}{2\beta + d}} \cdot T^{-\frac{\beta}{2\beta + d}} \cdot \mathcal{H}(K, \sigma)),$$

*where the arm-dependence factor $\mathcal{H}(K, \sigma)$ is (traditional update and counterfactual update):*

$$\mathcal{H}(K, \sigma) = \begin{cases} O\left( \sigma^{\frac{2\beta}{2\beta + d}} K^{\frac{\beta}{2\beta + d}} \right), & \text{Single Point,} \\ O\left( \sigma^{\frac{2\beta}{2\beta + d}} (\log K)^{\frac{\beta}{2\beta + d}} \right), & \text{Counterfactual.} \end{cases}$$

**Corollary 5.4** (Cumulative Regret with Successive Elimination). *When using Successive Elimination as the base learner $\mathcal{B}_{SE}$, which has $\gamma = 1/2$ and $\Psi(K, \sigma) = \sigma\sqrt{K}$ (Slivkins, 2019), the AGCB framework achieves:*

$$\mathbb{E}(R(T)) \leq \widetilde{O}(L^{\frac{d}{2\beta + d}} \cdot \Gamma_{\min}^{-\frac{\beta}{2\beta + d}} \cdot T^{\frac{\beta + d}{2\beta + d}} \cdot \mathcal{J}(K, \sigma)),$$

*where the arm-dependence factor $\mathcal{J}(K, \sigma)$ is (traditional update and counterfactual update):*

$$\mathcal{J}(K, \sigma) = \begin{cases} O\left( \sigma^{\frac{2\beta}{2\beta + d}} K^{\frac{\beta}{2\beta + d}} \right), & \text{Single Point,} \\ O\left( \sigma^{\frac{2\beta}{2\beta + d}} (\log K)^{\frac{\beta}{2\beta + d}} \right), & \text{Counterfactual.} \end{cases}$$

Our results, proved in Appendix G, match the standard minimax-optimal $T$-exponents for Hölder contextual bandits: $T^{-\beta/(2\beta + d)}$ for simple regret and $T^{(\beta+d)/(2\beta+d)}$ for cumulative regret (Takezawa, 2005; Rigollet & Zeevi, 2010). Thus, our optimality claim concerns the horizon dependence, which is the central sample-complexity parameter. For the arm dependence, standard unstructured bounds scale polynomially in $K$, whereas our counterfactual updates exploit monotonicity and reduce the dependence from $K^{\frac{\beta}{2\beta + d}}$ to $(\log K)^{\frac{\beta}{2\beta + d}}$ for both regret notions, consistent with Straitouri & Gomez Rodriguez (2024). The exponent in $d$ reflects the unavoidable curse of dimensionality, but AGCB still offers practical gains over non-adaptive baselines: the Zooming Mechanism avoids exponential fixed-grid pre-partitioning (Perchet & Rigollet, 2013), and the information-sharing factor $\Gamma_{\min}^{-\beta/(2\beta + d)}$ improves the leading constant. Empirically, AGCB consistently outperforms grid-based partitioning. Reducing the dimension dependence would require stronger structures, such as low-rank latent factors, which are often incompatible with the heterogeneity of human–AI decision making.

**Unknown Continuity Parameters and Model Selection**
The Zooming Mechanism uses the Hölder parameters $(\beta, L)$ to balance local approximation bias and statistical uncertainty. If the smoothness exponent $\beta$ is misspecified, the induced partition scale can be suboptimal: overestimating $\beta$ yields overly coarse groups, while underestimating $\beta$ yields overly fine groups. Thus, fixed misspecification of $\beta$ generally affects the regret rate rather than only the leading constant; see Appendix H.

*Table 1.* Simple regret, 30 runs, 100 Arms, 5-dim Context

| Algorithm | Binary Feedback | | Continuous Feedback | |
|---|---|---|---|---|
| | Regret | Time (s) | Regret | Time (s) |
| AGCB-SH (ours) | **0.298** | 4.11 | **0.111** | 4.36 |
| SH (vanilla) | 0.381 | 1.18 | 0.211 | 1.33 |
| Contextual-Gap | 0.352 | 9.63 | 0.124 | 9.32 |
| BOED | 0.349 | 10.86 | 0.200 | 10.56 |
| TTTS-C | 0.343 | 9.89 | 0.158 | 9.91 |
| LinUCB-PE | 0.374 | 5.13 | 0.163 | 5.81 |

*Table 2.* Simple regret, 30 runs, 10 Arms, 5-dim Context

| Algorithm | Binary Feedback | | Continuous Feedback | |
|---|---|---|---|---|
| | Regret | Time (s) | Regret | Time (s) |
| AGCB-SH (ours) | **0.193** | 1.05 | **0.086** | 1.09 |
| SH (vanilla) | 0.250 | 0.98 | 0.141 | 1.01 |
| Contextual-Gap | 0.214 | 4.91 | 0.087 | 6.87 |
| BOED | 0.190 | 3.06 | 0.071 | 3.26 |
| TTTS-C | 0.219 | 3.86 | 0.115 | 4.03 |
| LinUCB-PE | 0.223 | 2.32 | 0.109 | 2.16 |

When $(\beta, L)$ are unknown, one can instead use model-selection techniques. Following Liu et al. (2021), we may discretize the parameter space into $M$ candidate pairs $(\beta_i, L_i)$ and run $M$ AGCB instances in parallel, combined by a master algorithm such as Corral (Pacchiano et al., 2020). For cumulative regret, this gives

$$R(T) \leq \widetilde{O}\left(\min_i R_i(T) + \sqrt{T \log M}\right),$$

where $R_i(T)$ is the regret bound of the AGCB instance using $(\beta_i, L_i)$. This matches the known phenomenon that adaptation to unknown smoothness incurs an additional cost (Locatelli & Carpentier, 2018). Analogous model-selection or aggregation tools can be used for simple regret as well (Grill et al., 2015; Locatelli et al., 2017); details are in Appendix I.

## 6. Experiment

We evaluate the proposed algorithm on both synthetic data and real-human data to demonstrate its effectiveness. The synthetic experiments (Section 6.1) compare AGCB against several baseline methods under controlled settings, while the real-human experiments (Section 6.2) use the ImageNet16H-PS dataset to validate performance in realistic human-AI decision-making scenarios. Our code and results are available in https://github.com/YanchenWu2001/AGCB_CP.

### 6.1. Synthetic Data Experiment

We evaluate AGCB under two feedback settings: binary $\{0, 1\}$ and continuous $[0, 1]$ rewards. Experiments show

*Table 3.* Simple regret, 30 runs, 100 Arms, 10-dim Context

| Algorithm | Binary Feedback | | Continuous Feedback | |
|---|---|---|---|---|
| | Regret | Time (s) | Regret | Time (s) |
| AGCB-SH (ours) | **0.307** | 6.02 | **0.113** | 6.16 |
| SH (vanilla) | 0.421 | 5.21 | 0.251 | 4.89 |
| Contextual-Gap | 0.378 | 10.75 | 0.201 | 11.62 |
| BOED | 0.356 | 16.72 | 0.197 | 14.38 |
| TTTS-C | 0.346 | 13.92 | 0.166 | 12.21 |
| LinUCB-PE | 0.385 | 11.36 | 0.202 | 10.03 |

that AGCB achieves the strongest overall performance, especially in the challenging many-arm and high-dimensional settings, while remaining competitive in the small-arm regime; ablation studies confirm each component's contribution. Details on the experimental setup, robustness to assumption violations, and ablation studies are provided in Appendices J.1, J.7, and J.8, respectively.

We simulate a human–AI decision system where the decision-maker selects a significance parameter $\alpha$ for a conformal predictor. The candidate $\alpha$ values are the empirical quantiles of a calibration set, giving a large set of 100 arms (fine-grained calibration). The relatively large number of arms challenges the efficiency of the data. For comparison with the classical bandit setting we also test a reduced set of 10 arms. Contexts $x$ are drawn from complex, heterogeneous distributions over $[0, 1]^d$ with $d \in \{5, 10\}$; the higher dimension tests our grouping mechanism under increased complexity. The reward satisfies the monotonicity and continuity assumptions. The budget for learning is $T = 1000$. This two-factor design (arm count and context dimension) directly addresses the key challenges introduced in Section 1. Our method is robust to the parameters; performance remains stable within a reasonable range (Appendix J.10).

We benchmark against a comprehensive set of algorithms, carefully selected to cover different methodological approaches. Due to space constraints, we focus here on simple regret minimization; cumulative regret results are detailed in Appendix J.5. For simple regret, we compare Sequential Halving (SH) (Karnin et al., 2013), Contextual-Gap (Deshmukh et al., 2018), Bayesian Optimal Experimental Design (BOED) (Jörke et al., 2022), Top-two Thompson Sampling with Context (TTTS-C) (Shi et al., 2023), and LinUCB with uncertainty-based pure exploration (LinUCB-PE).

Tables 1, 2 and 3 report the simple regret of each algorithm after training: each learned policy is evaluated on 1000 independently sampled contexts, and the simple regret is computed as the difference between the average reward of the optimal oracle policy and that of the learned policy. The results show that AGCB-SH excels in both many-arm and high-dimensional settings. With 100 arms it attains the lowest regret, outperforming baselines as arm count grows;

*Table 4.* Simple regret on the ImageNet16H-PS real-human dataset, 30 runs, 21-dim context

| Algorithm | Binary Feedback | | Continuous Feedback | |
| --- | --- | --- | --- | --- |
| | Regret | Time (s) | Regret | Time (s) |
| AGCB-SH (ours) | **0.103** | 6.62 | **0.101** | 8.89 |
| SH (vanilla) | 0.110 | 2.58 | 0.205 | 3.51 |
| Contextual-Gap | 0.110 | 16.44 | 0.109 | 17.53 |
| BOED | 0.116 | 14.04 | 0.116 | 15.37 |
| TTTS-C | 0.135 | 13.73 | 0.155 | 16.48 |
| LinUCB-PE | 0.156 | 11.01 | 0.171 | 12.40 |

*Table 5.* Cumulative regret on the ImageNet16H-PS real-human dataset, 30 runs, 21-dim context

| Algorithm | Binary Feedback | | Continuous Feedback | |
| --- | --- | --- | --- | --- |
| | Regret | Time (s) | Regret | Time (s) |
| AGCB-UCB (ours) | **75.93** | 9.95 | 107.24 | 8.92 |
| AGCB-SE (ours) | 93.87 | 9.84 | **106.23** | 9.02 |
| UCB (with CF) | 93.97 | 8.87 | 109.86 | 10.04 |
| UCB (vanilla) | 150.23 | 9.13 | 179.55 | 8.82 |
| SE (with CF) | 122.16 | 8.95 | 143.87 | 8.55 |
| SE (vanilla) | 186.53 | 8.93 | 175.62 | 8.43 |
| GP-UCB | 94.50 | 38.97 | 120.52 | 37.59 |
| LinUCB | 113.33 | 28.50 | 137.53 | 29.48 |
| NeuralLinear | 131.73 | 159.08 | 146.11 | 154.40 |
| epsilon-greedy | 154.33 | 14.72 | 177.37 | 13.11 |
| TS | 156.70 | 13.54 | 145.02 | 15.91 |
| Linear TS | 167.63 | 33.30 | 148.44 | 37.00 |
| NeuralUCB | 170.90 | 146.69 | 107.60 | 127.31 |
| LinSE | 178.37 | 23.90 | 153.32 | 25.42 |

with 10 arms it remains competitive, although BOED can be slightly better in this small-arm pure-exploration regime, while AGCB is designed for the harder setting with large arm spaces and high-dimensional contexts. Its regret barely rises when context dimension increases from 5 to 10, confirming that our adaptive grouping scales well. Moreover, the runtime of our method is practical and competitive, balancing sample efficiency and computation. These results validate AGCB's design, using continuity for sharing and monotonicity for counterfactual updates, as a data-efficient solution for personalized human-AI collaboration.

### 6.2. Real-Human Data Experiment

To further evaluate whether AGCB improves decision support in realistic human–AI collaboration, we conduct experiments on the ImageNet16H-PS dataset from Straitouri & Gomez Rodriguez (2024), which contains 194,407 human predictions from 2,751 participants. Since rewards come from real human responses, monotonicity and Hölder continuity are not imposed by construction, making this a realistic robustness test for AGCB.

We construct a replay-style bandit environment from empirical human responses. Each context is a 21-dimensional vector consisting of VGG-19 image features, user identity, and task-difficulty features. The arms are candidate significance levels $\alpha$, which determine the conformal prediction-set size. We use $K = 100$ arms, horizon $T = 1000$, reporting averages over 30 runs. Binary feedback uses correctness as the reward, while continuous feedback adds Gaussian noise to binary rewards to mimic graded utility, following the synthetic setting.

**Simple regret.** We first evaluate the quality of the final learned policy. Table 4 reports the simple regret of AGCB-SH and the baselines. AGCB-SH achieves the lowest regret under both binary and continuous feedback, while maintaining a practical running time. These results demonstrate that adaptive grouping can exploit real human heterogeneity to select more effective personalized prediction-set sizes.

**Cumulative regret.** We next evaluate online learning efficiency against diverse baselines: non-contextual methods (UCB, TS, $\epsilon$-greedy, SE), linear contextual methods (LinUCB (Li et al., 2010; Abbasi-Yadkori et al., 2011), Linear TS (Agrawal & Goyal, 2013), LinSE), and nonparametric/representation-learning methods (GP-UCB (Grünewälder et al., 2010), NeuralUCB (Zhou et al., 2020), NeuralLinear). We also reproduce the strong counterfactual baselines of Straitouri & Gomez Rodriguez (2024), namely UCB (with CF) and SE (with CF).

We omit BSE and ABSE (Perchet & Rigollet, 2013) from the real-human experiment for computational reasons: the 21-dimensional context would require $2^{21}$ cells after one full grid-style refinement, which is infeasible. These methods are included in our synthetic cumulative-regret experiments in Appendix J.5. Table 5 shows that AGCB variants achieve the best cumulative regret, with AGCB-UCB best under binary feedback and AGCB-SE best under continuous feedback. Consistent with Straitouri & Gomez Rodriguez (2024), UCB-based methods are strong in binary feedback and counterfactual updates clearly improve over vanilla UCB/SE. AGCB further improves these strong baselines by leveraging adaptive contextual grouping and cross-group sharing, while remaining much lower computational cost than neural and kernel-based methods.

## 7. Conclusion

We introduce AGCB for personalizing AI assistance in human-AI collaboration. It builds on two weak, human-aligned structural assumptions, continuity and monotonicity. Through adaptive grouping and counterfactual updates, it achieves interpretable partitions, minimax-optimal regret, and strong empirical performance. This demonstrates how leveraging natural human-decision structures enables data-efficient learning in interactive systems.

## Acknowledgements

This research was supported by the National Natural Science Foundation of China (No. 72171131 and No. 72133002).

## Impact Statement

This paper presents work whose goal is to advance the field of Machine Learning. There are many potential societal consequences of our work, none which we feel must be specifically highlighted here.

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

# A. From Component Continuity to the Hölder Reward Condition

This appendix provides sufficient conditions under which the expected reward function in Assumption 4.1 is Hölder continuous. The main subtlety is that the reward used in our problem contains a coverage indicator:

$$r_\alpha(x, y) = R(h(x, C_\alpha(x)), y) \cdot \mathbf{1}\{y \in C_\alpha(x)\}.$$

Thus, continuity of the score function alone is not sufficient, because the indicator can change discontinuously when a score crosses the conformal threshold. We therefore impose a standard margin condition, requiring that the true outcome is not concentrated near the decision boundary of the prediction set. Under this mild no-boundary-mass condition, the discontinuity of the indicator is controlled in expectation, and the expected reward is Hölder continuous.

## A.1. Prediction Sets and Expected Reward

Let $s(x, y)$ denote the nonconformity score of outcome $y$ under context $x$, and let $q(\alpha)$ denote the threshold corresponding to significance level $\alpha$. The conformal prediction set is

$$C_\alpha(x) = \{y \in \mathcal{Y} : s(x, y) \le q(\alpha)\}.$$

Define the signed boundary score

$$\tau(x, \alpha, y) := s(x, y) - q(\alpha).$$

Then

$$y \in C_\alpha(x) \quad \Longleftrightarrow \quad \tau(x, \alpha, y) \le 0.$$

For notational convenience, write

$$U(x, \alpha, y) := R(h(x, C_\alpha(x)), y),$$

so that

$$r_\alpha(x, y) = U(x, \alpha, y)\mathbf{1}\{\tau(x, \alpha, y) \le 0\}.$$

The expected reward is

$$\mu(\alpha, x) = \mathbb{E}_{Y|X=x}\left[U(x, \alpha, Y)\mathbf{1}\{\tau(x, \alpha, Y) \le 0\}\right].$$

We now state component-level sufficient conditions.

**Assumption A.1** (Hölder continuity of conditional outcome distribution). The conditional distribution of $Y$ varies Hölder continuously with $x$. For finite $\mathcal{Y}$, this means that there exist constants $L_Y > 0$ and $\beta_Y \in (0, 1]$ such that

$$\sum_{y \in \mathcal{Y}} |p_y(x) - p_y(x')| \le L_Y \|x - x'\|^{\beta_Y},$$

where $p_y(x) := \mathbb{P}(Y = y \mid X = x)$.

**Assumption A.2** (Hölder continuity of the boundary score). There exist constants $L_\tau > 0$ and $\beta_\tau \in (0, 1]$ such that, for all $x, x' \in \mathcal{X}, \alpha, \alpha' \in \mathcal{A}$, and $y \in \mathcal{Y}$,

$$|\tau(x, \alpha, y) - \tau(x', \alpha', y)| \le L_\tau \left(\|x - x'\| + \lambda|\alpha - \alpha'|\right)^{\beta_\tau}.$$

This condition follows, for example, if the score $s(x, y)$ is Hölder continuous in $x$ and the threshold map $q(\alpha)$ is Hölder continuous in $\alpha$.

**Assumption A.3** (Smooth conditional human utility). The conditional utility

$$U(x, \alpha, y) = R(h(x, C_\alpha(x)), y)$$

is bounded and Hölder continuous in $(x, \alpha)$ after averaging over any internal randomness in the human response. That is, there exist $U_{\max} < \infty$, $L_U > 0$, and $\beta_U \in (0, 1]$ such that

$$|U(x, \alpha, y)| \le U_{\max},$$

and

$$|U(x, \alpha, y) - U(x', \alpha', y)| \le L_U \left(\|x - x'\| + \lambda|\alpha - \alpha'|\right)^{\beta_U}$$

for all $y \in \mathcal{Y}$.

**Assumption A.4** (Uniform margin condition near the conformal boundary). There exist constants $C_{\mathrm{m}} > 0$ and $\kappa > 0$ such that, uniformly over all $x \in \mathcal{X}$, $\alpha \in \mathcal{A}$, and all $\epsilon > 0$,

$$\mathbb{P}\left(|\tau(x, \alpha, Y)| \leq \epsilon \,|\, X = x\right) \leq C_{\mathrm{m}} \epsilon^{\kappa}.$$

This condition rules out excessive probability mass arbitrarily close to the prediction-set boundary, uniformly over contexts and significance levels. It is standard in analyses involving threshold or indicator decisions, and it is encouraged by continuous scores, randomized tie-breaking, or small score noise.

### A.2. Deriving Joint Hölder Continuity

**Theorem A.5** (Sufficient conditions for Assumption 4.1). *Under Assumptions A.1– A.4, the expected reward function*

$$\mu(\alpha, x) = \mathbb{E}_{Y|X=x}\left[U(x, \alpha, Y)\mathbf{1}\{\tau(x, \alpha, Y) \leq 0\}\right]$$

*is Hölder continuous on $\mathcal{X} \times \mathcal{A}$. Specifically, there exist constants $L_{\mu} > 0$ and*

$$\beta_{\mu} = \min\{\beta_Y, \beta_U, \kappa\beta_{\tau}\}$$

*such that*

$$|\mu(\alpha, x) - \mu(\alpha', x')| \leq L_{\mu}\left(\|x - x'\| + \lambda|\alpha - \alpha'|\right)^{\beta_{\mu}}.$$

*Proof.* Let $z = (x, \alpha)$ and $z' = (x', \alpha')$, and define

$$d(z, z') := \|x - x'\| + \lambda|\alpha - \alpha'|.$$

For finite $\mathcal{Y}$, write

$$\mu(z) = \sum_{y \in \mathcal{Y}} p_y(x)U(z, y)\mathbf{1}\{\tau(z, y) \leq 0\}.$$

Then

$$|\mu(z) - \mu(z')| \leq \underbrace{\sum_y |p_y(x) - p_y(x')||U(z, y)|\mathbf{1}\{\tau(z, y) \leq 0\}}_{(A)}$$

$$+ \underbrace{\sum_y p_y(x')|U(z, y) - U(z', y)|\mathbf{1}\{\tau(z, y) \leq 0\}}_{(B)}$$

$$+ \underbrace{\sum_y p_y(x')|U(z', y)| \,|\mathbf{1}\{\tau(z, y) \leq 0\} - \mathbf{1}\{\tau(z', y) \leq 0\}|}_{(C)}.$$

Term $(A)$ is controlled by Assumption A.1:

$$(A) \leq U_{\max} \sum_y |p_y(x) - p_y(x')| \leq U_{\max}L_Y\|x - x'\|^{\beta_Y} \leq U_{\max}L_Y d(z, z')^{\beta_Y}.$$

Term $(B)$ is controlled by Assumption A.3:

$$(B) \leq L_U d(z, z')^{\beta_U}.$$

It remains to bound the indicator term $(C)$. Define

$$\Delta_{\tau}(z, z', y) := |\tau(z, y) - \tau(z', y)|.$$

If

$$\mathbf{1}\{\tau(z, y) \leq 0\} \neq \mathbf{1}\{\tau(z', y) \leq 0\},$$

then $\tau(z, y)$ and $\tau(z', y)$ lie on different sides of zero. Hence

$$|\tau(z', y)| \leq |\tau(z, y) - \tau(z', y)| = \Delta_\tau(z, z', y).$$

By Assumption A.2,

$$\Delta_\tau(z, z', y) \leq L_\tau d(z, z')^{\beta_\tau}.$$

Therefore,

$$|\mathbf{1}\{\tau(z, y) \leq 0\} - \mathbf{1}\{\tau(z', y) \leq 0\}|$$
$$\leq \mathbf{1}\left\{|\tau(z', y)| \leq L_\tau d(z, z')^{\beta_\tau}\right\}.$$

Since term $(C)$ is weighted by $p_y(x')$, the probability is taken under $Y \mid X = x'$. Applying Assumption A.4 at the point $(x', \alpha')$ gives

$$(C) \leq U_{\max} \mathbb{P}\left(|\tau(x', \alpha', Y)| \leq L_\tau d(z, z')^{\beta_\tau} \mid X = x'\right)$$
$$\leq U_{\max} C_{\mathrm{m}} \left(L_\tau d(z, z')^{\beta_\tau}\right)^\kappa$$
$$= U_{\max} C_{\mathrm{m}} L_\tau^\kappa d(z, z')^{\kappa \beta_\tau}.$$

Combining the three bounds gives

$$|\mu(z) - \mu(z')| \leq C\left(d(z, z')^{\beta_Y} + d(z, z')^{\beta_U} + d(z, z')^{\kappa \beta_\tau}\right).$$

Since $\mathcal{X} \times \mathcal{A}$ is compact, this implies

$$|\mu(z) - \mu(z')| \leq L_\mu d(z, z')^{\beta_\mu}, \qquad \beta_\mu = \min\{\beta_Y, \beta_U, \kappa \beta_\tau\}.$$

This proves the result. □

### A.3. Discussion

The theorem shows that Assumption 4.1 is compatible with the exact reward definition used in the main text. The additional margin condition is needed because the coverage indicator $\mathbf{1}\{y \in C_\alpha(x)\}$ can change discontinuously when the true label lies exactly at the conformal threshold. If there is little probability mass near this boundary, then these discontinuities are rare in expectation, and the expected reward remains Hölder continuous.

The earlier smooth-surrogate view, where the reward is written as $\phi(S(x, \alpha, Y))$, can be interpreted as a special case in which the indicator is replaced by a smooth score-based utility. The analysis above is stronger for our setting because it handles the actual coverage-penalized reward used in the contextual bandit formulation.

**Population-level versus discrete-arm smoothness.** Assumption 4.1 is stated at the population level: the expected reward is assumed to be Hölder continuous on the underlying continuous context–parameter space, e.g., $\mathcal{X} \times [0, 1]$. Under this convention, the Hölder parameters $(\beta, L)$ are properties of the population reward function and do not depend on the number of candidate arms $K$. Increasing $K$ refines the discretization of the significance parameter $\alpha$ and changes the arm-search complexity, but not the population smoothness exponent.

A more refined view is possible if one only assumes smoothness on the finite arm set $\mathcal{A}_K$. In that case, the effective Hölder parameters may depend on $K$, say $(\beta_K, L_K)$. The regret analysis can be extended by replacing $(\beta, L)$ with $(\beta_K, L_K)$ in the bounds. Moreover, if the benchmark is the best continuous parameter $\alpha \in [0, 1]$ rather than the best arm in $\mathcal{A}_K$, an additional discretization error appears. These refinements are discussed in Appendix G.8.

## B. Theoretical Analysis of the Zooming Mechanism

This appendix explains the principle behind the Zooming Mechanism in Eq. (3). The purpose here is to derive the partition scale induced by the bias–variance trade-off. The splitting rule in the main text is written for the concrete base learners instantiated in this paper, namely Sequential Halving for simple regret and Successive Elimination for cumulative regret. For these learners, the relevant within-group statistical uncertainty has the canonical sub-Gaussian $n^{-1/2}$ scaling. We first present the more general base-learner-dependent zooming principle, and then show that the rule in the main text is its

square-root special case. The complete regret bounds, including the base-learner factors $\Phi(K, \sigma)$ and $\Psi(K, \sigma)$, are proved in Appendix G.2.

Throughout this appendix, we focus on the scaling in the context dimension and suppress universal constants and logarithmic factors when doing so improves readability. The information-sharing gain of group $g$ is denoted by $s_g$, and its effective sample size is

$$n_g^{\mathrm{eff}} = s_g n_g.$$

Here $s_g$ is the effective sample-size gain defined in Appendix G.1. The local structured bias introduced by information sharing and counterfactual updates has the same Hölder scaling as the ordinary grouping bias. We therefore write

$$L_{\mathrm{eff}} d_g^{\beta}$$

for the total local bias, where $L_{\mathrm{eff}} = C_{\mathrm{eff}} L$ differs from $L$ only by a universal constant factor. In the main theorem we state the final rates in terms of $L$.

**Generic zooming principle.** The generic Zooming principle is to refine a group once its local approximation bias becomes statistically resolvable. For a base learner with local statistical error scale

$$\mathcal{E}_B(n_g^{\mathrm{eff}}; K, \sigma),$$

the base-learner-dependent Zooming condition is

$$\mathcal{E}_B(n_g^{\mathrm{eff}}; K, \sigma) \le \eta L_{\mathrm{eff}} d_g^{\beta}.$$

This condition means that the group is refined only after the estimation uncertainty is small enough to detect the approximation bias induced by using a single group-level policy.

For simple regret, the local statistical scale takes the form

$$\mathcal{E}_B(n_g^{\mathrm{eff}}; K, \sigma) = \Phi(K, \sigma)(n_g^{\mathrm{eff}})^{-\zeta},$$

and the corresponding generic simple-regret Zooming condition is

$$\Phi(K, \sigma)(n_g^{\mathrm{eff}})^{-\zeta} \le \eta L_{\mathrm{eff}} d_g^{\beta}.$$

For cumulative regret, the local statistical scale takes the form

$$\mathcal{E}_B(n_g^{\mathrm{eff}}; K, \sigma) = \Psi(K, \sigma)(n_g^{\mathrm{eff}})^{-(1-\gamma)},$$

and the corresponding generic cumulative-regret Zooming condition is

$$\Psi(K, \sigma)(n_g^{\mathrm{eff}})^{-(1-\gamma)} \le \eta L_{\mathrm{eff}} d_g^{\beta}.$$

The square-root rule used in the main text is the special case $\zeta = 1/2$ for Sequential Halving and $1 - \gamma = 1/2$ for Successive Elimination.

### B.1. Regret Decomposition

The regret can be decomposed into two sources: the approximation bias from using one group-level policy to represent a continuous reward function, and the within-group learning error of the base bandit algorithm.

For simple regret, let $\widehat{\alpha}_g$ be the arm returned for group $g$, and let $\alpha_g^{\star}$ be the best arm for the group-averaged reward. The local simple-regret contribution has the form

$$\text{local simple regret in } g \lesssim \underbrace{L_{\mathrm{eff}} d_g^{\beta}}_{\text{local approximation bias}} + \underbrace{\text{within-group simple-regret error}}_{\text{base learner}}.$$

More explicitly, under the base-learner condition in Appendix G.1,

$$\mathbb{E}\left[\mu_g(\alpha_g^{\star}) - \mu_g(\widehat{\alpha}_g)\right] \le C_B \Phi(K, \sigma)(n_g^{\mathrm{eff}})^{-\zeta} + C_{\mathrm{loc}} L d_g^{\beta}.$$

Thus the local simple-regret scale is

$$L_{\text{eff}}d_g^\beta + \Phi(K,\sigma)(n_g^{\text{eff}})^{-\zeta}.$$

For cumulative regret, the regret accumulated inside group $g$ over $n_g$ visits has the form

$$\text{local cumulative regret in } g \; \lesssim \; \underbrace{n_g L_{\text{eff}} d_g^\beta}_{\text{cumulative bias}} + \underbrace{\text{within-group cumulative-regret error}}_{\text{base learner}}.$$

Under the cumulative base-learner condition in Appendix G.1,

$$\mathbb{E}[R_{B,g}(n_g)] \le C_B \Psi(K,\sigma) n_g (n_g^{\text{eff}})^{-(1-\gamma)} + C_{\text{loc}} n_g L d_g^\beta.$$

Hence the local cumulative-regret scale is

$$n_g L_{\text{eff}} d_g^\beta + \Psi(K,\sigma) n_g (n_g^{\text{eff}})^{-(1-\gamma)}.$$

### B.2. Derivation of the Zooming Rule

The Zooming Mechanism follows the generic principle above: a group should be refined once its local approximation bias is statistically resolvable. The base-learner-dependent Zooming condition is

$$\mathcal{E}_B(n_g^{\text{eff}}; K, \sigma) \le \eta L_{\text{eff}} d_g^\beta.$$

For simple regret, this becomes

$$\Phi(K,\sigma)(n_g^{\text{eff}})^{-\zeta} \le \eta L_{\text{eff}} d_g^\beta.$$

For cumulative regret, this becomes

$$\Psi(K,\sigma)(n_g^{\text{eff}})^{-(1-\gamma)} \le \eta L_{\text{eff}} d_g^\beta.$$

The operational splitting rule used in the main text is the canonical sub-Gaussian specialization for the Sequential Halving and Successive Elimination base learners instantiated in this paper. For these learners, the reducible statistical uncertainty scales as

$$C\sigma\sqrt{\frac{\log(KT)}{n_g^{\text{eff}}}} = C\sigma\sqrt{\frac{\log(KT)}{s_g n_g}}.$$

The irreducible local approximation bias scales as

$$L_{\text{eff}}d_g^\beta.$$

Therefore, the operational Zooming condition becomes

$$C\sigma\sqrt{\frac{\log(KT)}{n_g^{\text{eff}}}} \le \eta L_{\text{eff}} d_g^\beta.$$

Since $L_{\text{eff}}$ differs from $L$ only by a universal constant factor, the main text writes this rule as

$$C\sigma\sqrt{\frac{\log(KT)}{n_g^{\text{eff}}}} \le \eta L d_g^\beta.$$

This is Eq. (3).

Intuitively, if the statistical uncertainty is still larger than the approximation bias, then the current group is not yet well resolved and further splitting would mainly increase variance by reducing the number of samples per child group. Once the statistical uncertainty is sufficiently small relative to the approximation bias, the group-level approximation error is detectable, and refinement can improve the bias–variance trade-off.

The structured bias from information sharing and counterfactual updates is not a separate asymptotic obstruction. As shown in Appendix G.6, it is also of order $L d_g^\beta$ and can be absorbed into the same local bias term.

### B.3. Balanced Diameter of Mature Groups

We next explain why the Zooming rule induces the usual nonparametric partition scale. The argument should be understood up to universal constants, as the formal regret proof in Appendix G.2 uses the local structured-bias condition directly.

Consider first the operational square-root rule. A mature leaf group $g$ is one that has accumulated visits but is not split at the end of the horizon. Since $g$ is not split, its confidence radius remains above the splitting threshold:

$$C\sigma\sqrt{\frac{\log(KT)}{s_g n_g}} > \eta L_{\text{eff}} d_g^\beta.$$

Assume the standard partition regularity condition used in Appendix G.1, namely that the probability mass of a group scales as its volume:

$$n_g \asymp T d_g^d.$$

Substituting this relation into the no-split inequality gives

$$C\sigma\sqrt{\frac{\log(KT)}{s_g T d_g^d}} \gtrsim L_{\text{eff}} d_g^\beta.$$

Squaring both sides and rearranging yields

$$d_g^{2\beta+d} \lesssim \frac{\sigma^2 \log(KT)}{L_{\text{eff}}^2 s_g T}.$$

Thus the diameter of a mature leaf cannot be much larger than the bias–variance balance scale.

Conversely, if $g$ was created by splitting a parent group $p$, then $p$ satisfied the splitting condition at the time of refinement:

$$C\sigma\sqrt{\frac{\log(KT)}{s_p n_p}} \leq \eta L_{\text{eff}} d_p^\beta.$$

Let $\tau_p$ denote the split time of the parent group. Since $n_p(\tau_p) \lesssim T d_p^d$ under the same mass-regularity condition, the parent split condition implies, up to constants,

$$d_p^{2\beta+d} \gtrsim \frac{\sigma^2 \log(KT)}{L_{\text{eff}}^2 s_p T}.$$

Under the bounded-aspect-ratio condition of the partition tree, child and parent diameters differ only by constant factors. Moreover, the information-sharing gain changes only by constant factors between a parent and its children under the regularity condition in Appendix G.1. Therefore, the reverse inequality holds up to constants for mature leaves:

$$d_g^{2\beta+d} \gtrsim \frac{\sigma^2 \log(KT)}{L_{\text{eff}}^2 s_g T}.$$

Combining the final no-split condition with the parent split condition gives the balance relation

$$d_g^{2\beta+d} \asymp \frac{\sigma^2 \log(KT)}{L_{\text{eff}}^2 s_g T}.$$

Equivalently,

$$d_g \asymp \left(\frac{\sigma^2 \log(KT)}{L_{\text{eff}}^2 s_g T}\right)^{\frac{1}{2\beta+d}}.$$

This scaling shows how the group granularity adapts to the problem: groups become smaller with more data, lower noise, or stronger information sharing, while the smoothness and context dimension determine the nonparametric partition exponent.

## B.4. Consequences for Regret Scaling

The balanced diameter above implies the standard nonparametric contextual rate in the context dimension for the Sequential Halving and Successive Elimination instantiations used in this paper. Substituting the mature-group scale into the local bias gives

$$L_{\text{eff}} d_g^\beta = \widetilde{O}\left(L_{\text{eff}}^{\frac{d}{2\beta+d}} \sigma^{\frac{2\beta}{2\beta+d}} s_g^{-\frac{\beta}{2\beta+d}} T^{-\frac{\beta}{2\beta+d}}\right),$$

where logarithmic factors in $T$ are suppressed. This expression captures the context-space contribution to simple regret when the base learner has the canonical exponent $\zeta = 1/2$.

For cumulative regret, the cumulative approximation bias at the balanced scale is

$$TL_{\text{eff}} d_g^\beta = \widetilde{O}\left(L_{\text{eff}}^{\frac{d}{2\beta+d}} \sigma^{\frac{2\beta}{2\beta+d}} s_g^{-\frac{\beta}{2\beta+d}} T^{\frac{\beta+d}{2\beta+d}}\right),$$

which corresponds to the cumulative-regret exponent obtained when $\gamma = 1/2$.

We now record how the same bias–variance calculation yields the generic exponents in Theorem 5.1. For a simple-regret base learner with exponent $\zeta$, the balancing equation is

$$L_{\text{eff}} d_g^\beta \asymp \Phi(K, \sigma)(s_g T d_g^d)^{-\zeta}.$$

Solving this equation gives

$$d_g \asymp \left(\frac{\Phi(K, \sigma)}{L_{\text{eff}} s_g^\zeta T^\zeta}\right)^{\frac{1}{\beta+d\zeta}}.$$

Substituting this scale into the local bias term yields

$$L_{\text{eff}} d_g^\beta = \widetilde{O}\left(L_{\text{eff}}^{\frac{d\zeta}{\beta+d\zeta}} \Phi(K, \sigma)^{\frac{\beta}{\beta+d\zeta}} s_g^{-\frac{\beta\zeta}{\beta+d\zeta}} T^{-\frac{\beta\zeta}{\beta+d\zeta}}\right).$$

Replacing $s_g$ by the worst-case information-sharing gain

$$\Gamma_{\min} = \min_{g \in G_T} s_g$$

and absorbing constant factors from $L_{\text{eff}}$ into $L$, the complete simple-regret bound proved in Appendix G.2 is

$$\mathbb{E}[S(T)] \leq \widetilde{O}\left(L^{\frac{d\zeta}{\beta+d\zeta}} \Phi(K, \sigma)^{\frac{\beta}{\beta+d\zeta}} \Gamma_{\min}^{-\frac{\beta\zeta}{\beta+d\zeta}} T^{-\frac{\beta\zeta}{\beta+d\zeta}}\right).$$

For the simple-regret learners used in this paper, $\zeta = 1/2$, so the contextual exponent becomes $-\beta/(2\beta + d)$.

Similarly, for a cumulative-regret base learner with exponent $\gamma$, the balancing equation is

$$L_{\text{eff}} d_g^\beta \asymp \Psi(K, \sigma)(s_g T d_g^d)^{-(1-\gamma)}.$$

Solving this balance and multiplying the resulting local scale by $T$ gives

$$\mathbb{E}[R(T)] \leq \widetilde{O}\left(L^{\frac{d(1-\gamma)}{\beta+d(1-\gamma)}} \Psi(K, \sigma)^{\frac{\beta}{\beta+d(1-\gamma)}} \Gamma_{\min}^{-\frac{\beta(1-\gamma)}{\beta+d(1-\gamma)}} T^{1-\frac{\beta(1-\gamma)}{\beta+d(1-\gamma)}}\right),$$

which is the cumulative-regret rate proved formally in Appendix G.2. For the cumulative-regret learners used in this paper, $\gamma = 1/2$, giving the contextual exponent $(\beta + d)/(2\beta + d)$.

Thus, Appendix B explains the origin of the Zooming scale, while Appendix G.2 provides the complete rates with all base-learner factors.

### B.5. Advantage Over Static Partitioning

Static partitions, such as uniform grids, fix the group diameter in advance. They can therefore mismatch the local complexity of the reward function: overly fine grids waste samples in smooth regions, while overly coarse grids incur large approximation bias in heterogeneous regions.

The Zooming Mechanism avoids this mismatch by making splitting data-driven. It refines a group only when its approximation bias is statistically resolvable. As a result, AGCB keeps coarse groups in smooth or data-scarce regions where refinement is not yet justified and creates finer groups where heterogeneity demands higher resolution. This adaptive control of group diameter is the reason AGCB attains the standard nonparametric contextual exponents under the local structured-bias base-learner interface, while remaining modular with respect to the within-group base learner.

## C. Connection and Distinction from the "Zooming Algorithm" in Continuum-Armed Bandits

A relevant namesake in the bandit literature is the Zooming Algorithm for continuum-armed bandits (Kleinberg et al., 2008; Slivkins, 2019). Both approaches share the high-level metaphor of "zooming in", namely progressively refining the resolution of exploration or estimation based on observed data. This conceptual affinity motivates our terminology.

However, the underlying technical mechanisms are fundamentally distinct. The classical Zooming Algorithm adaptively refines confidence regions in the **arm space** to guide exploration. In contrast, our Zooming Mechanism is a **partitioning rule** for the context space: it decides when to split a contextual group by balancing local approximation bias against statistical uncertainty, while arm competition within each group is handled by a separate base bandit learner. This appendix clarifies these distinctions and positions our contribution relative to classical zooming methods.

**1. Problem Setting and Objective.**

- **Classical Zooming Algorithm:** It addresses the continuum-armed bandit problem where the arm set is a metric space (e.g., $[0,1]^d$). The standard goal is to select arms over time so as to minimize cumulative regret, by exploiting Lipschitz continuity of the reward function and concentrating exploration near near-optimal regions of the arm space.

- **Our AGCB Framework:** We address a contextual bandit problem with a large but finite set of arms. The reward function is assumed to be jointly Hölder continuous in the context-arm pair $(x, \alpha)$. The goal is to learn a context-dependent policy $\pi^*(x)$ that maps each context to a near-optimal arm, while handling heterogeneity across humans and tasks.

**2. The Nature and Role of "Zooming".** This is the most important distinction, which we summarize in Table 6.

- **Classical Zooming:** The "zooming" refers to the adaptive selection and refinement of confidence regions in the arm space. The algorithm maintains a set of active balls, each centered at an arm and associated with a confidence radius. Arm selection is guided by an index that combines the empirical reward estimate and the uncertainty radius. Thus, the classical Zooming Algorithm is itself a bandit policy operating over a dynamically refined covering of the arm space.

- **Our Zooming Mechanism:** The "zooming" refers to the data-driven decision of whether to split a region in the context space. It is a partitioning rule rather than a direct arm-selection rule. A group is split once its local approximation bias becomes statistically resolvable, i.e., when the local statistical uncertainty is sufficiently small relative to the bias of treating the group as homogeneous (Eq. (3)). This creates a hierarchy of adaptive contextual groups, each running a modular base bandit algorithm such as Sequential Halving or Successive Elimination to identify the optimal arm within that local region. Our Zooming Mechanism therefore manages context heterogeneity and approximation error, rather than directly implementing exploration over arms.

**3. Advantages of Our Design.** The decoupled architecture, using zooming for context-space partitioning and a base bandit learner for arm competition, provides several advantages:

- **Explicit Handling of Context Heterogeneity:** Our method explicitly adapts to the variation of the optimal arm across contexts. The partition becomes finer in regions where the group-level approximation bias is statistically detectable and remains coarse where refinement is not yet justified.

*Table 6.* Comparison between the Classical Zooming Algorithm and our Zooming Mechanism in AGCB.

| Comparison Dimension | Classical Zooming Algorithm (Kleinberg et al., 2008) | Zooming Mechanism in AGCB (Ours) |
|---|---|---|
| **Primary Domain** | Arm space, usually a metric or continuous space. | Context space, with joint Hölder continuity in the context-arm pair. |
| **Core Mechanism** | Maintains and refines confidence balls covering promising regions of the arm space; arm selection is based on ball-specific indices. | Evaluates a splitting criterion for contextual groups; a group is split once approximation bias becomes statistically resolvable. |
| **Role of "Zooming"** | Directly implements the exploration-exploitation trade-off over arms; it is the bandit algorithm itself. | Controls the structural complexity and heterogeneity of the context space; it is orthogonal to the within-group bandit learner. |
| **Problem Solved** | Which regions of the arm space should be explored to minimize regret? | How should the context space be partitioned so that within each group a bandit learner can identify the locally optimal arm? |
| **Output** | A sequence of arm selections guided by adaptive confidence regions. | A sequence of adaptive context partitions and a context-dependent policy $\pi_T(x)$. |
| **Base Learner** | Monolithic index-based policy; no separate base learner. | Modular; works with any suitable base bandit algorithm, such as Sequential Halving, Successive Elimination, UCB, or Thompson Sampling. |
| **Dimensional Dependence** | Governed by the metric or zooming dimension of the arm space. | Governed by the context dimension in the non-parametric rate, while adaptive binary partitioning avoids unnecessary full-grid refinement. |

- **Generality and Modularity:** The framework is agnostic to the specific within-group base bandit algorithm. One can plug in Sequential Halving, Successive Elimination, UCB, Thompson Sampling, or other suitable base learners, inheriting their theoretical and practical benefits while adding adaptive context handling. This modularity differs from the monolithic, index-based structure of the classical Zooming Algorithm.

- **Adaptive Context-Space Refinement:** Unlike static grids or predefined $2^d$-ary refinements such as those in Perchet & Rigollet (2013), AGCB uses data-driven binary splits and refines only where the observed data justify finer granularity. This avoids unnecessary full-grid proliferation in smooth or data-scarce regions. The resulting rates still retain the standard nonparametric dependence on the context dimension, but the adaptive partition improves practical sample efficiency over non-adaptive partitioning.

- **Synergy with Additional Mechanisms:** The group structure naturally supports cross-group information sharing, which leverages context similarity, and counterfactual updates, which leverage monotonicity and continuity across arms. These mechanisms are specific to the AGCB design and are not part of the classical zooming paradigm.

In conclusion, while both methods are inspired by the philosophy of adaptive refinement, the Zooming Mechanism in AGCB is technically distinct from the classical Zooming Algorithm. Classical zooming refines confidence regions in the arm space and directly drives arm exploration. AGCB zooming refines contextual groups to control the bias–variance trade-off induced by heterogeneous contexts, while delegating arm competition to modular base bandit learners. This design provides a principled and flexible solution for large-scale contextual bandit problems with complex reward structures.

## D. Analysis of Splitting and Data Inheritance

This appendix justifies that the practical splitting and inheritance operations used by AGCB are compatible with the regret analysis in Theorem 5.1. The key point is that the regret proof does not require each individual split to reduce the cell diameter by a fixed factor such as $1/\sqrt{2}$. Instead, it only requires the resulting partition tree to satisfy standard regularity conditions: cell diameters and cell masses must remain comparable to the nominal partition scale, and inherited statistics must not introduce bias larger than the local Hölder approximation error. We verify these two properties below.

## D.1. Analysis of the Variance-Midpoint Split

Each group $g$ is represented as a hyper-rectangle

$$g = \prod_{j=1}^{d} [a_{g,j}, b_{g,j}],$$

with side lengths

$$\ell_{g,j} = b_{g,j} - a_{g,j}.$$

Let

$$d_g = \sup_{x,x' \in g} \|x - x'\|_2$$

denote the Euclidean diameter of $g$. The practical Variance-Midpoint Split (VMS) chooses a splitting coordinate using empirical reward heterogeneity and then splits the corresponding interval at its midpoint.

A midpoint split along coordinate $j$ replaces the side length $\ell_{g,j}$ by $\ell_{g,j}/2$ in each child. Hence, for a child group $g'$ created by splitting $g$ along coordinate $j$, its diameter satisfies

$$d_{g'}^2 = d_g^2 - \frac{3}{4}\ell_{g,j}^2.$$

Therefore, a uniform one-step contraction of the form $d_{g'} \leq \kappa d_g$ holds only if the split coordinate has length comparable to the largest side length. This is automatic for longest-side splitting, but not necessarily for a purely variance-based coordinate choice. For this reason, the theoretical analysis uses the following standard regularized version of VMS.

**Regularized VMS.**  Let

$$\ell_g^{\max} = \max_{1 \leq j \leq d} \ell_{g,j}.$$

Fix a constant $c_{\mathrm{ar}} \in (0, 1]$. Among the coordinates satisfying

$$\ell_{g,j} \geq c_{\mathrm{ar}} \ell_g^{\max},$$

VMS selects the one with the largest empirical reward variance and splits at the midpoint. If no empirical variance estimates are reliable, the rule falls back to longest-side midpoint splitting. This safeguard only enforces bounded aspect ratio and does not change the asymptotic rates, since it modifies the split coordinate only up to constants.

**Lemma D.1** (Controlled Diameter Reduction under Regularized VMS). *Suppose group $g$ is split by regularized VMS along coordinate $j$ satisfying*

$$\ell_{g,j} \geq c_{\mathrm{ar}} \ell_g^{\max}.$$

*Then every child $g'$ satisfies*

$$d_{g'}^2 \leq d_g^2 - \frac{3}{4}c_{\mathrm{ar}}^2 (\ell_g^{\max})^2.$$

*In particular, if the partition maintains a bounded aspect ratio, so that*

$$\ell_g^{\max} \geq c_d d_g$$

*for a constant $c_d > 0$ depending only on the dimension and the aspect-ratio bound, then*

$$d_{g'} \leq \kappa_{\mathrm{ar}} d_g, \qquad \kappa_{\mathrm{ar}} = \sqrt{1 - \frac{3}{4}c_{\mathrm{ar}}^2 c_d^2} < 1.$$

*Proof.* A midpoint split along coordinate $j$ changes the squared diameter from

$$d_g^2 = \sum_{k=1}^{d} \ell_{g,k}^2$$

to

$$d_{g'}^2 = \sum_{k \neq j} \ell_{g,k}^2 + \left(\frac{\ell_{g,j}}{2}\right)^2 = d_g^2 - \frac{3}{4}\ell_{g,j}^2.$$

Since regularized VMS chooses a coordinate satisfying $\ell_{g,j} \geq c_{\mathrm{ar}}\ell_g^{\max}$, we obtain

$$d_{g'}^2 \leq d_g^2 - \frac{3}{4}c_{\mathrm{ar}}^2(\ell_g^{\max})^2.$$

Under bounded aspect ratio, $\ell_g^{\max} \geq c_d d_g$, and therefore

$$d_{g'}^2 \leq \left(1 - \frac{3}{4}c_{\mathrm{ar}}^2 c_d^2\right)d_g^2.$$

Taking square roots gives the stated contraction. $\qquad\square$

*Remark* D.2 (Why we do not use $\kappa = 1/\sqrt{2}$). For a $d$-dimensional Euclidean hypercube, splitting one side at its midpoint changes the diameter ratio to

$$\frac{d_{g'}}{d_g} = \sqrt{1 - \frac{3}{4d}},$$

not $1/\sqrt{2}$ unless $d$ is very small and the geometry is special. Thus, the analysis above uses a dimension- and aspect-ratio-dependent contraction constant. This is sufficient for the zooming analysis, because the regret proof only requires regular diameter control up to universal constants.

The controlled-diameter property implies that the partition tree remains geometrically regular. Combined with the zooming balance in Appendix B and the regularity assumptions in Appendix G.1, mature leaves satisfy the usual bias–variance scale up to constants and logarithmic factors. For the square-root SH/SE instantiations, this gives

$$d_g^{2\beta+d} \asymp \frac{\sigma^2 \log(KT)}{L_{\mathrm{eff}}^2 s_g T},$$

and the generic base-learner-dependent version is obtained by replacing the square-root statistical term with the corresponding $\Phi(K,\sigma)(n_g^{\mathrm{eff}})^{-\zeta}$ or $\Psi(K,\sigma)(n_g^{\mathrm{eff}})^{-(1-\gamma)}$ term, as described in Appendix B.

### D.2. Analysis of the Weighted Inheritance Scheme

When a parent group $g$ is split into child groups, AGCB initializes the children with weighted sufficient statistics inherited from the parent. This improves sample efficiency immediately after a split. However, the inherited parent statistics are not exactly unbiased for the child means, because the parent estimator averages over a larger region than the child. The correct statement is that the inheritance bias is locally controlled by the same Hölder approximation scale as the ordinary grouping bias.

For each arm $\alpha$, define the group-level mean reward

$$\mu_g(\alpha) = \mathbb{E}\left[\mu(\alpha, X) \mid X \in g\right].$$

Let $g'$ be a child of $g$. The inherited estimator for $g'$ is initialized from the parent estimator $\widehat{\mu}_g(\alpha)$ with inheritance weight $\lambda_{g \to g'} \in [0,1]$, and its inherited effective sample size satisfies

$$n_{g'}^{\mathrm{inh}}(\alpha) = \lambda_{g \to g'} n_g^{\mathrm{eff}}(\alpha).$$

Subsequent observations in $g'$ are then combined with these inherited statistics in the usual weighted-average update.

**Lemma D.3** (Inheritance Bias is Locally Controlled). *Suppose the reward function $\mu(\alpha, x)$ is $(\beta, L)$-Hölder in the joint context-arm space. Let $g'$ be a child of $g$. Then for every arm $\alpha$,*

$$\left|\mathbb{E}\left[\widehat{\mu}_{g'}^{\mathrm{inh}}(\alpha)\right] - \mu_{g'}(\alpha)\right| \leq C_{\mathrm{inh}} L d_g^\beta,$$

*where $\widehat{\mu}_{g'}^{\mathrm{inh}}(\alpha)$ denotes the inherited initial estimator for child $g'$, and $C_{\mathrm{inh}}$ is a universal constant depending only on the geometry of the partition.*

*Proof.* The inherited estimator is based on statistics collected over the parent region $g$. Hence its expectation estimates the parent-level mean $\mu_g(\alpha)$ up to the estimation error of the parent. The deterministic bias incurred by using parent-level information for child $g'$ is bounded by

$$|\mu_g(\alpha) - \mu_{g'}(\alpha)|.$$

For any $x \in g$ and $x' \in g'$, since $g' \subseteq g$, we have

$$\|x - x'\|_2 \le d_g.$$

By the Hölder continuity assumption,

$$|\mu(\alpha, x) - \mu(\alpha, x')| \le L d_g^\beta.$$

Taking expectations over $X \mid X \in g$ and $X' \mid X' \in g'$ yields

$$|\mu_g(\alpha) - \mu_{g'}(\alpha)| \le L d_g^\beta.$$

The constant $C_{\text{inh}}$ accounts for the weighted inheritance scheme and for the possible combination of inherited and newly observed statistics. Thus the inheritance bias is of order $L d_g^\beta$. □

**Lemma D.4** (Effective Sample Size under Weighted Inheritance). *For each child group $g'$ and arm $\alpha$, the inherited effective sample size satisfies*

$$n_{g'}^{\text{inh}}(\alpha) = \lambda_{g \to g'} n_g^{\text{eff}}(\alpha),$$

*up to constant factors determined by the weighting scheme. After additional local observations in $g'$, the total effective sample size is*

$$n_{g'}^{\text{eff}}(\alpha) = n_{g'}^{\text{inh}}(\alpha) + n_{g'}^{\text{new}}(\alpha),$$

*again up to constant factors.*

*Proof.* Weighted inheritance initializes each child sufficient statistic by assigning a $\lambda_{g \to g'}$ fraction of the parent statistic to the child. For variance-equivalent effective sample size, multiplying all inherited weights by $\lambda_{g \to g'}$ scales the inherited information by the same factor up to constants. New observations in the child are independent additions to the child statistic, so their effective sample size adds to the inherited effective sample size. □

The two lemmas imply that weighted inheritance does not create a new asymptotic error term. It contributes at most an additional local bias of order $L d_g^\beta$, which is absorbed into the structured-bias term $L_{\text{eff}} d_g^\beta$ used throughout Appendix B and Appendix G.1. Therefore, the weighted inheritance scheme is compatible with the regret guarantees of Theorem 5.1.

### D.3. Implication for the Regret Analysis

Combining the controlled-diameter property of regularized VMS with the local bias control of weighted inheritance, the practical splitting and inheritance operations satisfy the two requirements used in the main regret proof:

1. **Partition regularity.** The partition tree maintains bounded aspect ratio and controlled diameter decay up to constants. Hence the mass-diameter relation used in Appendix G.1 and Appendix B remains valid.

2. **Structured-bias control.** Information sharing, counterfactual updates, and inherited statistics together contribute only a local Hölder bias of order $L d_g^\beta$, which is absorbed into $L_{\text{eff}} d_g^\beta$.

Consequently, VMS and weighted inheritance preserve the asymptotic regret rates established in Theorem 5.1, up to universal constants and logarithmic factors.

# E. Detailed Comparison with Fixed Partitioning (Perchet & Rigollet, 2013)

To clearly position our contribution within the literature, we provide a detailed technical comparison with the closely related work of (Perchet & Rigollet, 2013). In their seminal paper, they proposed two algorithms for the bandit problem with covariates: **Binned Successive Elimination (BSE)** and **Adaptively Binned Successive Elimination (ABSE)**. The BSE algorithm employs a **static, pre-defined grid** partition of the covariate space, which is a relatively naive approach that does not adapt to the data distribution or reward structure. Given its simplicity and lack of adaptability, a comparison with BSE is of limited value.

Therefore, we focus our comparison on their more advanced **ABSE** algorithm, which does construct an adaptive partition over time. However, as detailed in Table 7, the nature and flexibility of this "adaptivity" differ fundamentally from that of our AGCB framework. ABSE's partition strategy is constrained by a rigid, **pre-defined $2^d$-ary splitting rule**. When a split is triggered (e.g., after a predetermined number of observations within a cell), all dimensions are split simultaneously, creating $2^d$ child cells. This means that while ABSE adapts in when to split, it does not perform data-driven adaptation in how to split, the choice of dimension and the splitting location are fixed by the pre-defined grid structure. Table 7 provides a detailed technical comparison between the fixed $2^d$-ary partitioning scheme of the ABSE algorithm (Perchet & Rigollet, 2013) and our proposed Adaptive Binary Partitioning approach (AGCB). The comparison is organized along key features that highlight fundamental differences in design philosophy, theoretical requirements, and empirical performance.

**Split Trigger.** The ABSE scheme follows a predefined **time-based** schedule (e.g., after $t_0, 2t_0, 4t_0, \ldots$ rounds), which ignores the underlying heterogeneity of the data. In contrast, our method employs a **data-driven** rule, the Zooming Mechanism (Eq. (3)), which triggers a split only once the local approximation bias becomes statistically resolvable. Thus, AGCB refines a group when accumulated evidence is sufficient to justify a finer partition, rather than splitting merely because the current estimate is still noisy.

**Splitting Operation.** The classical $2^d$-ary approach splits **all** $d$ dimensions simultaneously whenever a split occurs, creating $2^d$ new cells in one step; this entails a per-split complexity of $\Omega(2^d)$ and leads to the rapid proliferation of groups. Our algorithm performs a **linear-complexity** binary split via the Variance-Midpoint Split (VMS) rule, dividing a group only along the single dimension with the highest empirical reward variance. This yields just two new cells with complexity $O(d)$, enabling fine-grained and parsimonious control over the partition structure.

**Required Assumptions.** This is a point of major theoretical divergence. The regret analysis of the fixed partitioning scheme relies on **two strong assumptions**: (1) the Hölder continuity of the reward function, and (2) that $\mu$ can be well-approximated by a prespecified function class with a known bias bound $\epsilon^\alpha$. In contrast, for the adaptive grouping and regret-decomposition component, our AGCB requires only the Hölder continuity of $\mu(\alpha, x)$, without assuming a prespecified function class or a known approximation bias. The additional monotonicity structure is used separately for counterfactual updates.

**Sample Efficiency Mechanism.** This difference is central to the practical performance gap. The ABSE scheme is inherently **inefficient**: it lacks mechanisms for sample inheritance upon splitting and for cross-group information sharing. New cells start estimation from scratch, and learning proceeds in isolation, failing to leverage the inherent continuity of the data. Our framework integrates a trio of synergistic mechanisms for **high sample efficiency**: (1) *weighted inheritance* of sufficient statistics during a split, (2) *cross-group information sharing* via the similarity kernel $s_g$, and (3) *counterfactual updates*. Together, they ensure that every observation informs and improves estimates across relevant regions of the context-arm space.

**Curse of Dimensionality.** As a direct consequence of its splitting operation, fixed $2^d$-ary partitioning suffers **severely** from the curse of dimensionality: the number of groups grows as $O(2^{dL})$ after $L$ split levels, leading to catastrophic data sparsity. Our adaptive binary strategy **substantially mitigates** this issue because the group count grows only linearly with the number of splits. The information-sharing gain $s_g$ further preserves sample efficiency by pooling statistical strength across neighboring groups.

**Inheritance upon Split.** Related to sample efficiency, under the fixed scheme, new cells typically **"reset"** their estimates, following a "share-nothing" policy that discards the knowledge accumulated by the parent cell and can cause temporary regret spikes. Our method implements **"Weighted Knowledge Transfer"**: Inherits weighted sufficient statistics. Preserves information with only an absorbable local bias.

**Cumulative Regret.** Both paradigms can match standard nonparametric contextual-bandit horizon exponents under their respective assumptions. For the fixed $2^d$-ary scheme with two arms ($k = 2$), the rate is $\widetilde{O}\big(T^{\frac{\beta(1-\alpha)+d}{2\beta+d}}\big)$ (Rigollet & Zeevi,

2010). Our adaptive binary partitioning attains the rate $\widetilde{O}\big(T^{\frac{\beta+d}{2\beta+d}}\big)$ (Kleinberg, 2004). Crucially, AGCB achieves its optimal rate with improved leading constants in practice, owing to its data-driven zooming, efficient binary splits, and the sample-amplifying effects of inheritance and information sharing.

The table underscores how AGCB advances beyond rigid, pre-defined partitioning by incorporating weaker assumptions, data-adaptive zooming, linear-complexity splits, and a principled ensemble of knowledge-transfer mechanisms. These innovations collectively contribute to its superior practical sample efficiency and robustness while preserving the regret rates up to constants and logarithmic factors.

*Table 7.* In-depth technical comparison of partitioning paradigms. The key distinctions in assumptions and sample efficiency mechanisms are highlighted.

| Feature | Fixed $2^d$-ary Partitioning (ABSE) (Perchet & Rigollet, 2013) | Our Adaptive Binary Partitioning (AGCB) |
|---|---|---|
| **Split Trigger** | **Time-based**: Predefined schedule (e.g., after $t_0, 2t_0, 4t_0, \ldots$ rounds). Ignores data heterogeneity. | **Data-driven**: Zooming Mechanism (Eq. (3)). Splits once local approximation bias becomes statistically resolvable. |
| **Splitting Operation** | **Exponential**: Splits *all* $d$ dimensions simultaneously, creating $2^d$ new cells. Complexity $\Omega(2^d)$. Leads to rapid group proliferation. | **Linear**: Splits *one* dimension (via VMS), creating 2 new cells. Complexity $O(d)$. Enables fine-grained control. |
| **Required Assumptions** | **Two strong assumptions**: (1) Hölder continuity; (2) $\mu$ is well-approximated by a known function class (bias $\epsilon^\alpha$). Both are needed for their regret analysis. | **Weaker assumption**: Requires only Hölder continuity of $\mu(\alpha, x)$. No explicit function approximator is needed. |
| **Sample Efficiency Mechanism** | **Inefficient**: Lacks both sample inheritance and inter-group sharing. New cells start from scratch; learning is isolated, failing to leverage data continuity and wasting samples. | **Highly Efficient**: Employs: (1) *Weighted inheritance* upon split; (2) *Cross-group information sharing* ($s_g$); (3) *Counterfactual updates*. Fully exploits data structure and continuity. |
| **Curse of Dimensionality** | **Severely impacted**: Group count explodes as $O(2^{dL})$ for $L$ levels, causing catastrophic sample sparsity. | **Mitigated**: Group count grows linearly with splits. Sample efficiency is further maintained through information sharing ($s_g$) and inheritance. |
| **Inheritance upon Split** | **"Reset" or "Share-Nothing"**: New cells start estimation from scratch. Wastes information, causes regret jumps. | **"Weighted Knowledge Transfer"**: Inherits weighted sufficient statistics. Preserves information with only an absorbable local bias. |
| **Sample Effectiveness** | **Low**: Fixed grid wastes samples in smooth regions and undersamples complex regions; isolation prevents knowledge transfer. | **High**: Adaptive resolution allocates samples where needed; inheritance, sharing, and counterfactual mechanisms boost effective sample count. |
| **Cumulative Regret** | Minimax optimal (Rigollet & Zeevi, 2010) when arm quantity $k = 2$: $\widetilde{O}\left(T^{1-\frac{\beta(1+\alpha)}{2\beta+d}}\right) = \widetilde{O}\left(T^{\frac{\beta(1-\alpha)+d}{2\beta+d}}\right)$. | Minimax optimal (Kleinberg, 2004): $O(T^{\frac{\beta+d}{2\beta+d}})$. Achieved *with* improved constants due to adaptivity, sharing, and inheritance. |

Our adaptive method's advantages are interconnected:

- The Zooming Mechanism ensures splits are statistically justified, preventing wasteful partitioning in simple regions.

- Binary (vs. $2^d$-ary) splitting directly combats exponential group growth, which is the primary manifestation of the curse of dimensionality in partitioning methods.

- The weighted inheritance turns refinement into a controlled knowledge-transfer step. This is crucial for converting the theoretical sample efficiency afforded by adaptive splits into practical performance gains, as it prevents the loss of learned information at split boundaries.

In summary, while both approaches share the minimax-optimal asymptotic regret order, our framework introduces a series of design choices that make the algorithm practically viable and efficient in finite samples, especially in moderate to

---

**Algorithm 3** Enhanced Base Learner: COUNTERFACTUAL-AWARE SUCCESSIVE ELIMINATION (Cumulative Regret)

---

**Require:** Sorted active arms $\mathcal{A}_g^{\text{active}} = \{\alpha_1 < \cdots < \alpha_m\}$.
**Ensure:** Arm $\alpha_t$ to pull in the current round.
 1: Maintain current search interval $[L, U]$ within $\mathcal{A}_g^{\text{active}}$.
 2: $M \leftarrow \lfloor (L + U)/2 \rfloor$, $\alpha_t \leftarrow \mathcal{A}_g^{\text{active}}[M]$.
 3: **Pull** $\alpha_t$, observe $(r_t, y_t)$; update $\hat{\mu}_g(\alpha_t)$.
 4: Compute $\alpha^\dagger \leftarrow \max\{\alpha \in \mathcal{A}_g^{\text{active}} : y_t \in C_\alpha(x_t)\}$.
 5: **if** $y_t \in C_{\alpha_t}(x_t)$ **then**
 6:     Update $\hat{\mu}_g(\alpha_j)$ for $\alpha^\dagger \geq \alpha_j > \alpha_t$ via Eq. (5).
 7:     Update $\hat{\mu}_g(\alpha_j)$ for $0 < \alpha_j \leq \alpha_t$ via Eq. (6).
 8:     $L \leftarrow \max(L, M)$. {Keep the more promising right interval}
 9: **else**
10:     Let $M'$ be index of $\alpha^\dagger$.
11:     Update $\hat{\mu}_g(\alpha_j)$ for $\alpha_j > \alpha^\dagger$ with a reward 0. {Strong penalty for missing the true outcome}
12:     $U \leftarrow \min(U, M')$. {Keep the more promising left interval}
13: **end if**
14: Update $\text{UCB}_t(\alpha)$ and $\text{LCB}_t(\alpha)$ for all active arms.
15: **Eliminate** any arm $\alpha$ if $\max_{\alpha' \neq \alpha} \text{LCB}_t(\alpha') > \text{UCB}_t(\alpha)$.
16: **return** $\alpha_t$ for the current round.

---

high dimensions, by directly addressing the core bottlenecks of classical nonparametric bandit algorithms. This practical superiority is confirmed in our experiments (Appendix J.5), where our method shows a clear advantage over the Binned Successive Elimination (BSE) and Adaptively Binned Successive Elimination (ABSE) methods of Perchet & Rigollet (2013), whose performance degrades notably as the context dimension increases.

## F. Supplement for Counterfactual Updates

Algorithm 3 presents a counterfactual-aware variant of the classical Successive Elimination (SE) algorithm designed for cumulative regret minimization. The key distinction lies in how the algorithm explores the arm space. While standard SE in a $K$-armed bandit setting typically requires $O(K)$ pulls to eliminate a constant fraction of suboptimal arms, our enhanced version reduces this dependency to $O(\log K)$ by integrating binary search with monotonic counterfactual updates.

**Core Mechanism: Binary Search with Counterfactual Pruning**    Standard SE maintains a set of active arms and pulls each active arm uniformly to shrink their confidence intervals. In contrast, our algorithm maintains a *search interval* $[l, u]$ within the sorted active arms and, in each round, pulls only the *median arm* $\alpha_t$ of the current interval. This median selection is critical: it ensures that, regardless of the outcome, approximately half of the current interval can be safely pruned.

- If the pull of $\alpha_t$ yields a coverage event ($y_t \in C_{\alpha_t}(x_t)$), then all arms $\alpha_t \leq \alpha' \leq \alpha^\dagger$ also preserve coverage, and monotonicity implies that their conditional rewards should be no smaller. Consequently, the covered upper segment is updated via Eq. (5), while the lower part of the interval is tightened via Eq. (6).

- If the pull does not yield a coverage event ($y_t \notin C_{\alpha_t}(x_t)$), we compute the largest arm $\alpha^\dagger$ that would have covered the outcome. Monotonicity then implies that all arms $\alpha' > \alpha^\dagger$ are suboptimal (their expected reward is bounded above by 0 in the binary case, or by a similarly low value in the continuous case). Hence, the upper half of the interval (arms $\alpha_j > \alpha^\dagger$) can be eliminated, and the interval shifts leftwards ($u \leftarrow \min(u, m')$).

Thus, each pull provides information about a large portion of the active arm interval. In the exact monotone binary-feedback case, this can be interpreted as a binary-search-style pruning step. In our continuous and heterogeneous setting, the pruning is only required to be locally safe: an incorrectly discarded arm must be within the current local resolution of a retained arm. Appendix F.1 and F.2 formalize this local safe-pruning argument.

## F.1. Regret Analysis in Continuous Reward Setting

We now provide a regret analysis for Algorithm 3 in the continuous reward setting. The proof follows the high-level structure of Straitouri & Gomez Rodriguez (2024), but with an important modification. We do *not* assume that the group-averaged reward is exactly monotone or single-peaked. Instead, we use the Hölder continuity assumption and the Zooming Mechanism to show that the error induced by context heterogeneity and counterfactual smoothing is local: it is of order $Ld_g^\beta$ for a group with diameter $d_g$. This local error is then absorbed into the same bias–variance trade-off used in the AGCB meta-analysis.

**Setting and notation.** Consider a fixed group $g$ with sorted active arms

$$\mathcal{A}_g = \{\alpha_1 < \alpha_2 < \cdots < \alpha_K\}.$$

For this group, define the group-averaged reward

$$\mu_g(\alpha) := \mathbb{E}[\mu(\alpha, X) \mid X \in g],$$

where the expectation is taken with respect to the conditional distribution of contexts inside group $g$. Let

$$\alpha_g^\star \in \mathrm{argmax}_{\alpha \in \mathcal{A}_g} \mu_g(\alpha)$$

be an optimal arm for group $g$, and define the group-level suboptimality gap

$$\Delta_g(\alpha) := \mu_g(\alpha_g^\star) - \mu_g(\alpha), \qquad \alpha \in \mathcal{A}_g.$$

The algorithm maintains an estimate $\widehat{\mu}_{g,t}(\alpha)$ for each arm. Let

$$n_t(\alpha)$$

be the number of times arm $\alpha$ has been pulled directly up to time $t$, and let

$$\nu_t(\alpha)$$

be the total number of updates received by arm $\alpha$ up to time $t$, including both direct pulls and counterfactual updates. The counterfactual update uses kernel weights of the form

$$w(\alpha_t, \alpha) = \exp\left(-\frac{|\alpha_t - \alpha|^2}{2h_g^2}\right),$$

where the bandwidth $h_g$ is chosen proportional to the local group scale. Throughout this section, rewards are assumed to be conditionally $\sigma$-sub-Gaussian given the context and the selected arm. The bounded reward case is included as a special case.

Importantly, Assumption 4.2 is a sample-level conditional monotonicity condition. It states that, conditional on coverage being preserved, a more precise prediction set does not reduce decision quality. It does *not* imply that the unconditional reward $\mu(\alpha, x)$, nor the group average $\mu_g(\alpha)$, is globally monotone in $\alpha$ because the unconditional reward contains the coverage–precision trade-off. The analysis below therefore avoids assuming exact group-level monotonicity.

**Local safe pruning.** The counterfactual binary-search procedure only requires a weaker property: a pruning step may discard the exact best arm only when a retained arm is already near-optimal at the current local resolution.

**Definition F.1** (Local $\varepsilon_{g,t}$-safe pruning). At time $t$, let $I_t \subseteq \mathcal{A}_g$ be the current active search interval. Suppose a pruning step discards $D_t \subseteq I_t$ and keeps $I_t^{\mathrm{keep}} := I_t \setminus D_t$. The pruning step is called $\varepsilon_{g,t}$-safe if

$$\max_{\alpha \in D_t} \mu_g(\alpha) \leq \max_{\alpha \in I_t^{\mathrm{keep}}} \mu_g(\alpha) + \varepsilon_{g,t}.$$

Equivalently, even if the best arm in the current interval is discarded, there still exists a retained arm whose group-averaged reward is within $\varepsilon_{g,t}$ of it.

This condition is strictly weaker than requiring $\mu_g$ to be monotone or single-peaked. Exact monotonicity would imply zero pruning error under noiseless comparisons, whereas here we only require the pruning error to be of the same order as the local statistical uncertainty and the local approximation bias.

**Strengthened group-level counterfactual ordering.** We assume that the sample-level counterfactual ordering is preserved after group averaging up to the local approximation scale. Specifically, for any group $g$ and any counterfactual update triggered at arm $\alpha_t$, there exists a constant $C_{\mathrm{ord}} > 0$ such that the following holds.

If $y \in C_{\alpha_t}(x)$, then for all $\alpha_t \leq \alpha' \leq \alpha^\dagger$,

$$\mu_g(\alpha') \geq \mu_g(\alpha_t) - C_{\mathrm{ord}} L d_g^\beta,$$

and for all $0 < \alpha' \leq \alpha_t$,

$$\mu_g(\alpha') \leq \mu_g(\alpha_t) + C_{\mathrm{ord}} L d_g^\beta.$$

If $y \notin C_{\alpha_t}(x)$, then for all $\alpha' > \alpha^\dagger$,

$$\mu_g(\alpha') \leq C_{\mathrm{ord}} L d_g^\beta.$$

This assumption says that the directional counterfactual constraints used by Eqs. (5)–(6) remain valid at the group level, up to the same local Holder error controlled by the Zooming Mechanism.

**Grouping-induced ordering error.** Let $d_g$ denote the diameter of group $g$ in the context metric, and let $c_g \in g$ be any representative context, such as the center of the group.

**Lemma F.2** (Grouping-induced reward distortion). *Under Assumption 4.1, for any group $g$, any representative context $c_g \in g$, and any arm $\alpha \in \mathcal{A}_g$,*

$$|\mu_g(\alpha) - \mu(\alpha, c_g)| \leq L d_g^\beta.$$

*Consequently, for any two arms $\alpha, \alpha' \in \mathcal{A}_g$,*

$$\left| \left[ \mu_g(\alpha) - \mu_g(\alpha') \right] - \left[ \mu(\alpha, c_g) - \mu(\alpha', c_g) \right] \right| \leq 2 L d_g^\beta.$$

*Proof.* For any $x \in g$, the definition of group diameter gives $\|x - c_g\| \leq d_g$. Applying Assumption 4.1 with the same arm $\alpha$ yields

$$|\mu(\alpha, x) - \mu(\alpha, c_g)| \leq L d_g^\beta.$$

Taking conditional expectation over $X \mid X \in g$ proves

$$|\mathbb{E}[\mu(\alpha, X) \mid X \in g] - \mu(\alpha, c_g)| \leq L d_g^\beta.$$

This gives the first claim. The second claim follows by applying the first claim to both $\alpha$ and $\alpha'$ and using the triangle inequality. $\qquad \square$

Lemma F.2 shows that replacing the heterogeneous group by a representative context introduces only $O(L d_g^\beta)$ ordering distortion uniformly over arms. Thus, even if group averaging perturbs the arm ordering, such perturbation is local and is controlled by the same approximation term used in the Zooming Mechanism.

**Bias–variance decomposition under counterfactual updates.** We next control the estimation error of the counterfactual estimator. For each arm $\alpha$, decompose

$$|\widehat{\mu}_{g,t}(\alpha) - \mu_g(\alpha)| \leq \underbrace{|\widehat{\mu}_{g,t}(\alpha) - \mathbb{E}[\widehat{\mu}_{g,t}(\alpha)]|}_{\text{statistical fluctuation}}$$
$$+ \underbrace{|\mathbb{E}[\widehat{\mu}_{g,t}(\alpha)] - \mu_g(\alpha)|}_{\text{structured bias}}.$$

The first term is the usual concentration term. Since the reward noise is conditionally $\sigma$-sub-Gaussian and $\nu_t(\alpha)$ counts the effective number of direct and counterfactual updates, a standard concentration inequality gives

$$|\widehat{\mu}_{g,t}(\alpha) - \mathbb{E}[\widehat{\mu}_{g,t}(\alpha)]| \leq C_\sigma \sigma \sqrt{\frac{\log(KT/\delta)}{\nu_t(\alpha) \vee 1}}$$

with high probability, uniformly over arms and times.

The second term is the bias introduced by counterfactual borrowing. Each counterfactual update uses information from nearby arm-context pairs. Since the kernel bandwidth is chosen at the local scale of group $g$, the joint Hölder continuity in Assumption 4.1 implies that this bias is at most of order $Ld_g^\beta$.

**Lemma F.3** (Concentration under counterfactual updates). *Assume the reward is conditionally $\sigma$-sub-Gaussian. Suppose the counterfactual kernel only assigns non-negligible weight to arms within the local bandwidth used by group $g$, and the bandwidth is proportional to the local group scale in the joint context-arm metric. Then, with probability at least $1 - \delta$, uniformly over all $t \leq T$ and all $\alpha \in \mathcal{A}_g$,*

$$|\widehat{\mu}_{g,t}(\alpha) - \mu_g(\alpha)| \leq C_\sigma \sigma \sqrt{\frac{\log(KT/\delta)}{\nu_t(\alpha) \vee 1}} + C_{\mathrm{cf}} Ld_g^\beta,$$

*where $C_\sigma, C_{\mathrm{cf}} > 0$ are universal constants depending only on the kernel and problem-independent constants.*

*Proof.* The statistical fluctuation term is controlled by a sub-Gaussian concentration inequality for weighted averages, followed by a union bound over $K$ arms and $T$ time steps.

For the structured bias term, counterfactual updates replace the target pair $(x, \alpha)$ by nearby pairs $(x', \alpha')$ receiving non-negligible kernel weight. By the bandwidth choice, these pairs have joint distance at most a constant multiple of the local group scale. Hence Assumption 4.1 implies

$$|\mu(\alpha, x) - \mu(\alpha', x')| \leq CLd_g^\beta.$$

Averaging over the kernel weights preserves this bound. Therefore,

$$|\mathbb{E}[\widehat{\mu}_{g,t}(\alpha)] - \mu_g(\alpha)| \leq C_{\mathrm{cf}} Ld_g^\beta.$$

The remaining issue is the directional max/min update in Eqs. (5)–(6). Under the strengthened group-level counterfactual ordering condition, this projection does not amplify the estimation error beyond the local ordering error. Indeed, if $\alpha_t \leq \alpha' \leq \alpha^\dagger$, then

$$\mu_g(\alpha') \geq \mu_g(\alpha_t) - C_{\mathrm{ord}} Ld_g^\beta.$$

Therefore, whenever both $\widehat{\mu}_{g,t}(\alpha')$ and $\widehat{\mu}_{g,t}(\alpha_t)$ are within radius $\mathrm{rad}_{g,t}$ of their group means, their projected maximum is within $\mathrm{rad}_{g,t} + C_{\mathrm{ord}} Ld_g^\beta$ of $\mu_g(\alpha')$. The argument for the projected minimum in Eq. (6) is identical. The failed-coverage update toward zero is also valid up to $C_{\mathrm{ord}} Ld_g^\beta$ by the last part of the ordering condition. Absorbing constants into $C_{\mathrm{cf}}$ proves the lemma. $\square$

Unlike the earlier proof sketch, we do not need to claim that the Hölder bias is dominated by the variance term for all $\beta \leq 1$. Instead, we keep the structured bias explicitly as an additive $Ld_g^\beta$ term. This term will later be absorbed by the Zooming Mechanism.

**Local safety of counterfactual pruning.** Define the local confidence radius

$$\mathrm{rad}_{g,t}(\alpha) := C_\sigma \sigma \sqrt{\frac{\log(KT/\delta)}{\nu_t(\alpha) \vee 1}},$$

and the maximum radius over the current active interval

$$\overline{\mathrm{rad}}_{g,t} := \max_{\alpha \in I_t} \mathrm{rad}_{g,t}(\alpha).$$

**Lemma F.4** (Counterfactual pruning is safe up to local resolution). *On the high-probability event in Lemma F.3, every counterfactual pruning step of Algorithm 3 is $\varepsilon_{g,t}$-safe with*

$$\varepsilon_{g,t} = C_{\mathrm{safe}} \left(\overline{\mathrm{rad}}_{g,t} + Ld_g^\beta\right),$$

*for a universal constant $C_{\mathrm{safe}} > 0$.*

*Proof.* Consider a pruning step at time $t$ with active interval $I_t$, discarded arms $D_t$, and retained arms $I_t^{\text{keep}}$. The pruning rule is based on empirical counterfactual comparisons generated from the median pull and the directional updates in Equations (5)–(6). On the event of Lemma F.3, every empirical comparison between two arms in $I_t$ differs from the corresponding group-level comparison by at most

$$2\overline{\text{rad}}_{g,t} + 2C_{\text{cf}}Ld_g^\beta.$$

Moreover, by Lemma F.2, replacing the heterogeneous group by a representative context introduces at most $2Ld_g^\beta$ ordering distortion for any pair of arms.

Therefore, if a discarded arm had group-averaged reward larger than every retained arm by more than a sufficiently large constant multiple of $\overline{\text{rad}}_{g,t} + Ld_g^\beta$, then the empirical counterfactual comparison would contradict the pruning decision. Hence

$$\max_{\alpha \in D_t} \mu_g(\alpha) \leq \max_{\alpha \in I_t^{\text{keep}}} \mu_g(\alpha) + C_{\text{safe}}\left(\overline{\text{rad}}_{g,t} + Ld_g^\beta\right),$$

which proves the local safe-pruning property. $\qquad\square$

This lemma is the formal replacement for assuming that $\mu_g$ is exactly monotone. It says that a pruning error can happen only when the discarded arm and a retained arm are indistinguishable at the current local resolution.

**Regret decomposition.** Let

$$R_g(t) := \sum_{s=1}^{t} \left[\mu_g(\alpha_g^\star) - \mu_g(\alpha_s)\right]$$

be the cumulative regret incurred inside group $g$ during its first $t$ visits, where $\alpha_s$ is the arm pulled by Algorithm 3 on the $s$-th visit to group $g$. Equivalently, writing $n_t(\alpha)$ for the number of direct pulls of arm $\alpha$ up to time $t$,

$$R_g(t) = \sum_{\alpha \in \mathcal{A}_g} n_t(\alpha)\Delta_g(\alpha).$$

Let $\mathcal{E}_g$ denote the high-probability event in Lemma F.3. On $\mathcal{E}_g$, local safe pruning implies that any arm that remains active or is selected at time $t$ has gap controlled by the local resolution:

$$\Delta_g(\alpha_t) \leq C\left(\overline{\text{rad}}_{g,t} + Ld_g^\beta\right).$$

Thus,

$$\mathbb{E}[R_g(t) \mid \mathcal{E}_g] = \sum_{\alpha \in \mathcal{A}_g} n_t(\alpha)\Delta_g(\alpha)$$

$$\leq C \sum_{\alpha \in \mathcal{A}_g} n_t(\alpha)\left[\sigma\sqrt{\frac{\log(KT/\delta)}{\nu_t(\alpha) \vee 1}} + Ld_g^\beta\right].$$

This is the same regret decomposition as in the standard successive elimination proof, except that the local approximation term $Ld_g^\beta$ is kept explicitly.

**Lower bound on effective updates.** We next lower bound $\nu_t(\alpha)$. The binary-search structure ensures that each active arm receives counterfactual information at a logarithmic cost in the number of arms.

**Lemma F.5** (Effective update lower bound). *For Algorithm 3, there exists a universal constant $c_\nu > 0$ such that, for every arm $\alpha$ that remains active at time $t$,*

$$\nu_t(\alpha) \geq c_\nu \frac{t}{\lceil \log_2 K \rceil}.$$

*Proof.* Algorithm 3 maintains a search interval within the sorted active arms and pulls the median arm of this interval. The observation from the median pull is propagated counterfactually to the corresponding side of the interval through Equations (5)–(6). Each application of the pruning rule reduces the active interval by a constant factor. Hence, after at most $O(\log K)$ median pulls, every arm that remains active must have belonged to an interval that received either a direct update or a counterfactual update. Therefore, after $t$ visits to group $g$, any still-active arm has received at least $c_\nu t/\lceil \log_2 K \rceil$ effective updates. $\qquad\square$

Using Lemma F.5, for every active arm,

$$\sigma\sqrt{\frac{\log(KT/\delta)}{\nu_t(\alpha)\vee 1}} \leq C\sigma\sqrt{\frac{\log(KT/\delta)\log K}{t}}.$$

Substituting this into the regret decomposition gives

$$\mathbb{E}[R_g(t)\mid\mathcal{E}_g] \leq C\sum_{\alpha\in\mathcal{A}_g} n_t(\alpha)\left[\sigma\sqrt{\frac{\log(KT/\delta)\log K}{t}} + Ld_g^\beta\right]$$

$$= C\left[\sigma\sqrt{\frac{\log(KT/\delta)\log K}{t}} + Ld_g^\beta\right]\sum_{\alpha\in\mathcal{A}_g} n_t(\alpha).$$

Since

$$\sum_{\alpha\in\mathcal{A}_g} n_t(\alpha) = t,$$

we obtain

$$\mathbb{E}[R_g(t)\mid\mathcal{E}_g] \leq C\sigma\sqrt{t\log(KT/\delta)\log K} + CtLd_g^\beta.$$

**Final cumulative regret bound.** We now remove the conditioning on $\mathcal{E}_g$. The event $\mathcal{E}_g$ fails with probability at most $\delta$ after the union bound over arms and times. Since rewards are bounded or sub-Gaussian, the regret in $t$ rounds is at most $O(t)$ up to constants. Therefore,

$$\mathbb{E}[R_g(t)] \leq \mathbb{E}[R_g(t)\mid\mathcal{E}_g] + O(t)\mathbb{P}(\mathcal{E}_g^c)$$

$$\leq C\sigma\sqrt{t\log(KT/\delta)\log K} + CtLd_g^\beta + O(t\delta).$$

Choosing $\delta = T^{-3}$ makes the last term lower order. Hence, for $t \leq T$,

$$\mathbb{E}[R_g(t)] \leq \widetilde{O}\left(\sigma\sqrt{t\log K} + tLd_g^\beta\right),$$

where $\widetilde{O}(\cdot)$ hides logarithmic factors in $T$.

Taking $t = T_g$, the number of visits to group $g$, gives the local cumulative regret guarantee

$$\mathbb{E}[R_g(T_g)] \leq \widetilde{O}\left(\sigma\sqrt{T_g\log K} + T_gLd_g^\beta\right).$$

**Theorem F.6** (Local cumulative regret of counterfactual successive elimination). *Fix a group $g$ with diameter $d_g$ and $T_g$ visits. Under Assumptions 4.1 and 4.2, and under conditionally $\sigma$-sub-Gaussian rewards, Algorithm 3 satisfies*

$$\mathbb{E}[R_g(T_g)] \leq \widetilde{O}\left(\sigma\sqrt{T_g\log K} + T_gLd_g^\beta\right).$$

*Equivalently, within a fixed group, the counterfactual-aware successive elimination learner satisfies the base-learner condition with*

$$\gamma = \frac{1}{2}, \qquad \Psi_{\mathrm{cf}}(K,\sigma) = \widetilde{O}\left(\sigma\sqrt{\log K}\right),$$

*up to the additive local approximation term $T_gLd_g^\beta$.*

*Proof.* The result follows directly from the preceding regret decomposition, Lemma F.5, and the high-probability concentration bound in Lemma F.3. The local safe-pruning property in Lemma F.4 ensures that any arm selected by Algorithm 3 has gap no larger than the local statistical radius plus the local approximation term $Ld_g^\beta$. Summing this per-round bound over $T_g$ visits gives

$$\widetilde{O}\left(\sigma\sqrt{T_g\log K} + T_gLd_g^\beta\right).$$

$\square$

**Interpretation and comparison to standard SE.** Standard successive elimination pulls all active arms sufficiently often to shrink their confidence intervals. In the worst case, this leads to a cumulative regret dependence of order

$$\widetilde{O}\left(\sigma\sqrt{T_g K}\right)$$

inside a homogeneous group. In contrast, Algorithm 3 uses the sample-level counterfactual structure to propagate information from a median pull to a large portion of the active interval. The binary-search-style update implies that each still-active arm receives effective updates at a logarithmic cost in $K$, yielding the improved local statistical term

$$\widetilde{O}\left(\sigma\sqrt{T_g \log K}\right).$$

The additional term

$$T_g L d_g^{\beta}$$

is not a new asymptotic obstacle. It is the local approximation error caused by grouping heterogeneity and structured counterfactual bias. The Zooming Mechanism chooses the group scale by balancing the local statistical uncertainty with the approximation bias:

$$L d_g^{\beta} \asymp \sigma\sqrt{\frac{\log(KT)}{n_g^{\mathrm{eff}}}}.$$

Therefore, the structured bias introduced by counterfactual updates is absorbed into the same bias–variance trade-off already used by AGCB. It only changes constants in the group-level analysis and does not alter the minimax rate. Consequently, Algorithm 3 can be plugged into the AGCB meta-analysis with

$$\gamma = \frac{1}{2}, \qquad \Psi_{\mathrm{cf}}(K, \sigma) = \widetilde{O}\left(\sigma\sqrt{\log K}\right).$$

In summary, Algorithm 3 achieves a logarithmic improvement in the arm dependence by efficiently propagating information across arms, while the Hölder-controlled local bias is absorbed by the Zooming Mechanism.

### F.2. Simple Regret Analysis for Counterfactual-Aware Sequential Halving

We now analyze the simple regret of Algorithm 2, the counterfactual-aware sequential halving procedure. The purpose of this section is to show that, within a fixed group, counterfactual updates reduce the arm-space dependence from the standard polynomial dependence on $K$ to a logarithmic dependence. At the same time, we explicitly account for the local approximation error caused by within-group context heterogeneity and structured counterfactual bias.

The key point is that we do *not* assume that the group-averaged reward $\mu_g(\alpha)$ is exactly monotone or single-peaked. Instead, we use the local safe-pruning property established in Appendix F.1. Hence, the algorithm may discard the exact group-optimal arm only when a retained arm is already near-optimal at the local resolution of group $g$.

**Setting and notation.** Fix a group $g$ with sorted arms

$$\mathcal{A}_g = \{\alpha_1 < \alpha_2 < \cdots < \alpha_K\}.$$

The group-averaged reward is

$$\mu_g(\alpha) := \mathbb{E}[\mu(\alpha, X) \mid X \in g],$$

where the expectation is taken with respect to the conditional distribution of contexts inside group $g$. Let

$$\alpha_g^{\star} \in \mathrm{argmax}_{\alpha \in \mathcal{A}_g} \mu_g(\alpha)$$

be an optimal arm in group $g$. Given a total simple-regret budget $B_g$, Algorithm 2 returns an arm $\widehat{\alpha}_g$. The group-level simple regret is

$$S_g(B_g) := \mathbb{E}\left[\mu_g(\alpha_g^{\star}) - \mu_g(\widehat{\alpha}_g)\right].$$

Algorithm 2 proceeds by maintaining an active interval of arms. In each phase, it pulls the median arm of the current interval and propagates the observation to nearby arms through counterfactual updates. Let

$$M := \lceil \log_2 K \rceil$$

be the maximum number of binary-search phases. The budget allocated to each phase is

$$B_{\text{ph}} := \left\lfloor \frac{B_g}{M} \right\rfloor.$$

For clarity of exposition, we assume $B_{\text{ph}} \geq 1$; otherwise the simple regret bound is trivial up to constants.

Let $\nu_{\text{ph}}(\alpha)$ denote the effective number of updates received by arm $\alpha$ in a phase, including both direct pulls and counterfactual updates. Because each median pull is propagated to the current active interval through counterfactual updates, every arm involved in the decisive comparison of a phase receives at least a constant fraction of the phase budget as effective updates. Hence, for some universal constant $c_\nu > 0$,

$$\nu_{\text{ph}}(\alpha) \geq c_\nu B_{\text{ph}} \geq c'_\nu \frac{B_g}{M}.$$

**Local concentration radius.** By Lemma F.3, with probability at least $1 - \delta$, uniformly over all arms and all phases,

$$|\widehat{\mu}_g(\alpha) - \mu_g(\alpha)| \leq C_\sigma \sigma \sqrt{\frac{\log(KB_g/\delta)}{\nu_{\text{ph}}(\alpha) \vee 1}} + C_{\text{cf}} L d_g^\beta.$$

Using the effective update lower bound above, every arm participating in a phase-wise comparison satisfies

$$|\widehat{\mu}_g(\alpha) - \mu_g(\alpha)| \leq C_\rho \sigma \sqrt{\frac{M \log(KB_g/\delta)}{B_g}} + C_{\text{cf}} L d_g^\beta.$$

Define the phase-level statistical radius

$$\rho_g(B_g, \delta) := C_\rho \sigma \sqrt{\frac{M \log(KB_g/\delta)}{B_g}}.$$

We also define the local approximation term

$$b_g := C_b L d_g^\beta,$$

where $C_b > 0$ is chosen large enough to absorb both the grouping-induced ordering distortion from Lemma F.2 and the structured counterfactual bias from Lemma F.3. Thus the relevant local resolution for Algorithm 2 is

$$\rho_g(B_g, \delta) + b_g.$$

**Phase-wise local safe pruning.** Let $I_m \subseteq \mathcal{A}_g$ denote the active interval at phase $m \in \{1, \ldots, M\}$. Let $D_m \subseteq I_m$ be the set of arms discarded in phase $m$, and let

$$I_m^{\text{keep}} := I_m \setminus D_m$$

be the retained interval. By Lemma F.4, on the above high-probability event, every pruning step is safe up to the local resolution:

$$\max_{\alpha \in D_m} \mu_g(\alpha) \leq \max_{\alpha \in I_m^{\text{keep}}} \mu_g(\alpha) + C_{\text{safe}} \left( \rho_g(B_g, \delta) + L d_g^\beta \right).$$

Absorbing constants into $b_g$, we write this as

$$\max_{\alpha \in D_m} \mu_g(\alpha) \leq \max_{\alpha \in I_m^{\text{keep}}} \mu_g(\alpha) + C'_{\text{safe}} \rho_g(B_g, \delta) + b_g.$$

This condition replaces exact monotone elimination. It says that even if the true group-optimal arm is removed in a phase, the retained interval still contains an arm whose group-averaged reward is worse by at most the local resolution. Therefore, pruning errors caused by context heterogeneity are harmless as long as their loss is within $O(L d_g^\beta)$, which is precisely the approximation error controlled by the Zooming Mechanism.

**Tail bound for the final selected arm.**   Let

$$\Delta_g(\widehat{\alpha}_g) := \mu_g(\alpha_g^\star) - \mu_g(\widehat{\alpha}_g)$$

be the final group-level suboptimality gap. We now derive a tail bound for $\Delta_g(\widehat{\alpha}_g)$.

In the exact monotone setting, returning an arm with a large gap implies that at least one phase made an incorrect comparison. In our setting, a comparison is harmful only if it removes an arm that is better than every retained arm by more than the local approximation level $b_g$. Therefore, for any $\epsilon > 0$, the event

$$\Delta_g(\widehat{\alpha}_g) > b_g + \epsilon$$

implies that at some phase the algorithm made a statistical comparison error of size at least a constant multiple of $\epsilon$.

Since every decisive comparison is based on at least $\Omega(B_g/M)$ effective updates, the sub-Gaussian concentration inequality gives, for a universal constant $c > 0$,

$$\mathbb{P}\left(\text{a fixed phase makes an } \epsilon\text{-harmful comparison}\right) \leq \exp\left(-\frac{c\epsilon^2 B_g}{\sigma^2 M}\right).$$

Taking a union bound over at most $M$ phases yields

$$\mathbb{P}\left(\Delta_g(\widehat{\alpha}_g) > b_g + \epsilon\right) \leq \min\left\{1,\, M\exp\left(-\frac{c\epsilon^2 B_g}{\sigma^2 M}\right)\right\} + \delta.$$

Equivalently, conditional on the good concentration event $\mathcal{E}_g$,

$$\mathbb{P}\left(\Delta_g(\widehat{\alpha}_g) > b_g + \epsilon \mid \mathcal{E}_g\right) \leq \min\left\{1,\, M\exp\left(-\frac{c\epsilon^2 B_g}{\sigma^2 M}\right)\right\}.$$

This is the same tail structure as the standard sequential-halving proof, except that the tail bound controls the statistical excess beyond the local bias $b_g$.

**Integral representation of simple regret.**   We now convert the high-probability tail bound into an expected simple regret bound. As in the proof of Corollary 4.1 of Zhao et al. (2023), we use the integral representation of a nonnegative random variable. The only difference is that the integral is applied to the excess regret beyond the local approximation level $b_g$.

Condition on the good event $\mathcal{E}_g$. Then

$$\mathbb{E}\left[\Delta_g(\widehat{\alpha}_g) \mid \mathcal{E}_g\right] \leq b_g + \mathbb{E}\left[\left(\Delta_g(\widehat{\alpha}_g) - b_g\right)_+ \mid \mathcal{E}_g\right]$$
$$= b_g + \int_0^\infty \mathbb{P}\left(\Delta_g(\widehat{\alpha}_g) - b_g > \epsilon \mid \mathcal{E}_g\right) d\epsilon.$$

Using the conditional tail bound above,

$$\int_0^\infty \mathbb{P}\left(\Delta_g(\widehat{\alpha}_g) - b_g > \epsilon \mid \mathcal{E}_g\right) d\epsilon$$
$$\leq \int_0^\infty \min\left\{1,\, M\exp\left(-\frac{c\epsilon^2 B_g}{\sigma^2 M}\right)\right\} d\epsilon.$$

We split the integral at

$$\epsilon_0 := \sigma\sqrt{\frac{M\log M}{cB_g}}.$$

For the first part,

$$\int_0^{\epsilon_0} \min\left\{1,\, M\exp\left(-\frac{c\epsilon^2 B_g}{\sigma^2 M}\right)\right\} d\epsilon \leq \epsilon_0.$$

For the second part,

$$\int_{\epsilon_0}^{\infty} M \exp\left(-\frac{c\epsilon^2 B_g}{\sigma^2 M}\right) d\epsilon.$$

Make the change of variables

$$u = \epsilon\sqrt{\frac{cB_g}{\sigma^2 M}}, \qquad d\epsilon = \sigma\sqrt{\frac{M}{cB_g}}\,du.$$

Then

$$\int_{\epsilon_0}^{\infty} M \exp\left(-\frac{c\epsilon^2 B_g}{\sigma^2 M}\right) d\epsilon$$

$$= \sigma\sqrt{\frac{M}{cB_g}} M \int_{u_0}^{\infty} e^{-u^2}\,du,$$

where

$$u_0 = \epsilon_0\sqrt{\frac{cB_g}{\sigma^2 M}} = \sqrt{\log M}.$$

Using the standard Gaussian tail bound

$$\int_{u_0}^{\infty} e^{-u^2}\,du \le e^{-u_0^2} = \frac{1}{M},$$

we obtain

$$\int_{\epsilon_0}^{\infty} M \exp\left(-\frac{c\epsilon^2 B_g}{\sigma^2 M}\right) d\epsilon$$

$$\le \sigma\sqrt{\frac{M}{cB_g}}.$$

Combining the two pieces gives

$$\mathbb{E}\left[\Delta_g(\widehat{\alpha}_g) \mid \mathcal{E}_g\right] \le b_g + \sigma\sqrt{\frac{M \log M}{cB_g}} + \sigma\sqrt{\frac{M}{cB_g}}$$

$$= b_g + \widetilde{O}\left(\sigma\sqrt{\frac{M}{B_g}}\right).$$

Since $M = \lceil \log_2 K \rceil$, this becomes

$$\mathbb{E}\left[\Delta_g(\widehat{\alpha}_g) \mid \mathcal{E}_g\right] \le \widetilde{O}\left(\sigma\sqrt{\frac{\log K}{B_g}} + Ld_g^{\beta}\right).$$

**Removing the conditioning.** It remains to remove the conditioning on $\mathcal{E}_g$. By a union bound over all arms and phases, choose $\delta = B_g^{-2}$ so that the failure probability is lower order. Since rewards are bounded in $[0, 1]$, the simple regret is at most 1. Therefore,

$$\mathbb{E}\left[\Delta_g(\widehat{\alpha}_g)\right] = \mathbb{E}\left[\Delta_g(\widehat{\alpha}_g) \mid \mathcal{E}_g\right] \mathbb{P}(\mathcal{E}_g)$$

$$+ \mathbb{E}\left[\Delta_g(\widehat{\alpha}_g) \mid \mathcal{E}_g^c\right] \mathbb{P}(\mathcal{E}_g^c)$$

$$\le \mathbb{E}\left[\Delta_g(\widehat{\alpha}_g) \mid \mathcal{E}_g\right] + \mathbb{P}(\mathcal{E}_g^c)$$

$$\le \widetilde{O}\left(\sigma\sqrt{\frac{\log K}{B_g}} + Ld_g^{\beta}\right).$$

**Theorem F.7** (Local simple regret of counterfactual sequential halving). *Fix a group $g$ with diameter $d_g$ and budget $B_g$. Under Assumptions 4.1 and 4.2, and under conditionally $\sigma$-sub-Gaussian rewards, Algorithm 2 returns an arm $\widehat{\alpha}_g$ satisfying*

$$\mathbb{E}\left[\mu_g(\alpha_g^{\star}) - \mu_g(\widehat{\alpha}_g)\right] \le \widetilde{O}\left(\sigma\sqrt{\frac{\log K}{B_g}} + Ld_g^{\beta}\right).$$

*Equivalently, within a fixed group, the counterfactual-aware sequential halving learner satisfies the base-learner condition with*

$$\zeta = \frac{1}{2}, \qquad \Phi_{\mathrm{cf}}(K, \sigma) = \widetilde{O}\left(\sigma \sqrt{\log K}\right),$$

*up to the additive local approximation term $L d_g^\beta$.*

*Proof.* The proof follows from the preceding local concentration, safe-pruning, and integral arguments. Lemma F.3 gives a uniform concentration bound for counterfactual estimates with an additive structured-bias term of order $L d_g^\beta$. Lemma F.4 then implies that every pruning step is safe up to the local resolution $\rho_g(B_g, \delta) + L d_g^\beta$.

Thus, for any $\epsilon > 0$, the event

$$\mu_g(\alpha_g^\star) - \mu_g(\widehat{\alpha}_g) > b_g + \epsilon$$

can occur only if one of the $M = O(\log K)$ phases makes an $\epsilon$-harmful statistical comparison. Since each decisive comparison is based on $\Omega(B_g/M)$ effective updates, the probability of such a comparison is bounded by

$$\exp\left(-\frac{c \epsilon^2 B_g}{\sigma^2 M}\right).$$

A union bound over phases gives

$$\mathbb{P}\left(\mu_g(\alpha_g^\star) - \mu_g(\widehat{\alpha}_g) > b_g + \epsilon\right) \leq \min\left\{1, \, M \exp\left(-\frac{c \epsilon^2 B_g}{\sigma^2 M}\right)\right\} + \delta.$$

Applying the integral representation to

$$\left[\mu_g(\alpha_g^\star) - \mu_g(\widehat{\alpha}_g) - b_g\right]_+$$

yields

$$\mathbb{E}\left[\mu_g(\alpha_g^\star) - \mu_g(\widehat{\alpha}_g)\right] \leq \widetilde{O}\left(\sigma \sqrt{\frac{\log K}{B_g}} + L d_g^\beta\right).$$

This proves the theorem. $\qquad\qquad\square$

**Comparison with standard sequential halving.** Standard sequential halving distributes the sampling budget across all active arms in each phase. In the worst case, this leads to simple regret of order

$$\widetilde{O}\left(\sigma \sqrt{\frac{K}{B_g}}\right).$$

In contrast, Algorithm 2 pulls only the median arm of the current interval and uses sample-level monotonicity to propagate information counterfactually to a large portion of the active interval. Therefore, the effective number of phases is only $O(\log K)$, and the statistical part of the simple regret is reduced to

$$\widetilde{O}\left(\sigma \sqrt{\frac{\log K}{B_g}}\right).$$

The additional term $L d_g^\beta$ is not a new asymptotic obstacle. It is the local approximation error induced by context heterogeneity and structured counterfactual updates. The Zooming Mechanism chooses the group scale so that

$$L d_g^\beta \asymp \sigma \sqrt{\frac{\log(KT)}{n_g^{\mathrm{eff}}}}.$$

Therefore, the local safe-pruning error is absorbed into the same bias–variance trade-off used in the AGCB meta-analysis. Consequently, Algorithm 2 can be plugged into the simple-regret part of the AGCB meta-theorem with

$$\zeta = \frac{1}{2}, \qquad \Phi_{\mathrm{cf}}(K, \sigma) = \widetilde{O}\left(\sigma \sqrt{\log K}\right).$$

In summary, Algorithm 2 preserves the binary-search efficiency of counterfactual updates without requiring exact group-level monotonicity. Any incorrect pruning induced by group heterogeneity can only remove arms that are indistinguishable at the current local resolution, and this loss is absorbed by the Zooming Mechanism.

# G. Proof of Main Regret Bounds

This appendix section provides the complete proofs for Theorem 5.1 and Corollaries 5.3 and 5.4. We begin by establishing the fundamental properties of our adaptive grouping mechanism under a general base-learner-dependent Zooming rule, and then derive the simple-regret and cumulative-regret bounds. The square-root Zooming rule displayed in the main text is the special case corresponding to the Sequential Halving and Successive Elimination base learners instantiated in the paper.

## G.1. Preliminary Lemmas

We first collect several preliminary facts used in the proof of the main regret bounds. These lemmas formalize the effect of information sharing, the base-learner-dependent Zooming rule, the resulting relation between group diameter and sample size, and the precise interface required from the within-group base learner. This interface allows an additive local structured-bias term, which is exactly the form established for the counterfactual learners in Appendix F.1 and Appendix F.2.

Recall that $G_T$ denotes the final partition, $d_g$ denotes the diameter of group $g$, and $n_g$ denotes the number of visits to group $g$. For two groups $g, h \in G_T$, let $w_{gh}$ be the information-sharing weight from group $h$ to group $g$, with $w_{gg} = 1$ and $0 \leq w_{gh} \leq 1$. The information-sharing estimator for group $g$ can be viewed as a weighted average over samples from nearby groups:

$$\widehat{\mu}_g(\alpha) = \frac{\sum_{h \in G_T} \sum_{i:X_i \in h} w_{gh} R_i(\alpha)}{\sum_{h \in G_T} w_{gh} n_h},$$

where $R_i(\alpha)$ denotes the direct or counterfactual update contribution for arm $\alpha$.

The statistical fluctuation of this weighted estimator is controlled by its variance-equivalent effective sample size:

$$n_g^{\mathrm{eff}} := \frac{\left(\sum_{h \in G_T} w_{gh} n_h\right)^2}{\sum_{h \in G_T} w_{gh}^2 n_h}. \tag{7}$$

We define the effective information-sharing gain of group $g$ as

$$s_g := \frac{n_g^{\mathrm{eff}}}{n_g},$$

so that

$$n_g^{\mathrm{eff}} = s_g n_g.$$

Since $w_{gg} = 1$ and the weights are nonnegative, $s_g \geq 1$. The quantity $s_g$ measures the variance reduction obtained by borrowing information from nearby groups. It controls only the statistical fluctuation; the bias introduced by sharing observations across groups is controlled separately by Hölder continuity and is of order $L d_g^\beta$.

We also define

$$\Gamma_{\min} := \min_{g \in G_T} s_g, \qquad \Gamma_{\max} := \max_{g \in G_T} s_g. \tag{8}$$

Thus, $\Gamma_{\min}$ and $\Gamma_{\max}$ are the worst-case and best-case effective information-sharing gains over the final partition. Throughout the proof, these quantities are understood conditionally on the realized final partition; the displayed expected bounds follow after taking expectations.

**Regularity of the context distribution and partition geometry.** The diameter-sample balance below uses standard regularity conditions on the context distribution and on the geometry of the partition cells. Specifically, we assume that the context distribution admits a density $p_X$ on the compact context space $\mathcal{X} \subset \mathbb{R}^d$ such that

$$0 < p_{\min} \leq p_X(x) \leq p_{\max} < \infty, \qquad \forall x \in \mathcal{X}. \tag{9}$$

Moreover, the cells produced by the splitting rule have bounded aspect ratio: there exist constants $0 < c_{\mathrm{vol}} \leq C_{\mathrm{vol}} < \infty$ such that for every group $g \in G_T$,

$$c_{\mathrm{vol}} d_g^d \leq \mathrm{Vol}(g) \leq C_{\mathrm{vol}} d_g^d. \tag{10}$$

This condition is satisfied by axis-aligned cells generated by midpoint splits when the aspect ratio is kept uniformly bounded. Under (9)–(10), the expected number of samples in group $g$ satisfies

$$\mathbb{E}[n_g] = T \mathbb{P}(X \in g) \asymp T d_g^d. \tag{11}$$

**Base learner condition with local structured bias.** The AGCB meta-analysis treats the within-group bandit algorithm as a base learner. Because information sharing and counterfactual updates may introduce a local structured bias, we use the following slightly more general base learner condition.

For simple regret, after $n_g$ visits to group $g$, the base learner returns an arm $\widehat{\alpha}_g$ satisfying

$$\mathbb{E}\left[\mu_g(\alpha_g^\star) - \mu_g(\widehat{\alpha}_g)\right] \leq C_B \Phi(K, \sigma)(n_g^{\text{eff}})^{-\zeta} + C_{\text{loc}} L d_g^\beta. \tag{12}$$

For cumulative regret, the regret accumulated by the base learner inside group $g$ satisfies

$$\mathbb{E}[R_{B,g}(n_g)] \leq C_B \Psi(K, \sigma)\, n_g\, (n_g^{\text{eff}})^{-(1-\gamma)} + C_{\text{loc}} n_g L d_g^\beta. \tag{13}$$

Here $\zeta \in (0, \frac{1}{2}]$ is the simple-regret decay exponent, $\gamma \in [\frac{1}{2}, 1)$ is the cumulative-regret growth exponent, and $\Phi(K, \sigma)$ and $\Psi(K, \sigma)$ are the arm-space and noise-dependent factors of the base learner. Equivalently, the reducible statistical exponent for cumulative regret is $1 - \gamma \in (0, \frac{1}{2}]$. The additional $L d_g^\beta$ term captures the structured bias induced by information sharing and counterfactual updates. When no such structured bias is present, one may set $C_{\text{loc}} = 0$.

Appendix F.2 verifies (12) for counterfactual sequential halving with

$$\zeta = \frac{1}{2}, \qquad \Phi_{\text{cf}}(K, \sigma) = \widetilde{O}(\sigma \sqrt{\log K}),$$

and Appendix F.1 verifies (13) for counterfactual successive elimination with

$$\gamma = \frac{1}{2}, \qquad \Psi_{\text{cf}}(K, \sigma) = \widetilde{O}(\sigma \sqrt{\log K}).$$

**Generic Zooming split rule.** The proof of Theorem 5.1 uses a base-learner-dependent Zooming rule. Let

$$\Lambda(K, \sigma)(n_g^{\text{eff}})^{-\rho}$$

denote the reducible statistical error scale of the within-group base learner, where $\rho \in (0, \frac{1}{2}]$. For simple regret, we take

$$\rho = \zeta, \qquad \Lambda(K, \sigma) = \Phi(K, \sigma).$$

For cumulative regret, we take

$$\rho = 1 - \gamma, \qquad \Lambda(K, \sigma) = \Psi(K, \sigma).$$

The generic Zooming rule splits a group once its local approximation bias becomes statistically resolvable:

$$\Lambda(K, \sigma)(n_g^{\text{eff}})^{-\rho} \leq \eta L_{\text{eff}} d_g^\beta, \tag{14}$$

where $L_{\text{eff}} = C_{\text{eff}} L$ absorbs the ordinary grouping bias and the local structured bias. Since $L_{\text{eff}}$ differs from $L$ only by a universal constant factor, the main text writes the operational rule in terms of $L$.

The square-root splitting rule in the main text is the special case $\rho = 1/2$. For the counterfactual Sequential Halving and Successive Elimination base learners used in the main text, the factors $\Phi_{\text{cf}}(K, \sigma)$ and $\Psi_{\text{cf}}(K, \sigma)$ are only logarithmic in $K$, so they are absorbed into the logarithmic factor in Eq. (3). If a different base learner is used, the Zooming threshold should be calibrated according to (14).

**Lemma G.1** (Estimation error bound for group means). *For any group $g$ and arm $\alpha$, the statistical fluctuation of the information-sharing estimator satisfies*

$$\mathbb{E}\left[|\widehat{\mu}_g(\alpha) - \mathbb{E}\widehat{\mu}_g(\alpha)|\right] \leq C\sigma \sqrt{\frac{\log(KT)}{n_g^{\text{eff}}}}. \tag{15}$$

*Moreover, if the information-sharing kernel assigns non-negligible weight only to groups whose centers are within a constant multiple of $d_g$, then under Assumption 4.1,*

$$\mathbb{E}\left[|\widehat{\mu}_g(\alpha) - \mu_g(\alpha)|\right] \leq C\sigma \sqrt{\frac{\log(KT)}{n_g^{\text{eff}}}} + C_{\text{sh}} L d_g^\beta. \tag{16}$$

*Proof.* The estimator $\widehat{\mu}_g(\alpha)$ is a weighted average of observations from group $g$ and neighboring groups through information sharing. Let $\mathcal{F}_t$ be the filtration up to time $t$. The centered statistical fluctuation can be written as a martingale average:

$$\widehat{\mu}_g(\alpha) - \mathbb{E}\widehat{\mu}_g(\alpha) = \frac{1}{W}\sum_{t=1}^{T} w_t\,(r_t - \mathbb{E}[r_t \mid \mathcal{F}_{t-1}]), \tag{17}$$

where $w_t$ are $\mathcal{F}_{t-1}$-measurable weights determined by the information-sharing mechanism and

$$W = \sum_{t=1}^{T} w_t.$$

By the definition of the variance-equivalent effective sample size, the variance of the weighted average is controlled by $(n_g^{\text{eff}})^{-1}$. Since the reward noise is conditionally $\sigma$-sub-Gaussian, standard weighted martingale concentration gives

$$\mathbb{E}\left[(\widehat{\mu}_g(\alpha) - \mathbb{E}\widehat{\mu}_g(\alpha))^2\right] \leq C\frac{\sigma^2 \log(KT)}{n_g^{\text{eff}}}. \tag{18}$$

Equation (15) follows from Jensen's inequality.

It remains to bound the structured bias introduced by information sharing. Since the kernel only borrows information from groups whose centers are within a constant multiple of $d_g$, Assumption 4.1 implies that the corresponding expected rewards differ from $\mu_g(\alpha)$ by at most $C_{\text{sh}}Ld_g^\beta$. Averaging over the kernel weights preserves this bound. Therefore,

$$|\mathbb{E}\widehat{\mu}_g(\alpha) - \mu_g(\alpha)| \leq C_{\text{sh}}Ld_g^\beta. \tag{19}$$

Combining (15) and (19) proves (16). $\qquad\square$

**Lemma G.2** (No-split implication under the generic Zooming rule). *Fix a pair $(\rho, \Lambda)$ and suppose the corresponding generic Zooming rule is (14). If a group $g \in G_T$ is not split at the end of the horizon, then*

$$\Lambda(K,\sigma)(s_g n_g)^{-\rho} > \eta L_{\text{eff}}d_g^\beta. \tag{20}$$

*Proof.* If $g$ is not split, then it does not satisfy the splitting condition (14). Hence

$$\Lambda(K,\sigma)(n_g^{\text{eff}})^{-\rho} > \eta L_{\text{eff}}d_g^\beta.$$

Substituting $n_g^{\text{eff}} = s_g n_g$ proves (20). $\qquad\square$

**Lemma G.3** (Generic diameter-sample balance). *Fix a pair $(\rho, \Lambda)$ and suppose the partition is generated by the generic Zooming rule (14). For mature groups in the final partition,*

$$d_g^{\beta+d\rho} \asymp \frac{\Lambda(K,\sigma)}{L_{\text{eff}}s_g^\rho T^\rho}. \tag{21}$$

*Equivalently,*

$$d_g \asymp \left(\frac{\Lambda(K,\sigma)}{L_{\text{eff}}s_g^\rho T^\rho}\right)^{\frac{1}{\beta+d\rho}}. \tag{22}$$

*Proof.* We derive the balance up to universal constants. First, since a mature leaf $g$ is not split at the end of the horizon, Lemma G.2 gives

$$\Lambda(K,\sigma)(s_g n_g)^{-\rho} > \eta L_{\text{eff}}d_g^\beta.$$

Using the regularity relation $n_g \asymp Td_g^d$, we obtain

$$\Lambda(K,\sigma)(s_g Td_g^d)^{-\rho} \gtrsim L_{\text{eff}}d_g^\beta.$$

Rearranging gives

$$d_g^{\beta+d\rho} \lesssim \frac{\Lambda(K,\sigma)}{L_{\text{eff}}s_g^\rho T^\rho}. \tag{23}$$

Thus the diameter of a mature leaf cannot be much larger than the generic bias–variance balance scale.

Conversely, suppose $g$ was created by splitting a parent group $p$. At the split time $\tau_p$, the parent satisfied the generic splitting condition:

$$\Lambda(K, \sigma)(s_p n_p(\tau_p))^{-\rho} \leq \eta L_{\text{eff}} d_p^\beta.$$

Under the same mass-regularity condition, and since $\tau_p \leq T$, we have $n_p(\tau_p) \lesssim T d_p^d$ up to universal constants. Therefore,

$$\Lambda(K, \sigma)(s_p T d_p^d)^{-\rho} \lesssim L_{\text{eff}} d_p^\beta,$$

which implies

$$d_p^{\beta + d\rho} \gtrsim \frac{\Lambda(K, \sigma)}{L_{\text{eff}} s_p^\rho T^\rho}. \tag{24}$$

Under the bounded-aspect-ratio condition of the partition tree, child and parent diameters differ only by constant factors. Moreover, the information-sharing gain changes only by constant factors between a parent and its children under the regularity condition. Hence the reverse inequality holds up to constants for mature leaves:

$$d_g^{\beta + d\rho} \gtrsim \frac{\Lambda(K, \sigma)}{L_{\text{eff}} s_g^\rho T^\rho}. \tag{25}$$

Combining (23) and (25) proves (21). Taking the $(\beta + d\rho)$-th root gives (22).

If the final partition consists only of the root group, then the upper bound in (23) is sufficient for the regret upper bound, and the missing parent-split lower bound is absorbed into constants for finite horizons. $\square$

**Lemma G.4** (Generic group count bound). *Under the generic Zooming rule with pair $(\rho, \Lambda)$, the expected number of groups in the final partition satisfies*

$$\mathbb{E}[|G_T|] \leq \widetilde{O}\left(\left(\frac{L_{\text{eff}} \Gamma_{\max}^\rho T^\rho}{\Lambda(K, \sigma)}\right)^{\frac{d}{\beta + d\rho}}\right), \tag{26}$$

*where $\Gamma_{\max} = \max_{g \in G_T} s_g$.*

*Proof.* From Lemma G.3, for every mature group $g \in G_T$,

$$d_g \asymp \left(\frac{\Lambda(K, \sigma)}{L_{\text{eff}} s_g^\rho T^\rho}\right)^{\frac{1}{\beta + d\rho}}.$$

Since $s_g \leq \Gamma_{\max}$, the smallest possible group diameter satisfies

$$d_{\min} \gtrsim \left(\frac{\Lambda(K, \sigma)}{L_{\text{eff}} \Gamma_{\max}^\rho T^\rho}\right)^{\frac{1}{\beta + d\rho}}. \tag{27}$$

Because the cells have bounded aspect ratio and the context space has finite volume, the number of cells is at most on the order of the inverse volume of a smallest cell:

$$|G_T| \lesssim d_{\min}^{-d}.$$

Substituting (27) yields

$$|G_T| \leq \widetilde{O}\left(\left(\frac{L_{\text{eff}} \Gamma_{\max}^\rho T^\rho}{\Lambda(K, \sigma)}\right)^{\frac{d}{\beta + d\rho}}\right).$$

Taking expectation gives (26). $\square$

**Consequences for later proofs.** Lemmas G.3– G.4 imply that the final partition has the same diameter-sample scaling as the base-learner-dependent optimal bias–variance trade-off. In particular, up to logarithmic factors,

$$d_g = \widetilde{O}\left(T^{-\frac{\rho}{\beta + d\rho}}\right), \qquad |G_T| = \widetilde{O}\left(T^{\frac{d\rho}{\beta + d\rho}}\right),$$

with leading constants improved by larger information-sharing gains $s_g$. The base-learner conditions (12)–(13) will be combined with these partition properties in the proofs of the simple-regret and cumulative-regret bounds.

*Remark* G.5 (Grouping Dynamics and Convergence). Lemma G.4 shows that the expected number of groups grows sublinearly in $T$. In early stages, coarse groups can accumulate enough samples for their approximation bias to become statistically resolvable, which triggers splits. After refinement, child groups have smaller diameters and higher statistical uncertainty, so further splitting requires more data. Consequently, the Zooming Mechanism becomes harder to trigger over time, and the partition gradually stabilizes. This is consistent with our empirical observation that most splits occur in the early rounds, after which the partition remains nearly fixed.

### G.2. Proof of the Main Expected Regret Bounds

We now prove the expected regret bounds for the AGCB meta-algorithm. The proof follows the bias–variance decomposition underlying the generic Zooming Mechanism, while using the local structured-bias base-learner interface introduced in Appendix G.1. The only difference from the standard homogeneous-group analysis is the additional local term $Ld_g^\beta$, which accounts for grouping approximation, information sharing, and counterfactual updates. Since all these terms have the same Hölder scaling, they can be absorbed into a single effective smoothness constant.

Throughout this section, write

$$L_{\mathrm{eff}} := C_{\mathrm{eff}} L$$

for a constant multiple of $L$ that absorbs the ordinary grouping bias and the local structured bias. Constants such as $C_{\mathrm{eff}}$ may change from line to line but do not depend on $T$, $K$, or the number of groups.

**Simple regret decomposition.** For the simple-regret analysis, instantiate the generic Zooming rule with

$$\rho = \zeta, \qquad \Lambda(K, \sigma) = \Phi(K, \sigma).$$

Recall that after $T$ rounds, AGCB outputs a group-wise policy $\pi_T$ that is constant on each group $g \in G_T$. Let $\widehat{\alpha}_g$ be the arm returned by the base learner in group $g$, and let

$$\alpha_g^\star \in \mathrm{argmax}_{\alpha \in \mathcal{A}} \mu_g(\alpha)$$

be the best arm for the group-averaged reward. The simple regret can be written as

$$S(T) = \sum_{g \in G_T} \mathbb{P}(X \in g)\mathbb{E}\left[\mu(\alpha^\star(X), X) - \mu(\widehat{\alpha}_g, X) \mid X \in g\right].$$

For each group $g$, decompose the inner term into approximation and estimation components:

$$\mathbb{E}\left[\mu(\alpha^\star(X), X) - \mu(\widehat{\alpha}_g, X) \mid X \in g\right]$$
$$\leq \underbrace{CLd_g^\beta}_{\text{grouping approximation}} + \underbrace{\mathbb{E}\left[\mu_g(\alpha_g^\star) - \mu_g(\widehat{\alpha}_g)\right]}_{\text{within-group base learner error}}.$$

Using the base-learner condition (12), we obtain

$$\mathbb{E}\left[\mu(\alpha^\star(X), X) - \mu(\widehat{\alpha}_g, X) \mid X \in g\right]$$
$$\leq CLd_g^\beta + C_B\Phi(K, \sigma)(n_g^{\mathrm{eff}})^{-\zeta} + C_{\mathrm{loc}}Ld_g^\beta$$
$$\leq L_{\mathrm{eff}}d_g^\beta + C_B\Phi(K, \sigma)(n_g^{\mathrm{eff}})^{-\zeta}.$$

Therefore,

$$\mathbb{E}[S(T)] \leq \sum_{g \in G_T} \mathbb{P}(X \in g)\left[L_{\mathrm{eff}}d_g^\beta + C_B\Phi(K, \sigma)(n_g^{\mathrm{eff}})^{-\zeta}\right]. \tag{28}$$

By Lemma G.3 with $(\rho, \Lambda) = (\zeta, \Phi)$,

$$d_g \asymp \left( \frac{\Phi(K, \sigma)}{L_{\text{eff}} s_g^\zeta T^\zeta} \right)^{\frac{1}{\beta + d\zeta}}.$$

Consequently,

$$L_{\text{eff}} d_g^\beta = \widetilde{O}\left( L_{\text{eff}}^{\frac{d\zeta}{\beta + d\zeta}} \Phi(K, \sigma)^{\frac{\beta}{\beta + d\zeta}} s_g^{-\frac{\beta\zeta}{\beta + d\zeta}} T^{-\frac{\beta\zeta}{\beta + d\zeta}} \right).$$

The Zooming balance also implies that the statistical term $\Phi(K, \sigma)(n_g^{\text{eff}})^{-\zeta}$ is of the same order. Therefore, using $s_g \geq \Gamma_{\min}$ and $\sum_{g \in G_T} \mathbb{P}(X \in g) = 1$ in (28), we obtain

$$\mathbb{E}[S(T)] \leq \widetilde{O}\left( L_{\text{eff}}^{\frac{d\zeta}{\beta + d\zeta}} \Phi(K, \sigma)^{\frac{\beta}{\beta + d\zeta}} \Gamma_{\min}^{-\frac{\beta\zeta}{\beta + d\zeta}} T^{-\frac{\beta\zeta}{\beta + d\zeta}} \right). \tag{29}$$

Since $L_{\text{eff}}$ is only a constant multiple of $L$, this proves

$$\mathbb{E}[S(T)] \leq \widetilde{O}\left( L^{\frac{d\zeta}{\beta + d\zeta}} \Phi(K, \sigma)^{\frac{\beta}{\beta + d\zeta}} \Gamma_{\min}^{-\frac{\beta\zeta}{\beta + d\zeta}} T^{-\frac{\beta\zeta}{\beta + d\zeta}} \right).$$

For the stochastic best-arm identification base learners used in our paper, $\zeta = 1/2$. In this case,

$$\mathbb{E}[S(T)] \leq \widetilde{O}\left( L^{\frac{d}{2\beta + d}} \Phi(K, \sigma)^{\frac{2\beta}{2\beta + d}} \Gamma_{\min}^{-\frac{\beta}{2\beta + d}} T^{-\frac{\beta}{2\beta + d}} \right).$$

**Cumulative regret decomposition.** For the cumulative-regret analysis, instantiate the generic Zooming rule with

$$\rho = 1 - \gamma, \qquad \Lambda(K, \sigma) = \Psi(K, \sigma).$$

Let $n_g$ be the number of visits to group $g$. The cumulative regret decomposes as

$$R(T) = \sum_{g \in G_T} \sum_{t: X_t \in g} [\mu(\alpha^\star(X_t), X_t) - \mu(\alpha_t, X_t)].$$

For each group $g$, the regret consists of a cumulative approximation term and the within-group base-learner regret:

$$R_g(n_g) \leq C n_g L d_g^\beta + R_{B,g}(n_g).$$

Using the cumulative base-learner condition (13), we have

$$\mathbb{E}[R_g(n_g)] \leq C n_g L d_g^\beta + C_B \Psi(K, \sigma) n_g (n_g^{\text{eff}})^{-(1-\gamma)} + C_{\text{loc}} n_g L d_g^\beta$$
$$\leq L_{\text{eff}} n_g d_g^\beta + C_B \Psi(K, \sigma) n_g (n_g^{\text{eff}})^{-(1-\gamma)}.$$

Thus,

$$\mathbb{E}[R(T)] \leq \sum_{g \in G_T} \mathbb{E}[n_g] \left[ L_{\text{eff}} d_g^\beta + C_B \Psi(K, \sigma)(n_g^{\text{eff}})^{-(1-\gamma)} \right]. \tag{30}$$

By Lemma G.3 with $(\rho, \Lambda) = (1 - \gamma, \Psi)$,

$$d_g \asymp \left( \frac{\Psi(K, \sigma)}{L_{\text{eff}} s_g^{1-\gamma} T^{1-\gamma}} \right)^{\frac{1}{\beta + d(1-\gamma)}}.$$

Consequently,

$$L_{\text{eff}} d_g^\beta = \widetilde{O}\left( L_{\text{eff}}^{\frac{d(1-\gamma)}{\beta + d(1-\gamma)}} \Psi(K, \sigma)^{\frac{\beta}{\beta + d(1-\gamma)}} s_g^{-\frac{\beta(1-\gamma)}{\beta + d(1-\gamma)}} T^{-\frac{\beta(1-\gamma)}{\beta + d(1-\gamma)}} \right).$$

The Zooming balance implies that the reducible statistical term $\Psi(K,\sigma)(n_g^{\text{eff}})^{-(1-\gamma)}$ is of the same order. Using $s_g \geq \Gamma_{\min}$ and $\sum_{g \in G_T} \mathbb{E}[n_g] = T$ in (30), we obtain

$$\mathbb{E}[R(T)] \leq \widetilde{O}\left( L_{\text{eff}}^{\frac{d(1-\gamma)}{\beta+d(1-\gamma)}} \Psi(K,\sigma)^{\frac{\beta}{\beta+d(1-\gamma)}} \Gamma_{\min}^{-\frac{\beta(1-\gamma)}{\beta+d(1-\gamma)}} T^{1-\frac{\beta(1-\gamma)}{\beta+d(1-\gamma)}} \right). \tag{31}$$

Since $L_{\text{eff}}$ is only a constant multiple of $L$, this proves

$$\mathbb{E}[R(T)] \leq \widetilde{O}\left( L^{\frac{d(1-\gamma)}{\beta+d(1-\gamma)}} \Psi(K,\sigma)^{\frac{\beta}{\beta+d(1-\gamma)}} \Gamma_{\min}^{-\frac{\beta(1-\gamma)}{\beta+d(1-\gamma)}} T^{1-\frac{\beta(1-\gamma)}{\beta+d(1-\gamma)}} \right).$$

For the stochastic cumulative-regret base learners used in our paper, $\gamma = 1/2$. Therefore,

$$\mathbb{E}[R(T)] \leq \widetilde{O}\left( L^{\frac{d}{2\beta+d}} \Psi(K,\sigma)^{\frac{2\beta}{2\beta+d}} \Gamma_{\min}^{-\frac{\beta}{2\beta+d}} T^{\frac{\beta+d}{2\beta+d}} \right).$$

**Corollaries for traditional and counterfactual updates.** The corollaries follow by substituting the corresponding base-learner factors. For simple regret with Sequential Halving, $\zeta = 1/2$. Under single-point updates,

$$\Phi_{\text{sp}}(K,\sigma) = O(\sigma\sqrt{K}),$$

and under counterfactual updates,

$$\Phi_{\text{cf}}(K,\sigma) = \widetilde{O}(\sigma\sqrt{\log K}).$$

Substituting these into the simple-regret bound gives

$$H(K,\sigma) = \begin{cases} O\left(\sigma^{\frac{2\beta}{2\beta+d}} K^{\frac{\beta}{2\beta+d}}\right), & \text{Single Point,} \\ \widetilde{O}\left(\sigma^{\frac{2\beta}{2\beta+d}} (\log K)^{\frac{\beta}{2\beta+d}}\right), & \text{Counterfactual.} \end{cases}$$

This proves Corollary 5.3.

Similarly, for cumulative regret with Successive Elimination, $\gamma = 1/2$. Under single-point updates,

$$\Psi_{\text{sp}}(K,\sigma) = O(\sigma\sqrt{K}),$$

and under counterfactual updates,

$$\Psi_{\text{cf}}(K,\sigma) = \widetilde{O}(\sigma\sqrt{\log K}).$$

Substituting these into the cumulative-regret bound gives

$$J(K,\sigma) = \begin{cases} O\left(\sigma^{\frac{2\beta}{2\beta+d}} K^{\frac{\beta}{2\beta+d}}\right), & \text{Single Point,} \\ \widetilde{O}\left(\sigma^{\frac{2\beta}{2\beta+d}} (\log K)^{\frac{\beta}{2\beta+d}}\right), & \text{Counterfactual.} \end{cases}$$

This proves Corollary 5.4.

**Consistency with the operational Zooming rule.** We finally explain why the generic proof is consistent with the operational Zooming rule in the main text. For the two base learners instantiated in the paper,

$$\zeta = \frac{1}{2}, \qquad \gamma = \frac{1}{2}, \qquad 1-\gamma = \frac{1}{2}.$$

Therefore both the simple-regret and cumulative-regret Zooming balances use the square-root statistical scale. Up to logarithmic factors and constant multipliers, the generic rule

$$\Lambda(K,\sigma)(n_g^{\text{eff}})^{-\rho} \leq \eta L_{\text{eff}} d_g^{\beta}$$

reduces to

$$C\sigma\sqrt{\frac{\log(KT)}{n_g^{\text{eff}}}} \leq \eta L d_g^\beta,$$

which is Eq. (3). Thus the statistically optimal diameter and the algorithmically stable diameter have the same scaling. This consistency is not accidental: the Zooming Mechanism is derived from the same bias–variance trade-off that appears in the regret decomposition.

The role of the local structured bias is now transparent. It only replaces $L$ by $L_{\text{eff}} = C_{\text{eff}} L$ in the bias term. Since the functional form of the balance equation is unchanged, the exponents in $T$, $d$, and $K$ are unchanged. This is why the counterfactual and information-sharing biases do not affect the horizon exponent of the regret bounds.

### G.3. Extension to High-Probability Bounds

The expected-regret analysis above can be extended to high-probability bounds by incorporating an explicit failure probability $\delta$ into both the Zooming Mechanism and the within-group confidence radii. The key point is that the same base-learner-dependent zooming principle continues to apply, with logarithmic factors in $1/\delta$ added to the statistical radius.

To state the argument uniformly, let

$$\Lambda_\delta(K,\sigma)(n_g^{\text{eff}})^{-\rho}$$

denote the high-probability statistical error scale of the within-group base learner, where $\rho \in (0, 1/2]$. For simple regret, we take

$$\rho = \zeta, \qquad \Lambda_\delta(K,\sigma) = \Phi_\delta(K,\sigma),$$

where $\Phi_\delta(K,\sigma)$ has the same $K$-dependence as $\Phi(K,\sigma)$ and differs only by logarithmic factors in $T$ and $1/\delta$. For cumulative regret, we take

$$\rho = 1 - \gamma, \qquad \Lambda_\delta(K,\sigma) = \Psi_\delta(K,\sigma),$$

where $\Psi_\delta(K,\sigma)$ is defined analogously.

The $\delta$-dependent generic splitting rule is

$$\Lambda_\delta(K,\sigma)(n_g^{\text{eff}})^{-\rho} \leq \eta L_{\text{eff}} d_g^\beta. \tag{32}$$

Equivalently, since $n_g^{\text{eff}} = s_g n_g$, a group is split whenever

$$\Lambda_\delta(K,\sigma)(s_g n_g)^{-\rho} \leq \eta L_{\text{eff}} d_g^\beta.$$

This condition states that a group is refined only once its local approximation bias becomes statistically resolvable.

For the Sequential Halving and Successive Elimination instantiations used in the main text, $\rho = 1/2$ and the high-probability statistical radius takes the canonical sub-Gaussian form

$$\Lambda_\delta(K,\sigma) = C_{\text{z}}\sigma\sqrt{\log(KT/\delta)}.$$

Thus Eq. (32) specializes to

$$C_{\text{z}}\sigma\sqrt{\frac{\log(KT/\delta)}{n_g^{\text{eff}}}} \leq \eta L_{\text{eff}} d_g^\beta. \tag{33}$$

Equivalently,

$$C_{\text{z}}\sigma\sqrt{\frac{\log(KT/\delta)}{s_g n_g}} \leq \eta L_{\text{eff}} d_g^\beta.$$

Since $L_{\text{eff}}$ differs from $L$ only by a universal constant factor, the same rule can be written with $L$ after adjusting $\eta$.

Under this rule, a union bound over all groups, arms, and times yields that, with probability at least $1 - \delta$, the statistical fluctuation of every group-arm estimate satisfies

$$|\widehat{\mu}_g(\alpha) - \mathbb{E}\widehat{\mu}_g(\alpha)| \leq C_{\text{stat}}\Lambda_\delta(K,\sigma)(n_g^{\text{eff}})^{-\rho}, \qquad \forall g \in G_T, \ \forall \alpha \in \mathcal{A}.$$

For the square-root sub-Gaussian case, this reduces to

$$|\widehat{\mu}_g(\alpha) - \mathbb{E}\widehat{\mu}_g(\alpha)| \leq C\sigma\sqrt{\frac{\log(KT/\delta)}{n_g^{\mathrm{eff}}}}, \qquad \forall g \in G_T, \ \forall \alpha \in \mathcal{A}.$$

In addition, by the structured-bias argument in Lemma G.1, information sharing and counterfactual updates contribute only a local Hölder bias:

$$|\mathbb{E}\widehat{\mu}_g(\alpha) - \mu_g(\alpha)| \leq C_{\mathrm{sh}}L d_g^{\beta}.$$

Combining the two displays, on the same high-probability event,

$$|\widehat{\mu}_g(\alpha) - \mu_g(\alpha)| \leq C_{\mathrm{stat}}\Lambda_\delta(K,\sigma)(n_g^{\mathrm{eff}})^{-\rho} + C_{\mathrm{sh}}L d_g^{\beta}, \qquad \forall g, \alpha.$$

In the square-root sub-Gaussian case, this becomes

$$|\widehat{\mu}_g(\alpha) - \mu_g(\alpha)| \leq C\sigma\sqrt{\frac{\log(KT/\delta)}{n_g^{\mathrm{eff}}}} + C_{\mathrm{sh}}L d_g^{\beta}, \qquad \forall g, \alpha.$$

The rest of the proof follows the expected-regret analysis, with the statistical factor replaced by its high-probability counterpart. In the generic case, the mature-group balance becomes

$$\Lambda_\delta(K,\sigma)(s_g T d_g^d)^{-\rho} \asymp L_{\mathrm{eff}} d_g^{\beta}.$$

Solving this balance gives

$$d_g \asymp \left(\frac{\Lambda_\delta(K,\sigma)}{L_{\mathrm{eff}} s_g^{\rho} T^{\rho}}\right)^{\frac{1}{\beta+d\rho}}.$$

Equivalently,

$$L_{\mathrm{eff}} d_g^{\beta} = \widetilde{O}\left(L_{\mathrm{eff}}^{\frac{d\rho}{\beta+d\rho}} \Lambda_\delta(K,\sigma)^{\frac{\beta}{\beta+d\rho}} s_g^{-\frac{\beta\rho}{\beta+d\rho}} T^{-\frac{\beta\rho}{\beta+d\rho}}\right),$$

where $\widetilde{O}(\cdot)$ hides logarithmic factors in $T$ and $1/\delta$.

For the square-root sub-Gaussian specialization $\rho = 1/2$, this balance is equivalent to

$$d_g^{2\beta+d} \asymp \frac{\sigma^2 \log(KT/\delta)}{L_{\mathrm{eff}}^2 s_g T}.$$

Consequently, with probability at least $1 - \delta$, the simple regret bound becomes

$$S(T) \leq \widetilde{O}\left(L_{\mathrm{eff}}^{\frac{d\zeta}{\beta+d\zeta}} \Phi_\delta(K,\sigma)^{\frac{\beta}{\beta+d\zeta}} \Gamma_{\min}^{-\frac{\beta\zeta}{\beta+d\zeta}} T^{-\frac{\beta\zeta}{\beta+d\zeta}}\right). \tag{34}$$

Here $\Phi_\delta(K,\sigma)$ has the same arm dependence as $\Phi(K,\sigma)$, while the additional dependence on $\delta$ is logarithmic.

Similarly, with probability at least $1 - \delta$, the cumulative regret bound becomes

$$R(T) \leq \widetilde{O}\left(L_{\mathrm{eff}}^{\frac{d(1-\gamma)}{\beta+d(1-\gamma)}} \Psi_\delta(K,\sigma)^{\frac{\beta}{\beta+d(1-\gamma)}} \Gamma_{\min}^{-\frac{\beta(1-\gamma)}{\beta+d(1-\gamma)}} T^{1-\frac{\beta(1-\gamma)}{\beta+d(1-\gamma)}}\right). \tag{35}$$

Again, $\Psi_\delta(K,\sigma)$ has the same arm dependence as $\Psi(K,\sigma)$, with only logarithmic dependence on $T$ and $1/\delta$.

If one wants to state the result in expectation using the high-probability event, the failure event contributes at most $O(\delta)$ to simple regret and at most $O(T\delta)$ to cumulative regret, since rewards are bounded. Thus,

$$\mathbb{E}[S(T)] \leq \text{RHS of (34)} + O(\delta),$$

and

$$\mathbb{E}[R(T)] \leq \text{RHS of (35)} + O(T\delta).$$

Setting, for example, $\delta = 1/T$ makes the additional failure-event term lower order for cumulative regret and negligible for simple regret up to logarithmic factors. Therefore, the high-probability analysis yields the same asymptotic rates as the expected-regret bounds in Appendix G.2.

### G.4. Proof of Corollary 5.3

We instantiate the generic simple-regret bound in Appendix G.2 with sequential halving as the within-group simple-regret base learner. From (29), for a base learner with simple-regret exponent $\zeta$, we have

$$\mathbb{E}[S(T)] \leq \widetilde{O}\left( L_{\text{eff}}^{\frac{d\zeta}{\beta+d\zeta}} \Phi(K,\sigma)^{\frac{\beta}{\beta+d\zeta}} \Gamma_{\min}^{-\frac{\beta\zeta}{\beta+d\zeta}} T^{-\frac{\beta\zeta}{\beta+d\zeta}} \right).$$

For sequential halving, the simple-regret exponent is

$$\zeta = \frac{1}{2}.$$

Therefore,

$$\mathbb{E}[S(T)] \leq \widetilde{O}\left( L_{\text{eff}}^{\frac{d}{2\beta+d}} \Phi(K,\sigma)^{\frac{2\beta}{2\beta+d}} \Gamma_{\min}^{-\frac{\beta}{2\beta+d}} T^{-\frac{\beta}{2\beta+d}} \right).$$

Since $L_{\text{eff}}$ differs from $L$ only by a universal constant factor, we write the final rates in terms of $L$.

**Single-point update.** For standard sequential halving without counterfactual updates, the within-group simple-regret factor is

$$\Phi_{\text{sp}}(K,\sigma) = \widetilde{O}(\sigma\sqrt{K}).$$

Substituting this into the preceding display gives

$$\mathbb{E}[S_{\text{sp}}(T)] \leq \widetilde{O}\left( L^{\frac{d}{2\beta+d}} (\sigma\sqrt{K})^{\frac{2\beta}{2\beta+d}} \Gamma_{\min}^{-\frac{\beta}{2\beta+d}} T^{-\frac{\beta}{2\beta+d}} \right)$$

$$= \widetilde{O}\left( L^{\frac{d}{2\beta+d}} \sigma^{\frac{2\beta}{2\beta+d}} K^{\frac{\beta}{2\beta+d}} \Gamma_{\min}^{-\frac{\beta}{2\beta+d}} T^{-\frac{\beta}{2\beta+d}} \right).$$

Thus, without counterfactual updates, the arm dependence is $K^{\beta/(2\beta+d)}$.

**Counterfactual update.** Appendix F.2 shows that the counterfactual-aware sequential halving base learner satisfies

$$\Phi_{\text{cf}}(K,\sigma) = \widetilde{O}(\sigma\sqrt{\log K}),$$

up to the additive local approximation term $Ld_g^\beta$, which is already absorbed into $L_{\text{eff}}$ in the meta-analysis. Substituting this factor gives

$$\mathbb{E}[S_{\text{cf}}(T)] \leq \widetilde{O}\left( L^{\frac{d}{2\beta+d}} (\sigma\sqrt{\log K})^{\frac{2\beta}{2\beta+d}} \Gamma_{\min}^{-\frac{\beta}{2\beta+d}} T^{-\frac{\beta}{2\beta+d}} \right)$$

$$= \widetilde{O}\left( L^{\frac{d}{2\beta+d}} \sigma^{\frac{2\beta}{2\beta+d}} (\log K)^{\frac{\beta}{2\beta+d}} \Gamma_{\min}^{-\frac{\beta}{2\beta+d}} T^{-\frac{\beta}{2\beta+d}} \right).$$

Therefore, counterfactual updates reduce the arm dependence from

$$K^{\frac{\beta}{2\beta+d}} \quad \text{to} \quad (\log K)^{\frac{\beta}{2\beta+d}},$$

while preserving the contextual minimax rate $T^{-\beta/(2\beta+d)}$. This proves Corollary 5.3.

### G.5. Proof of Corollary 5.4

We next instantiate the generic cumulative-regret bound with successive elimination as the within-group cumulative-regret base learner. From (31), for a base learner with cumulative regret exponent $\gamma$, we have

$$\mathbb{E}[R(T)] \leq \widetilde{O}\left( L_{\text{eff}}^{\frac{d(1-\gamma)}{\beta+d(1-\gamma)}} \Psi(K,\sigma)^{\frac{\beta}{\beta+d(1-\gamma)}} \Gamma_{\min}^{-\frac{\beta(1-\gamma)}{\beta+d(1-\gamma)}} T^{1-\frac{\beta(1-\gamma)}{\beta+d(1-\gamma)}} \right).$$

For successive elimination, the cumulative-regret exponent is

$$\gamma = \frac{1}{2}.$$

Therefore,

$$\mathbb{E}[R(T)] \leq \widetilde{O}\left(L_{\text{eff}}^{\frac{d}{2\beta+d}} \Psi(K,\sigma)^{\frac{2\beta}{2\beta+d}} \Gamma_{\min}^{-\frac{\beta}{2\beta+d}} T^{\frac{\beta+d}{2\beta+d}}\right).$$

Again, since $L_{\text{eff}}$ is only a constant multiple of $L$, we write the final rates in terms of $L$.

**Single-point update.** For standard successive elimination without counterfactual updates, the within-group cumulative-regret factor is

$$\Psi_{\text{sp}}(K,\sigma) = \widetilde{O}(\sigma\sqrt{K}).$$

Substituting this factor gives

$$\mathbb{E}[R_{\text{sp}}(T)] \leq \widetilde{O}\left(L^{\frac{d}{2\beta+d}}(\sigma\sqrt{K})^{\frac{2\beta}{2\beta+d}} \Gamma_{\min}^{-\frac{\beta}{2\beta+d}} T^{\frac{\beta+d}{2\beta+d}}\right)$$
$$= \widetilde{O}\left(L^{\frac{d}{2\beta+d}} \sigma^{\frac{2\beta}{2\beta+d}} K^{\frac{\beta}{2\beta+d}} \Gamma_{\min}^{-\frac{\beta}{2\beta+d}} T^{\frac{\beta+d}{2\beta+d}}\right).$$

Thus, without counterfactual updates, the arm dependence is $K^{\beta/(2\beta+d)}$.

**Counterfactual update.** Appendix F.1 shows that the counterfactual-aware successive elimination base learner satisfies

$$\Psi_{\text{cf}}(K,\sigma) = \widetilde{O}(\sigma\sqrt{\log K}),$$

up to the additive local approximation term $n_g L d_g^\beta$, which is absorbed into $L_{\text{eff}}$ in the AGCB meta-analysis. Substituting this factor yields

$$\mathbb{E}[R_{\text{cf}}(T)] \leq \widetilde{O}\left(L^{\frac{d}{2\beta+d}}(\sigma\sqrt{\log K})^{\frac{2\beta}{2\beta+d}} \Gamma_{\min}^{-\frac{\beta}{2\beta+d}} T^{\frac{\beta+d}{2\beta+d}}\right)$$
$$= \widetilde{O}\left(L^{\frac{d}{2\beta+d}} \sigma^{\frac{2\beta}{2\beta+d}} (\log K)^{\frac{\beta}{2\beta+d}} \Gamma_{\min}^{-\frac{\beta}{2\beta+d}} T^{\frac{\beta+d}{2\beta+d}}\right).$$

Therefore, counterfactual updates reduce the arm dependence from

$$K^{\frac{\beta}{2\beta+d}} \quad \text{to} \quad (\log K)^{\frac{\beta}{2\beta+d}},$$

while preserving the contextual minimax cumulative-regret rate $T^{(\beta+d)/(2\beta+d)}$. This proves Corollary 5.4.

*Remark* G.6 (Comparison with standard minimax lower bounds). For standard nonparametric contextual bandits under Hölder continuity and without additional arm-order structure, the minimax lower bounds scale as

$$\Omega\left(K^{\frac{\beta}{2\beta+d}} T^{-\frac{\beta}{2\beta+d}}\right) \qquad \text{for simple regret,}$$

and

$$\Omega\left(K^{\frac{\beta}{2\beta+d}} T^{\frac{\beta+d}{2\beta+d}}\right) \qquad \text{for cumulative regret.}$$

By exploiting the ordered significance-level arms and conditional monotonicity under coverage, AGCB with counterfactual updates achieves

$$\widetilde{O}\left((\log K)^{\frac{\beta}{2\beta+d}} T^{-\frac{\beta}{2\beta+d}}\right) \qquad \text{for simple regret,}$$

and

$$\widetilde{O}\left((\log K)^{\frac{\beta}{2\beta+d}} T^{\frac{\beta+d}{2\beta+d}}\right) \qquad \text{for cumulative regret,}$$

Thus, our minimax-optimality claim refers to matching the standard optimal exponents in the horizon $T$. The improved logarithmic dependence on $K$ reflects the additional monotonicity structure, consistent with the counterfactual-update phenomenon in Straitouri & Gomez Rodriguez (2024).

### G.6. Why Structured Bias Does Not Hurt the Optimal Rate

A central feature of AGCB is that it deliberately introduces structured bias through information sharing and counterfactual updates. We show that this bias has the same Hölder scaling as the ordinary grouping approximation bias. Consequently, it only changes constants in the regret bounds and does not change the minimax rate.

**Error decomposition.** For a group $g$ and an arm $\alpha$, decompose the estimation error as

$$\widehat{\mu}_g(\alpha) - \mu_g(\alpha) = \underbrace{\widehat{\mu}_g(\alpha) - \mathbb{E}\widehat{\mu}_g(\alpha)}_{\text{statistical fluctuation}} + \underbrace{\mathbb{E}\widehat{\mu}_g(\alpha) - \mu_g(\alpha)}_{\text{structured bias}}.$$

In addition, the use of a single group-level arm for all contexts in $g$ introduces the usual grouping approximation bias:

$$\sup_{\alpha \in \mathcal{A}} |\mu(\alpha, x) - \mu(\alpha, c_g)| \leq Ld_g^\beta, \qquad x \in g,$$

where $c_g$ is a representative context of group $g$.

**Lemma G.7** (Structured bias is local). *Suppose the information-sharing kernel across groups and the counterfactual kernel across arms assign non-negligible weight only to context-arm pairs within distance proportional to the local group scale $d_g$ in the joint metric. Under Assumption 4.1, for every group $g$ and arm $\alpha$,*

$$|\mathbb{E}\widehat{\mu}_g(\alpha) - \mu_g(\alpha)| \leq C_{\text{str}}Ld_g^\beta.$$

*Proof.* Both information sharing and counterfactual updates replace the target context-arm pair $(x, \alpha)$ by nearby pairs $(x', \alpha')$. By the kernel bandwidth choice, these pairs have joint distance at most a constant multiple of the local group scale $d_g$. Therefore, Assumption 4.1 implies

$$|\mu(\alpha, x) - \mu(\alpha', x')| \leq Ld((x, \alpha), (x', \alpha'))^\beta \leq CLd_g^\beta.$$

Averaging this bound over the information-sharing and counterfactual kernel weights gives the claim. $\square$

Combining the ordinary grouping bias and the structured bias gives

$$\text{total local bias} \leq CLd_g^\beta + C_{\text{str}}Ld_g^\beta = L_{\text{eff}}d_g^\beta,$$

where $L_{\text{eff}} = C_{\text{eff}}L$ for a universal constant $C_{\text{eff}} > 0$. Thus the structured bias only changes the effective smoothness constant; it does not change the functional form of the bias term.

**Interaction with the Zooming Mechanism.** At equilibrium, the Zooming Mechanism balances the local statistical uncertainty with the local approximation bias:

$$\sigma\sqrt{\frac{\log(KT)}{n_g^{\text{eff}}}} \asymp Ld_g^\beta.$$

After incorporating the structured bias, the balance becomes

$$\sigma\sqrt{\frac{\log(KT)}{n_g^{\text{eff}}}} \asymp L_{\text{eff}}d_g^\beta.$$

This has the same form as the original balance equation. Solving it gives

$$d_g^{2\beta+d} \asymp \frac{\sigma^2 \log(KT)}{L_{\text{eff}}^2 s_g T}.$$

The dependence on $T$, $d$, and $\beta$ is unchanged. Only the constant depending on $L$ is modified.

**Rate preservation.** Substituting this balanced diameter into the regret decompositions of Appendix G.2 gives the same rates:

$$\mathbb{E}[S(T)] = \widetilde{O}\left(T^{-\frac{\beta}{2\beta+d}}\right)$$

for simple regret when $\zeta = 1/2$, and

$$\mathbb{E}[R(T)] = \widetilde{O}\left(T^{\frac{\beta+d}{2\beta+d}}\right)$$

for cumulative regret when $\gamma = 1/2$, with the corresponding arm-space factors determined by $\Phi(K, \sigma)$ and $\Psi(K, \sigma)$.

Therefore, the structured bias introduced by information sharing and counterfactual updates does not introduce a slower-decaying error term. It is absorbed into the same Hölder bias term that the Zooming Mechanism already balances against statistical uncertainty.

### G.7. Robustness of Elimination Under Structured Bias

We finally explain why the structured bias does not invalidate the elimination steps used by the within-group base learners. The key point is that the elimination rules in sequential halving and successive elimination depend on relative comparisons between arms. Structured bias may shift the estimates, but under Hölder continuity this shift is locally smooth, and its differential effect across arms is no larger than the local resolution already accounted for by the Zooming Mechanism.

**Relative comparison under structured bias.** Let $B_g(\alpha)$ denote the structured bias of arm $\alpha$ in group $g$:

$$B_g(\alpha) := \mathbb{E}\widehat{\mu}_g(\alpha) - \mu_g(\alpha).$$

For two arms $\alpha$ and $\alpha'$, the estimated difference decomposes as

$$\widehat{\mu}_g(\alpha) - \widehat{\mu}_g(\alpha') = \underbrace{\mu_g(\alpha) - \mu_g(\alpha')}_{\text{true reward difference}} + \underbrace{B_g(\alpha) - B_g(\alpha')}_{\text{differential structured bias}}$$
$$+ \underbrace{\left[\widehat{\mu}_g(\alpha) - \mathbb{E}\widehat{\mu}_g(\alpha)\right] - \left[\widehat{\mu}_g(\alpha') - \mathbb{E}\widehat{\mu}_g(\alpha')\right]}_{\text{zero-mean statistical fluctuation}}.$$

By Lemma G.7, and by the local bandwidth choice of the counterfactual kernel, the differential bias is bounded as

$$|B_g(\alpha) - B_g(\alpha')| \leq CLd_g^{\beta}$$

for arms compared inside the same local group. The statistical fluctuation is controlled by the usual confidence radius:

$$\mathrm{rad}_g = C\sigma\sqrt{\frac{\log(KT)}{n_g^{\mathrm{eff}}}}.$$

At the zooming equilibrium,

$$\mathrm{rad}_g \asymp Ld_g^{\beta}.$$

Therefore, the maximum comparison distortion caused by structured bias is of the same order as the statistical uncertainty already used by the base learners.

**Implication for local safe pruning.** The preceding display implies that any incorrect elimination caused by structured bias can only happen between arms whose true group-level gap is at most

$$O\left(Ld_g^{\beta} + \sigma\sqrt{\frac{\log(KT)}{n_g^{\mathrm{eff}}}}\right).$$

This is exactly the local safe-pruning resolution used in Appendix F.1 and Appendix F.2. If the true gap is larger than this threshold, the combination of statistical noise and differential bias is insufficient to reverse the comparison with high probability. If the true gap is smaller than this threshold, then the two arms are statistically indistinguishable at the current local resolution, and eliminating either one contributes only a lower-order local regret term.

**Implications for the two base learners.** For counterfactual sequential halving, each phase discards a portion of the active interval. Under local safe pruning, the exact best arm may be removed only if some retained arm is within the local resolution. Hence the final simple regret contains an additive $Ld_g^{\beta}$ term, which is precisely the term absorbed in Appendix G.2.

For counterfactual successive elimination, the algorithm removes arms whose confidence intervals are dominated by those of other active arms. Adding structured bias of order $Ld_g^{\beta}$ only enlarges the confidence threshold by a constant factor at the zooming equilibrium. Hence the cumulative regret contains an additive $n_g Ld_g^{\beta}$ term inside group $g$, which is again absorbed into the group-level bias term in the AGCB meta-analysis.

In summary, structured bias does not create a new source of elimination error that decays more slowly than the statistical uncertainty. Hölder continuity ensures that the bias is local and smooth, while the Zooming Mechanism calibrates the group diameter so that the maximum possible bias distortion is balanced against the confidence radius. Therefore, the high-probability and expected regret guarantees retain the same asymptotic rates.

## G.8. Discrete Arms, $K$-Dependent Smoothness, and Discretization Error

The main regret bounds assume a population-level Hölder condition on the continuous context–parameter space. Under this convention, the smoothness parameters $(\beta, L)$ are independent of the number of candidate arms $K$. This is the setting used in Theorem 5.1. In some applications, however, one may prefer to define smoothness only on the finite arm set $\mathcal{A}_K$. This appendix explains how the regret bounds change in that case and how to account for discretization error when the benchmark is continuous in $\alpha$.

Throughout this subsection, $\widetilde{O}$ hides universal constants and logarithmic factors in $T$, while all dependence on $K$ is displayed explicitly.

**Population-level smoothness.** Suppose there exists an underlying population reward function $\mu(\alpha, x)$ on $\mathcal{X} \times [0, 1]$ satisfying

$$|\mu(\alpha, x) - \mu(\alpha', x')| \leq L \left(\|x - x'\| + \lambda|\alpha - \alpha'|\right)^{\beta}.$$

The finite arm set $\mathcal{A}_K = \{\alpha_1, \ldots, \alpha_K\}$ is then a discretization of the same continuous parameter space. In this case, $(\beta, L)$ do not depend on $K$. Increasing $K$ decreases the spacing between neighboring arms, but the smoothness exponent remains a property of the population reward function. The $K$-dependence in the regret bounds comes only from the within-group arm-search factor, such as $\sqrt{K}$ for single-point updates or $\sqrt{\log K}$ for counterfactual updates.

**Discrete-arm effective smoothness.** If smoothness is only assumed on the finite set $\mathcal{A}_K$, then the effective Hölder parameters may depend on $K$. Specifically, suppose that for each $K$ there exist $(\beta_K, L_K)$ such that

$$|\mu(\alpha, x) - \mu(\alpha', x')| \leq L_K \left(\|x - x'\| + \lambda|\alpha - \alpha'|\right)^{\beta_K}, \qquad \alpha, \alpha' \in \mathcal{A}_K.$$

Then the same AGCB analysis applies after replacing $(\beta, L)$ by $(\beta_K, L_K)$.

For simple regret, the generic bound becomes

$$\mathbb{E}[S_K(T)] \leq \widetilde{O}\left(L_K^{\frac{d\zeta}{\beta_K + d\zeta}} \Phi(K, \sigma)^{\frac{\beta_K}{\beta_K + d\zeta}} \Gamma_{\min}^{-\frac{\beta_K \zeta}{\beta_K + d\zeta}} T^{-\frac{\beta_K \zeta}{\beta_K + d\zeta}}\right).$$

For cumulative regret, the corresponding bound is

$$\mathbb{E}[R_K(T)] \leq \widetilde{O}\left(L_K^{\frac{d(1-\gamma)}{\beta_K + d(1-\gamma)}} \Psi(K, \sigma)^{\frac{\beta_K}{\beta_K + d(1-\gamma)}} \Gamma_{\min}^{-\frac{\beta_K(1-\gamma)}{\beta_K + d(1-\gamma)}} T^{1 - \frac{\beta_K(1-\gamma)}{\beta_K + d(1-\gamma)}}\right).$$

In particular, for the counterfactual learners analyzed in Appendix F, we have $\zeta = \gamma = 1/2$ and

$$\Phi_{\mathrm{cf}}(K, \sigma) = \Psi_{\mathrm{cf}}(K, \sigma) = \widetilde{O}(\sigma\sqrt{\log K}).$$

Thus, the simple-regret bound becomes

$$\mathbb{E}[S_K(T)] \leq \widetilde{O}\left(L_K^{\frac{d}{2\beta_K + d}} \sigma^{\frac{2\beta_K}{2\beta_K + d}} (\log K)^{\frac{\beta_K}{2\beta_K + d}} \Gamma_{\min}^{-\frac{\beta_K}{2\beta_K + d}} T^{-\frac{\beta_K}{2\beta_K + d}}\right),$$

and the cumulative-regret bound becomes

$$\mathbb{E}[R_K(T)] \leq \widetilde{O}\left(L_K^{\frac{d}{2\beta_K + d}} \sigma^{\frac{2\beta_K}{2\beta_K + d}} (\log K)^{\frac{\beta_K}{2\beta_K + d}} \Gamma_{\min}^{-\frac{\beta_K}{2\beta_K + d}} T^{\frac{\beta_K + d}{2\beta_K + d}}\right).$$

These expressions make explicit how any degradation of the effective smoothness parameters with $K$ affects the regret. For example, if $\beta_K$ decreases with $K$, then the exponent $\beta_K/(2\beta_K + d)$ deteriorates. If $L_K$ grows with $K$, then the factor $L_K^{d/(2\beta_K + d)}$ introduces an additional $K$-dependent cost. Therefore, uniform rates over increasing arm sets require some uniform control, such as $\inf_K \beta_K > 0$ and moderate growth of $L_K$.

**Continuous benchmark and discretization error.** The regret definitions in the main text compare against the best arm in the finite set $\mathcal{A}_K$. If instead the benchmark is the best continuous parameter $\alpha \in [0, 1]$, then there is an additional discretization error.

Let

$$\Delta_K := \sup_{\alpha \in [0,1]} \min_{\alpha_j \in \mathcal{A}_K} |\alpha - \alpha_j|$$

be the mesh size of the arm discretization. If the population reward is Hölder continuous in $\alpha$ with exponent $\beta_\alpha$ and constant $L_\alpha$, then

$$D_K := \sup_x \left[ \max_{\alpha \in [0,1]} \mu(\alpha, x) - \max_{\alpha_j \in \mathcal{A}_K} \mu(\alpha, x_j) \right] \leq L_\alpha \Delta_K^{\beta_\alpha}.$$

Therefore, the continuous-benchmark simple regret satisfies

$$S_{\text{cont}}(T, K) \leq D_K + S_{\text{disc}}(T, K),$$

where $S_{\text{disc}}(T, K)$ is the regret relative to the best arm in $\mathcal{A}_K$. Similarly, for cumulative regret,

$$R_{\text{cont}}(T, K) \leq T D_K + R_{\text{disc}}(T, K).$$

For a uniform grid, $\Delta_K = O(K^{-1})$, so

$$D_K = O(K^{-\beta_\alpha}).$$

Thus, increasing $K$ reduces discretization error. The cost of doing so depends on the arm-search complexity of the base learner. With single-point updates, the discrete learning term has polynomial dependence on $K$; with counterfactual updates, it has only logarithmic dependence on $K$. This is one of the main advantages of the counterfactual update in fine-grained calibration problems.

**Growing arm sets.** The preceding bounds also allow $K$ to grow with $T$. For example, suppose the population smoothness parameters are fixed and the counterfactual simple-regret bound scales as

$$S_{\text{disc}}(T, K) = \widetilde{O}\left( T^{-\frac{\beta}{2\beta + d}} (\log K)^{\frac{\beta}{2\beta + d}} \right),$$

while the discretization error satisfies $D_K = O(K^{-\beta_\alpha})$. Choosing $K$ to grow polynomially with $T$ makes $D_K$ decrease polynomially while only adding logarithmic factors to the discrete learning term. In contrast, with single-point updates, the learning term contains a polynomial factor in $K$, so one must carefully balance discretization error against learning error.

This discussion shows that the main theorem corresponds to the uniform population-smoothness regime. If a problem exhibits $K$-dependent effective smoothness or if one benchmarks against the continuous optimum, the same analysis remains valid after adding the explicit terms above.

## H. Effect of Misspecified Hölder Parameters

This appendix analyzes how using misspecified Hölder parameters $(\beta_0, L_0)$ affects the Zooming Mechanism. The main conclusion is that misspecifying the smoothness exponent $\beta$ generally changes the partition scale and may lead to suboptimal regret rates. In contrast, if $\beta$ is correctly specified, misspecifying the Hölder constant $L$ changes the scale of the partition but not the rate exponent. This motivates the model-selection approach in Appendix I when the smoothness parameters are unknown.

Throughout this appendix, we focus on the dependence on $T$ and suppress common logarithmic factors, arm-dependent factors, and information-sharing constants. These factors do not affect the smoothness-misspecification exponents derived below.

### H.1. Balanced Group Diameters: True vs. Estimated Parameters

The Zooming Mechanism balances local statistical uncertainty against local approximation bias. Under the true parameters $(\beta, L)$, the ideal group diameter $d_g^\star$ satisfies

$$C\sigma \sqrt{\frac{\log(KT)}{s_g T (d_g^\star)^d}} \asymp \eta L (d_g^\star)^\beta.$$

Solving for $d_g^\star$ gives

$$d_g^\star \asymp \left( \frac{\sigma^2 \log(KT)}{L^2 s_g T} \right)^{\frac{1}{2\beta+d}} \asymp T^{-\frac{1}{2\beta+d}},$$

where the second relation suppresses constants and logarithmic factors.

If the algorithm instead uses estimated parameters $(\beta_0, L_0)$, the Zooming rule balances

$$C\sigma \sqrt{\frac{\log(KT)}{s_g T d_g^d}} \asymp \eta L_0 d_g^{\beta_0},$$

so the achieved group diameter satisfies

$$d_g \asymp \left( \frac{\sigma^2 \log(KT)}{L_0^2 s_g T} \right)^{\frac{1}{2\beta_0+d}} \asymp T^{-\frac{1}{2\beta_0+d}}.$$

Comparing $d_g$ with $d_g^\star$ gives

$$\frac{d_g}{d_g^\star} \asymp T^{\frac{1}{2\beta+d} - \frac{1}{2\beta_0+d}},$$

up to constants and logarithmic factors.

If $\beta_0 > \beta$, then

$$\frac{1}{2\beta+d} - \frac{1}{2\beta_0+d} > 0,$$

so $d_g/d_g^\star \to \infty$. The algorithm creates groups that are too coarse. If $\beta_0 < \beta$, then $d_g/d_g^\star \to 0$, so the algorithm creates groups that are too fine.

### H.2. Effect on Simple Regret

The simple regret at a representative group diameter $d_g$ has the usual bias–variance form

$$S(T; d_g) \lesssim \underbrace{L d_g^\beta}_{\text{true approximation bias}} + \underbrace{\sigma \sqrt{\frac{1}{T d_g^d}}}_{\text{statistical estimation error}},$$

where common logarithmic, arm-dependent, and information-sharing factors are suppressed.

Substituting the misspecified diameter $d_g \asymp T^{-1/(2\beta_0+d)}$ gives

$$L d_g^\beta \asymp T^{-\frac{\beta}{2\beta_0+d}}, \qquad \sigma \sqrt{\frac{1}{T d_g^d}} \asymp T^{-\frac{\beta_0}{2\beta_0+d}}.$$

Therefore,

$$S(T; \beta_0) \lesssim \widetilde{O}\left( T^{-\frac{\beta}{2\beta_0+d}} + T^{-\frac{\beta_0}{2\beta_0+d}} \right).$$

**Overestimating smoothness: $\beta_0 > \beta$.** When $\beta_0 > \beta$, the approximation-bias exponent is smaller:

$$\frac{\beta}{2\beta_0+d} < \frac{\beta_0}{2\beta_0+d}.$$

Thus the approximation bias dominates, and

$$S(T; \beta_0) = \widetilde{O}\left( T^{-\frac{\beta}{2\beta_0+d}} \right).$$

This is slower than the optimal rate

$$T^{-\frac{\beta}{2\beta+d}},$$

because $2\beta_0 + d > 2\beta + d$. Hence overestimating the smoothness exponent does not only change constants; it leads to overly coarse groups and a slower simple-regret rate.

**Underestimating smoothness: $\beta_0 < \beta$.** When $\beta_0 < \beta$, the statistical exponent is smaller:

$$\frac{\beta_0}{2\beta_0 + d} < \frac{\beta}{2\beta_0 + d}.$$

Thus the estimation error dominates, and

$$S(T; \beta_0) = \widetilde{O}\left(T^{-\frac{\beta_0}{2\beta_0 + d}}\right).$$

This is also slower than the optimal rate $T^{-\beta/(2\beta+d)}$. Underestimating the smoothness exponent creates overly fine groups and increases variance.

**Correct specification.** When $\beta_0 = \beta$, the two terms balance and the optimal simple-regret rate

$$\widetilde{O}\left(T^{-\frac{\beta}{2\beta + d}}\right)$$

is recovered.

## H.3. Effect on Cumulative Regret

The same phenomenon appears for cumulative regret. For a representative diameter $d_g$, the cumulative regret has the form

$$R(T; d_g) \lesssim \underbrace{TLd_g^\beta}_{\text{cumulative approximation bias}} + \underbrace{\sqrt{Td_g^{-d}}}_{\text{within-group learning cost}},$$

again suppressing common logarithmic, arm-dependent, and information-sharing factors.

Substituting $d_g \asymp T^{-1/(2\beta_0 + d)}$ gives

$$TLd_g^\beta \asymp T^{1 - \frac{\beta}{2\beta_0 + d}}, \qquad \sqrt{Td_g^{-d}} \asymp T^{\frac{\beta_0 + d}{2\beta_0 + d}}.$$

Therefore,

$$R(T; \beta_0) \lesssim \widetilde{O}\left(T^{1 - \frac{\beta}{2\beta_0 + d}} + T^{\frac{\beta_0 + d}{2\beta_0 + d}}\right).$$

If $\beta_0 > \beta$, the cumulative approximation-bias term dominates:

$$R(T; \beta_0) = \widetilde{O}\left(T^{1 - \frac{\beta}{2\beta_0 + d}}\right),$$

which is slower than the optimal cumulative-regret rate

$$T^{\frac{\beta + d}{2\beta + d}}.$$

If $\beta_0 < \beta$, the within-group learning cost dominates:

$$R(T; \beta_0) = \widetilde{O}\left(T^{\frac{\beta_0 + d}{2\beta_0 + d}}\right),$$

which is again slower than the optimal rate. Only when $\beta_0 = \beta$ do the two terms balance at

$$\widetilde{O}\left(T^{\frac{\beta + d}{2\beta + d}}\right).$$

## H.4. Effect of Misspecifying $L$

The Hölder constant $L$ affects the scale of the partition but not the rate exponent when the smoothness exponent $\beta$ is correctly specified. If $\beta_0 = \beta$ but the algorithm uses $L_0 \neq L$, then

$$d_g \asymp \left(\frac{1}{L_0^2 T}\right)^{\frac{1}{2\beta + d}}, \qquad d_g^\star \asymp \left(\frac{1}{L^2 T}\right)^{\frac{1}{2\beta + d}},$$

after suppressing common factors. Hence

$$\frac{d_g}{d_g^\star} \asymp \left(\frac{L}{L_0}\right)^{\frac{2}{2\beta+d}},$$

which is independent of $T$. Thus misspecifying $L$ changes constants but not the rate exponent, provided $\beta$ is correctly specified.

### H.5. Practical Implication

A fixed misspecified smoothness exponent $\beta_0$ generally changes the regret rate: overestimating $\beta$ produces overly coarse groups and dominant approximation bias, while underestimating $\beta$ produces overly fine groups and dominant variance. Therefore, when $\beta$ is unknown, a fixed conservative guess is not guaranteed to be rate-optimal. A principled solution is to use the model-selection wrapper in Appendix I, which competes with the best candidate smoothness parameter up to the standard model-selection penalty.

### H.6. Summary

The effect of misspecifying the Hölder exponent is not merely a constant-factor effect. For simple regret, using a fixed exponent $\beta_0$ leads to the rate

$$S(T;\beta_0) = \widetilde{O}\left(T^{-\frac{\beta}{2\beta_0+d}} + T^{-\frac{\beta_0}{2\beta_0+d}}\right),$$

up to common logarithmic and arm-dependent factors. For cumulative regret, the corresponding rate is

$$R(T;\beta_0) = \widetilde{O}\left(T^{1-\frac{\beta}{2\beta_0+d}} + T^{\frac{\beta_0+d}{2\beta_0+d}}\right).$$

The optimal rates are recovered when $\beta_0 = \beta$. Misspecifying $L$ alone affects constants but not the rate exponent. This motivates adaptive selection over candidate smoothness parameters.

## I. Adapting to Unknown Smoothness via Model Selection

This appendix explains how model-selection techniques can be integrated with AGCB to handle unknown Hölder parameters $(\beta, L)$. The main idea is to run several AGCB instances with different candidate parameter pairs and combine them using a master algorithm. This allows the learner to compete with the best candidate, up to the standard model-selection penalty.

### I.1. Discretization of the Parameter Space

Let $\{(\beta_i, L_i)\}_{i=1}^M$ be a finite collection of candidate Hölder parameters. Each candidate pair defines an AGCB instance $\mathcal{A}_i$ with its own Zooming Mechanism. The candidate grid should cover the plausible range of smoothness exponents and Hölder constants. Since the exponent $\beta$ controls the rate exponent in $T$, it is especially important that the grid covers the plausible values of $\beta$. The constant $L$ affects the partition scale and leading constants; it can be handled by a coarse grid or by a small set of representative values.

*Remark* I.1 (Role of $\beta$ and $L$). The Hölder exponent $\beta$ determines the exponent of $T$ in the regret rate, whereas $L$ affects the scale of the partition and the leading constant when $\beta$ is correctly specified. Therefore, adaptation to unknown smoothness primarily requires covering the possible values of $\beta$, while $L$ can often be handled with a coarser grid.

### I.2. Meta-Algorithm for Cumulative Regret

For cumulative regret, one can employ a model-selection master algorithm such as Corral (Pacchiano et al., 2020) to combine the $M$ AGCB instances. The master algorithm treats each $\mathcal{A}_i$ as a base algorithm and adaptively allocates rounds among them.

Applying such a model-selection wrapper yields a bound of the form

$$R(T) \le \widetilde{O}\left(\min_{i=1,\ldots,M} R_i(T) + \sqrt{T \log M}\right),$$

where $R_i(T)$ is the cumulative regret bound of the AGCB instance using $(\beta_i, L_i)$, and the second term is the standard model-selection penalty.

For example, when the $i$-th candidate is used with a cumulative-regret base learner of exponent $\gamma = 1/2$, Theorem 5.1 gives

$$R_i(T) \leq \widetilde{O}\left(L_i^{\frac{d}{2\beta_i+d}}\Psi(K,\sigma)^{\frac{2\beta_i}{2\beta_i+d}}\Gamma_{\min}^{-\frac{\beta_i}{2\beta_i+d}}T^{\frac{\beta_i+d}{2\beta_i+d}}\right).$$

Hence,

$$R(T) \leq \widetilde{O}\left(\min_i L_i^{\frac{d}{2\beta_i+d}}\Psi(K,\sigma)^{\frac{2\beta_i}{2\beta_i+d}}\Gamma_{\min}^{-\frac{\beta_i}{2\beta_i+d}}T^{\frac{\beta_i+d}{2\beta_i+d}} + \sqrt{T\log M}\right).$$

This is consistent with the known cost of adapting to unknown smoothness in nonparametric bandits (Locatelli & Carpentier, 2018). Since

$$\frac{\beta_i + d}{2\beta_i + d} > \frac{1}{2}$$

for every finite $\beta_i$ and $d > 0$, the model-selection penalty is lower order in the exponent, although it may still be relevant in constants and logarithmic factors when the problem is very smooth.

### I.3. Simple Regret

For simple regret, one can similarly combine AGCB instances using pure-exploration adaptation tools such as Parallel Optimistic Optimization (Grill et al., 2015) or nested aggregation (Locatelli et al., 2017). These methods can compete with the best candidate smoothness parameter up to logarithmic or model-selection factors. A full optimization of the simple-regret adaptation scheme is orthogonal to our main contribution; the key point is that AGCB can be used as the base algorithm inside existing adaptive smoothness-selection frameworks.

### I.4. Summary

Fixed misspecification of the smoothness exponent may lead to suboptimal rates, as shown in Appendix H. Model selection provides a principled way to adapt to unknown Hölder parameters: by running AGCB over a finite candidate grid and combining the instances, the resulting algorithm competes with the best candidate up to the standard model-selection penalty.

## J. Experimental Settings and Implementation Details

This appendix section provides the complete specification of our experimental framework, designed to validate the theoretical properties of our AGCB algorithm in realistic human-in-the-loop decision scenarios.

### J.1. Problem Formulation as Classification with Conformal Prediction

For instance, our experiments can be considered as a critical real-world application: adaptive decision support for human experts. Consider a medical diagnosis setting where:

- An AI system analyzes patient data (context $x$) and proposes possible diagnoses.

- The system can adjust its *precision level* $\alpha$: higher $\alpha$ means smaller, more precise prediction sets; lower $\alpha$ means larger, more conservative sets.

- A clinician reviews the proposed diagnoses and makes the final decision.

- The goal is to learn, for each patient type $x$, the optimal $\alpha$ that balances precision with the clinician's cognitive load.

Formally, We consider a 10-class classification problem. The classifier weights are represented by a matrix $W \in \mathbb{R}^{d\times 10}$, with entries independently drawn from a normal distribution: $W_{ij} \sim \mathcal{N}(0, 0.09)$. The matrix is generated once per experimental run using a fixed random seed. Given a context $x \in \mathbb{R}^d$, the true label $y$ is drawn from a categorical distribution with probabilities proportional to the exponentiated linear scores:

$$P(y = j \mid x) \propto \exp(x^\top W_{:,j}), \quad j \in \{1, \dots, 10\},$$

where $W_{:,j}$ denotes the $j$-th column of $W$. Thus, the true label $y$ is conditionally dependent on the context $x$ via the linear classifier.

## J.2. Context Space: Modeling Diverse Patient Populations

Patient data is represented by a $d$-dimensional feature vector designed to capture clinically relevant variations. For the 5-dimensional setting ($d = 5$):

- **Baseline Status** ($x_0$): Uniform in $[0, 1]$, representing general health indicators.

- **Risk Profile** ($x_1$): Beta(2,4) distributed, modeling skewed risk distributions (few high-risk, many low-risk patients).

- **Symptom Complexity** ($x_2$): Bimodal Gaussian mixture, distinguishing clear-cut cases (peak at 0.3) from complex presentations (peak at 0.7).

- **Urgency** ($x_3$): Clipped exponential, capturing the long-tail of emergency cases.

- **Demographic Group** ($x_4$): Bimodal uniform, representing two distinct population subgroups.

For the 10-dimensional setting ($d = 10$), we extend the feature space with five additional dimensions that capture more nuanced patient characteristics:

- **Treatment Response** ($x_5$): Bimodal Gaussian mixture modeling variability in treatment sensitivity.

- **Comorbidity Burden** ($x_6$): Beta(3,2) distributed, representing the presence of multiple conditions.

- **Genetic Marker** ($x_7$): Uniform in $[0, 1]$, simulating a continuous genetic risk score.

- **Time Since Onset** ($x_8$): Clipped exponential distribution, modeling disease progression timing.

- **Prior Intervention** ($x_9$): Binary indicator (0 or 1), representing whether previous treatments were administered.

These features are correlated to mimic real medical records: $x_1$ correlates with $x_0$ (risk depends on baseline status), $x_3$ correlates with $x_2$ (urgency relates to symptom complexity), $x_6$ correlates with $x_4$ (comorbidity relates to demographic group), and $x_8$ correlates with $x_5$ (disease progression relates to treatment response). This creates a rich, non-trivial context space that challenges learning algorithms.

## J.3. Decision Parameter Space: Precision Levels

We construct conformal prediction sets from a noisy linear nonconformity score. Recall that the true label is generated from the context-dependent categorical distribution

$$\mathbb{P}(y = j \mid x) = \frac{\exp(x^\top W_{:,j})}{\sum_{\ell=1}^{10} \exp(x^\top W_{:,\ell})}, \qquad j \in \{1, \dots, 10\}.$$

Thus, a larger linear logit $x^\top W_{:,\ell}$ indicates that label $\ell$ is more plausible under the data-generating model. To be consistent with the standard conformal convention that smaller nonconformity scores are more plausible, we define, for each candidate label $\ell \in \{1, \dots, 10\}$,

$$s(x, \ell) = -x^\top W_{:,\ell} + \epsilon, \qquad \epsilon \sim \mathcal{N}(0, \sigma_s^2), \qquad \sigma_s^2 = 0.01,$$

where the noise term $\epsilon$ is sampled independently for each score evaluation. This noisy negative-logit score preserves the ranking induced by the classifier while avoiding deterministic ties and creating a smoother conformal boundary.

The calibration set $\mathcal{D}_{\mathrm{cal}}$ consists of i.i.d. samples generated from the same distribution as the test contexts. For each calibration sample $(x_i, y_i)$, we compute the nonconformity score of the true label, $s(x_i, y_i)$. For each candidate significance level $\alpha$, let $\widehat{q}_{1-\alpha}$ denote the empirical $(1 - \alpha)$-quantile of these calibration scores. For a new context $x$, the conformal prediction set is then defined as

$$C_\alpha(x) = \left\{ \ell \in [10] : s(x, \ell) \leq \widehat{q}_{1-\alpha} \right\}.$$

If the set is empty, which rarely occurs because of the score noise, we include the label with the smallest nonconformity score:

$$C_\alpha(x) = \left\{ \arg\min_{\ell \in [10]} s(x, \ell) \right\}.$$

Equivalently, this fallback selects the label with the largest noisy linear logit.

In practice, we precompute $\widehat{q}_{1-\alpha}$ for all candidate significance levels

$$\mathcal{A} = \left\{ \alpha_k = 1 - \frac{k}{101} : k = 1, \ldots, 100 \right\}.$$

Larger values of $\alpha$ correspond to smaller quantile levels $1 - \alpha$, and therefore produce smaller and more precise prediction sets. Conversely, smaller values of $\alpha$ produce larger and more conservative prediction sets. This creates the intended coverage–precision trade-off: larger sets are safer but increase the human decision-maker's cognitive load, while smaller sets are easier to inspect but may exclude the true label.

## J.4. Human Decision Model: Expertise with Cognitive Load

In the general Human-AI collaboration framework, the human decision-maker's performance depends on both the AI's recommendation quality and their own cognitive constraints. Given that the true outcome $y$ is included in the set of recommendations, the conditional success (reward is 1) probability is modeled as:

- **Base Competence:** The human has a baseline success rate $\beta_0 = 0.8$ when the correct option is presented.

- **Cognitive Load Penalty:** Success probability decreases linearly with the size of the recommendation set. The penalty term is $\beta_1 \cdot \frac{|C_\alpha(x)|}{K}$, where $\beta_1 = -0.3$ (negative to reflect increased difficulty) and $K = 10$ is the total number of possible classes. The normalized factor $\frac{|C_\alpha(x)|}{K}$ represents the proportion of options presented.

- **Context-Specific Modulation:** Human expertise varies across contexts via a bounded nonlinear term $\Delta(x)$. For the 5-dimensional setting:

$$\begin{aligned}
\Delta(x) = {}& 0.3 \cdot \sin(2\pi x_0) \ + \ 0.25 \cdot \cos(3\pi x_1) \\
& + \ 0.2 \cdot (x_2 - 0.5)^2 \ + \ 0.15 \cdot \exp(-5(x_3 - 0.7)^2) \\
& + \ 0.1 \cdot (1 - 2|x_4 - 0.5|)
\end{aligned}$$

For the 10-dimensional setting, we extend $\Delta(x)$ to incorporate all features:

$$\begin{aligned}
\Delta(x) = {}& 0.12 \cdot \sin(2\pi x_0) \ + \ 0.10 \cdot \cos(3\pi x_1) \\
& + \ 0.09 \cdot (x_2 - 0.5)^2 \ + \ 0.08 \cdot \exp(-5(x_3 - 0.7)^2) \\
& + \ 0.07 \cdot (1 - 2|x_4 - 0.5|) \ + \ 0.06 \cdot \tanh(3x_5) \\
& + \ 0.05 \cdot \sin(4\pi x_6) \ + \ 0.04 \cdot (x_7 - 0.3)^3 \\
& + \ 0.03 \cdot \exp(-2(x_8 - 0.6)^2) \ + \ 0.02 \cdot (x_9 - 0.2)
\end{aligned}$$

Thus, when $y \in C_\alpha(x)$, the conditional success probability is:

$$p_{\text{success}}(x, \alpha \mid y \in C_\alpha(x)) = \text{clip}\left( \beta_0 + \beta_1 \cdot \frac{|C_\alpha(x)|}{K} + \Delta(x), \ 0, \ 1 \right)$$

where $\text{clip}(z, a, b) = \min(\max(z, a), b)$ ensures realistic bounds $[0, 1]$.

**Conditional Nature and Monotonicity:** Crucially, if $y \notin C_\alpha(x)$, the human cannot select the correct option and the reward is deterministically 0. This models the strict penalty in decision support systems when the optimal solution is not presented.

Therefore, in binary feedback setting, the expected reward for a given $(x, \alpha)$ is:

$$\mathbb{E}[r(x, \alpha)] = P(y \in C_\alpha(x) \mid x) \cdot p_{\text{success}}(x, \alpha \mid y \in C_\alpha(x))$$

More precisely, the reward distribution conditional on the coverage event is:

$$r(x, \alpha) \mid \{y \in C_\alpha(x)\} \sim \text{Bernoulli}\big( p_{\text{success}}(x, \alpha \mid y \in C_\alpha(x)) \big)$$

*Table 8.* Cumulative regret, $T = 1000$, 30 runs, 100 Arms, 5-dim Context

| Algorithm | Binary Feedback | | Continuous Feedback | |
|---|---|---|---|---|
| | Regret | Time (s) | Regret | Time (s) |
| AGCB-SE (ours) | **279.86** | 11.18 | **116.03** | 9.12 |
| SE (vanilla) | 301.54 | 10.21 | 127.04 | 9.05 |
| UCB | 303.68 | 10.13 | 132.91 | 11.66 |
| TS | 308.06 | 11.21 | 124.18 | 11.37 |
| $\epsilon$-greedy | 311.27 | 10.03 | 151.96 | 10.72 |
| LinUCB | 319.42 | 16.94 | 125.64 | 12.74 |
| Linear TS | 305.31 | 20.81 | 130.18 | 20.39 |
| LinSE | 289.83 | 11.67 | 126.58 | 13.28 |
| NeuralUCB | 325.12 | 21.14 | 175.68 | 29.31 |
| NeuralLinear | 306.92 | 31.74 | 137.42 | 33.81 |
| GP-UCB | 317.18 | 40.02 | 124.05 | 39.58 |
| BSE | 323.74 | 12.51 | 148.82 | 19.61 |
| ABSE | 312.45 | 12.18 | 172.57 | 12.66 |

while $r(x, \alpha) = 0$ when $y \notin C_\alpha(x)$. Here $\mathbb{I}(y \in C_\alpha(x) \mid x)$ is the coverage indicator, which equals 1 if and only if the true label $y$ is contained in the prediction set $C_\alpha(x)$.

Furthermore, in a more general setting with continuous feedback, the observed reward could be modeled as the expected reward plus zero-mean sub-Gaussian noise $\eta$:

$$r_{\text{observed}}(x, \alpha) = \mathbb{E}[r \mid x, \alpha] + \eta, \quad \eta \sim \text{subG}(\sigma^2)$$

We clip the noisy continuous reward to $[0, 1]$. This noise term captures the variability in human performance beyond the modeled success probability. Our algorithms are designed to handle both binary and continuous feedback structures.

The model satisfies our monotonicity assumption: for a fixed $x$, if $\alpha_1 > \alpha_2$, then $C_{\alpha_1}(x) \subseteq C_{\alpha_2}(x)$ and thus $|C_{\alpha_1}(x)| \leq |C_{\alpha_2}(x)|$. Since $\beta_1 < 0$, we have $p_{\text{success}}(x, \alpha_1 \mid y \in C_{\alpha_1}(x)) \geq p_{\text{success}}(x, \alpha_2 \mid y \in C_{\alpha_2}(x))$ whenever $y$ belongs to both sets. This creates the fundamental trade-off: higher $\alpha$ yields smaller sets with lower cognitive load (higher conditional success probability) but also lower coverage probability.

## J.5. Cumulative Regret Result and Analysis

While Section 6 focuses on simple regret, here we present the cumulative regret results in the same 100 or 10 arm, 5 or 10 dimension context setting.

We benchmark against a comprehensive set of algorithms, carefully selected to cover different methodological approaches: parametric, nonparametric, Bayesian, and frequentist. For cumulative regret, we compare UCB, Thompson Sampling (TS), $\varepsilon$-greedy, and Successive Elimination (SE) as non-contextual baselines; LinUCB (Li et al., 2010; Abbasi-Yadkori et al., 2011), Linear TS (Agrawal & Goyal, 2013), and LinSE (Linear Successive Elimination) as linear contextual methods; NeuralUCB (Zhou et al., 2020), NeuralLinear (feature extraction via representation learning for LinUCB), and GP-UCB (Grünewälder et al., 2010) as nonparametric approaches; and Binned Successive Elimination (BSE) and its adaptive variation ABSE (Perchet & Rigollet, 2013) as binned/spatial partitioning methods. Our proposed AGCB-SE method is compared against all these baselines.

Table 8, Table 9, and Table 10 report cumulative regret under three representative settings: many arms with moderate-dimensional contexts, fewer arms with moderate-dimensional contexts, and many arms with higher-dimensional contexts. Across these settings and under both binary and continuous feedback, AGCB-SE consistently delivers strong cumulative-regret performance while maintaining practical computational cost.

**Performance in the main many-arm setting.** In the primary setting with many candidate arms, AGCB-SE achieves the strongest overall performance among the compared methods. Its improvement over vanilla successive elimination suggests that adaptive grouping, counterfactual updates, and cross-group information sharing improve not only the quality of the final policy, as measured by simple regret, but also the efficiency of the online learning process. The comparison with non-contextual and contextual baselines further shows that exploiting both context heterogeneity and problem-specific structure is important in large-arm human–AI calibration tasks.

*Table 9.* Cumulative regret, $T = 1000$, 30 runs, 10 Arms, 5-dim Context

| Algorithm | Binary Feedback | | Continuous Feedback | |
|---|---|---|---|---|
| | **Regret** | **Time (s)** | **Regret** | **Time (s)** |
| AGCB-SE (ours) | **187.02** | 2.31 | **94.11** | 3.42 |
| SE (vanilla) | 204.46 | 2.15 | 107.37 | 3.26 |
| UCB | 216.34 | 4.05 | 102.48 | 3.31 |
| TS | 199.13 | 5.21 | 104.62 | 5.64 |
| $\epsilon$-greedy | 218.22 | 3.18 | 137.54 | 4.15 |
| LinUCB | 215.64 | 3.88 | 98.91 | 4.47 |
| Linear TS | 211.82 | 7.04 | 101.36 | 7.82 |
| LinSE | 190.79 | 4.36 | 95.54 | 5.63 |
| NeuralUCB | 217.11 | 8.93 | 142.36 | 10.28 |
| NeuralLinear | 195.74 | 10.61 | 104.48 | 9.77 |
| GP-UCB | 223.89 | 13.10 | 115.83 | 14.14 |
| BSE | 213.46 | 7.37 | 134.51 | 8.44 |
| ABSE | 209.31 | 3.22 | 161.49 | 3.16 |

*Table 10.* Cumulative regret, $T = 1000$, 30 runs, 100 Arms, 10-dim Context

| Algorithm | Binary Feedback | | Continuous Feedback | |
|---|---|---|---|---|
| | **Regret** | **Time (s)** | **Regret** | **Time (s)** |
| AGCB-SE (ours) | **301.37** | 11.18 | **142.83** | 11.24 |
| SE (vanilla) | 313.62 | 10.14 | 169.05 | 10.68 |
| UCB | 315.84 | 12.17 | 168.72 | 12.39 |
| TS | 324.68 | 13.38 | 185.51 | 13.09 |
| $\epsilon$-greedy | 318.47 | 12.75 | 172.46 | 12.61 |
| LinUCB | 326.28 | 29.48 | 167.24 | 33.21 |
| Linear TS | 327.59 | 39.15 | 164.71 | 42.55 |
| LinSE | 316.21 | 34.57 | 163.85 | 35.44 |
| NeuralUCB | 330.42 | 72.89 | 229.13 | 74.82 |
| NeuralLinear | 324.94 | 70.11 | 171.54 | 75.91 |
| GP-UCB | 319.91 | 53.42 | 191.58 | 41.91 |
| BSE | 346.32 | 50.88 | 225.47 | 51.72 |
| ABSE | 332.04 | 35.37 | 207.05 | 40.06 |

**Behavior with fewer arms.** When the number of arms is reduced, the learning problem becomes easier and several classical or linear baselines become more competitive. In this regime, AGCB-SE remains among the strongest methods, performing especially well under binary feedback and staying competitive under continuous feedback. This suggests that AGCB does not rely solely on the difficulty of the large-arm setting to obtain gains; rather, its grouping and sharing mechanisms remain useful even when the action space is relatively small. At the same time, the smaller-arm setting highlights that simple linear methods can be strong when the arm space is limited and the reward structure is favorable.

**Robustness in higher dimensions.** The higher-dimensional setting further illustrates the benefit of adaptive grouping. AGCB-SE remains relatively robust as the context dimension increases, whereas static or rigid partitioning methods become less effective due to data fragmentation. This supports the motivation behind the Zooming Mechanism: rather than refining all regions uniformly, AGCB splits only where the observed data indicate that finer resolution is useful. As a result, it avoids the severe sparsity issues associated with grid-style partitioning in moderately high-dimensional contexts.

**Comparison with other baselines.** Across the cumulative-regret experiments, neural and kernel-based methods often incur higher computational cost and can be less stable in the online, data-scarce regime. Classical non-contextual bandit methods ignore context heterogeneity, while linear contextual methods can perform well when their modeling assumptions are well aligned with the data but may be less reliable under more complex reward structures. AGCB-SE provides a favorable balance: it remains computationally lightweight, captures heterogeneous context structure through adaptive grouping, and leverages counterfactual information to improve sample efficiency over its vanilla SE backbone.

Overall, these cumulative-regret results complement the simple-regret experiments and confirm that AGCB-SE is effective not only for identifying a high-quality final policy but also for reducing regret during online learning. The results support

NeuralUCB: Parameter and Loss Dynamics

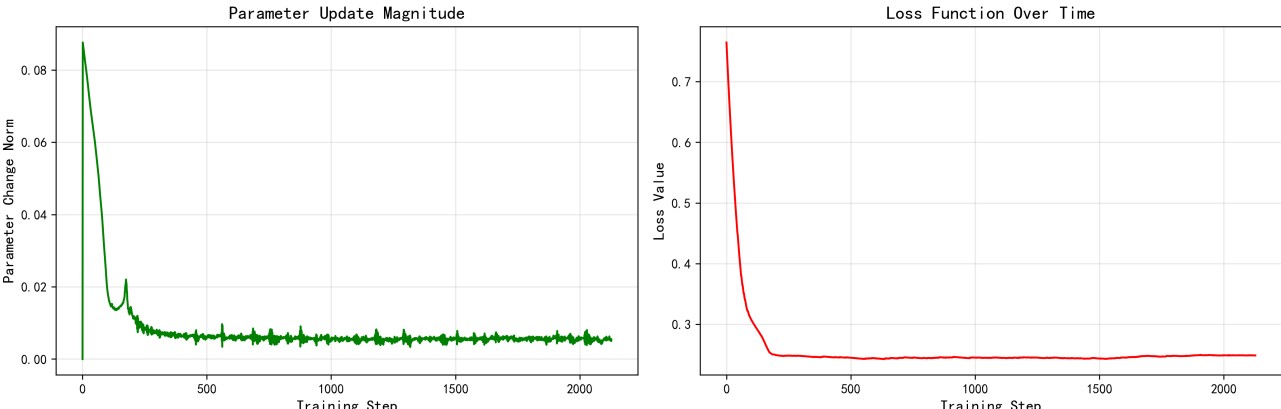

*Figure 1.* In this figure, Neural UCB demonstrates normal convergence with decreasing loss, confirming that the neural network learning method is at least effective in terms of prediction.

the central design principle of AGCB: combining adaptive context partitioning, cross-group information sharing, and monotonicity-based counterfactual updates yields a scalable and robust approach for human–AI decision-support calibration.

## J.6. Online Training Challenges of Complex Function Approximators

This section examines why complex function approximators (e.g., neural networks) often struggle in online human–AI settings. Theoretically, their regret bounds contain large complexity constants (e.g., effective dimension or maximum information gain), and even when such bounds are minimax-optimal, the hidden constants can make the algorithms perform poorly for practical horizons. Empirically, we illustrate this issue using neural-based bandit methods, NeuralUCB and NeuralLinear, in our 100-arm, 5-dimensional context, $T = 1000$ cumulative-regret setting with binary feedback. Without sufficient data, their online training is unstable and leads to high regret, whereas AGCB's structure-aware design avoids these pitfalls and learns reliably from scarce interactions.

### J.6.1. NEURAL METHODS IN ONLINE CONTEXTUAL BANDITS

NeuralUCB employs a neural network that takes as input the concatenated context and one-hot encoded action, directly predicting the expected reward. The algorithm selects actions based on an upper confidence bound that combines the neural network's prediction with a heuristic uncertainty estimate, typically derived from properties of the network's gradients or ensemble variations.

In our experiments, NeuralLinear adopts a two-stage architecture: a neural network feature extractor first transforms the context-action pair into a latent representation, followed by a linear model that operates on these features to predict rewards. The upper confidence bound calculation employs the linear model's analytical confidence intervals. This architecture theoretically enables decoupled learning, the neural component learns nonlinear feature representations while the linear layer rapidly adapts to the reward structure through closed-form updates.

### J.6.2. TRAINING INSTABILITIES IN ONLINE LEARNING

As shown in Figure 1 and 2, NeuralUCB demonstrated a peculiar decoupling between loss reduction and gradient behavior. While the mean squared error loss consistently decreased and stabilized, indicating improved predictive accuracy, the gradient statistics failed to converge. The gradient norm remained persistently elevated, and the cosine similarity between successive gradient directions fluctuated erratically. This phenomenon suggests that the neural network is perpetually adapting to a non-stationary target, where each new context-action-reward triplet defines a distinct optimization objective. The network's parameters thus chase a moving target rather than converging to a stable optimum. Since the exploration bonus in NeuralUCB is constructed from these very gradients, their instability critically undermines the algorithm's online performance.

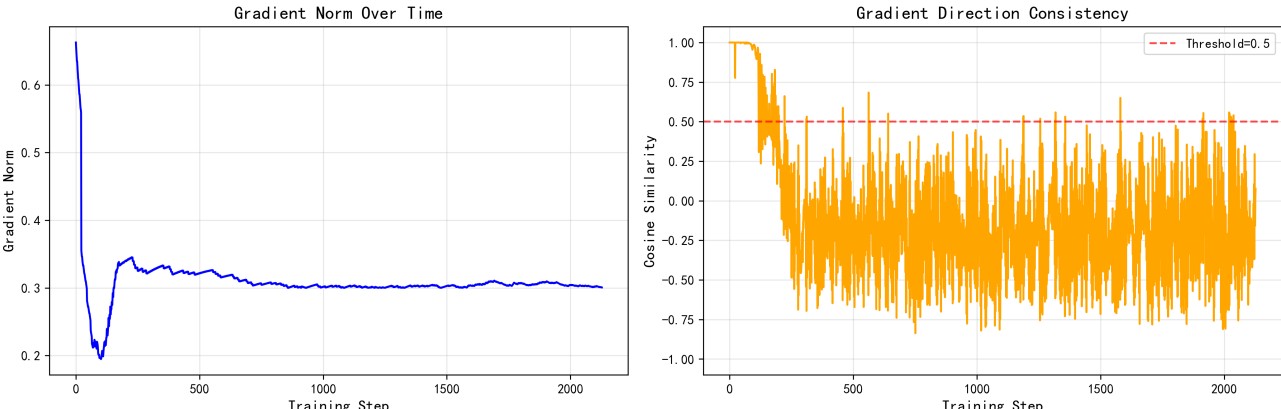

*Figure 2.* In this figure, the gradient norm of Neural UCB (left) initially decreases but then rises during online training, subsequently fluctuating persistently within a certain range without further reduction. Meanwhile, the cosine similarity of gradient directions for Neural UCB (right) remains consistent only in the early stages and fluctuates dramatically for most of the training period.

As illustrated in Figure 3, NeuralLinear exhibits gradient non-convergence issues similar to those observed in Neural UCB, with its gradient norm showing even greater fluctuations. However, its gradient directions tend to be more consistent on average. Overall, using these two neural methods as examples, the figure illustrates the training challenges they face in online learning scenarios, where sufficient data must gradually accumulate to effectively learn the complex functional relationships underlying human decisions. This also highlights the distinct advantage of our non-parametric adaptive grouping method. In the emerging era of Human-AI collaboration, our approach offers a practical and effective solution.

In the online setting, neural bandit methods suffer from unstable optimization and poor sample efficiency, as their global function approximators must continuously chase a non-stationary target, leading to non-convergent gradients and unreliable uncertainty estimates. Our AGCB framework overcomes these limitations through adaptive local partitioning, which circumvents the moving-target problem by maintaining stable, group-wise statistics. The Zooming Mechanism explicitly balances bias and variance, refining partitions only where statistically justified, while built-in inductive biases, monotonicity and continuity, enable sample-efficient learning through counterfactual updates and similarity-weighted information sharing. This results in a principled, interpretable, and data-efficient learner that is particularly suited for the emerging domain of Human-AI collaboration.

### J.7. Robustness Studies: Violation of Assumption

Our monotonicity assumption (Assumption 4.2) states that, conditional on coverage ($y \in C_\alpha(x)$), a more precise prediction set (larger $\alpha$) leads to no worse decision quality. While this assumption is empirically supported by large-scale human studies (Vishwakarma et al., 2025; Straitouri & Gomez Rodriguez, 2024), we investigate the robustness of AGCB when this assumption is deliberately violated.

**Why not a similar robustness analysis for Hölder continuity?** The continuity assumption (Assumption 4.1) plays a fundamentally different role: it ensures that similar contexts and $\alpha$ values yield similar rewards, which is required for the Zooming Mechanism to safely share information across groups and perform counterfactual updates. Unlike monotonicity, continuity is a mild condition widely observed in naturalistic decision-making; recent neuroscience research (Yoo et al., 2021) demonstrates that humans naturally make decisions over continuous option spaces, where nearby choices elicit smooth, gradual changes in response strategies, providing empirical grounding for the continuity assumption. Therefore, continuity is not a directional constraint that can be easily "flipped" in experiments, and it has direct empirical grounding in cognitive science. Moreover, our real-human experiments (Section 6.2)—where rewards come from actual human responses without any built-in smoothness guarantees—provide an empirical stress test under realistic deviations from the idealized continuity condition. The strong performance of AGCB on real data (Tables 4 and 5) indicates that moderate departures from the idealized continuity condition do not appear to prevent strong empirical performance. Hence we focus our robustness

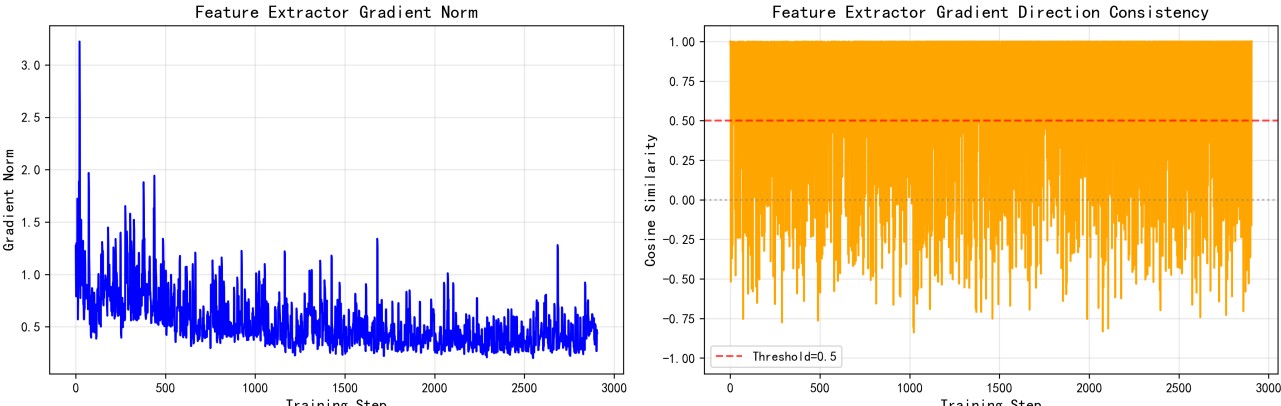

*Figure 3.* In this figure, the NeuralLinear method exhibits behavior similar to Neural UCB. On the left, the gradient of the representation learning network first decreases and then shows persistent, high-amplitude fluctuations (with gradient norm variations larger than those of Neural UCB). On the right, the cosine similarity of gradient directions fluctuates substantially throughout almost the entire process. However, on average, the gradient cosine similarity of the NeuralLinear method is higher compared to that of Neural UCB.

*Table 11.* Simple regret under monotonicity violation ($p = 0.5$). Setting: $K = 100$, $d = 5$, $T = 1000$, 30 runs.

| Algorithm | Binary Feedback | | Continuous Feedback | |
|---|---|---|---|---|
| | Regret | Time (s) | Regret | Time (s) |
| **AGCB-SH (ours)** | **0.352** | **4.92** | **0.131** | **5.03** |
| SH (vanilla) | 0.558 | 3.44 | 0.192 | 3.67 |
| Contextual-Gap | 0.523 | 12.15 | 0.174 | 11.89 |
| BOED | 0.394 | 14.03 | 0.180 | 13.96 |
| TTTS-C | 0.386 | 13.04 | 0.204 | 13.98 |
| LinUCB-PE | 0.532 | 9.30 | 0.146 | 10.99 |

study on monotonicity, which is more prescriptive and therefore more critical to validate empirically.

**Experimental setup for monotonicity violation.** We take the synthetic setting with $K = 100$ arms, $d = 5$ context dimension, and horizon $T = 1000$. To deliberately violate monotonicity, we randomly flip the direction of the monotonicity relation with probability $p = 0.5$: when coverage occurs, larger $\alpha$ (more precise sets) yields *lower* reward with probability $p$, and higher reward with probability $1 - p$ (the original direction). All other experimental conditions remain identical to those in Section 6.1. We report simple regret averaged over 30 runs.

**Results.** Table 11 shows the simple regret under this violation. Compared to the original setting where AGCB-SH achieves 0.298 (binary) and 0.111 (continuous), its regret increases to 0.352 and 0.131 respectively. However, compared to the original setting, AGCB-SH experiences only a moderate increase in regret. Under binary feedback, most baselines degrade more substantially; under continuous feedback, the effects vary across baselines, but AGCB-SH still achieves the lowest regret.

**Why is AGCB robust?** Two design choices contribute to graceful degradation under assumption violations:

1. **Soft counterfactual updates**: The Gaussian kernel $w(\alpha_t, \alpha')$ decays with distance, so updates from an arm that contradicts the monotonicity direction have limited influence on distant arms.

2. **Cross-group information sharing**: Sharing across similar groups dilutes the impact of any single erroneous update, as estimates are pooled over multiple groups.

These mechanisms help AGCB remain the best-performing method in this stress test, even under substantial monotonicity

*Table 12.* Ablation study: Average simple regret and run-time of AGCB-SH variants under binary and continuous feedback ($K = 100, d = 5, T = 1000$, averaged over 30 runs). The full configuration (T-T-T) achieves the lowest regret, validating the synergistic design. The results show that the Zooming Mechanism is most effective when combined with counterfactual updates and cross-group sharing. Numbers in parentheses indicate performance rank (lower regret is better).

| | Components | | | Binary Feedback | | Continuous Feedback | |
| --- | --- | --- | --- | --- | --- | --- | --- |
| **Variant** | **CF** | **Zoom** | **Share** | **Regret (Rank)** | **Time(s)** | **Regret (Rank)** | **Time(s)** |
| T-T-T | ✓ | ✓ | ✓ | **0.298 (1)** | 4.11 | **0.111 (1)** | 4.36 |
| F-T-F | – | ✓ | – | 0.300 (2) | 1.11 | 0.138 (5) | 1.30 |
| F-T-T | – | ✓ | ✓ | 0.306 (3) | 1.18 | 0.130 (4) | 1.31 |
| T-F-F | ✓ | – | – | 0.308 (4) | 4.23 | 0.141 (6) | 4.42 |
| T-T-F | ✓ | ✓ | – | 0.311 (5) | 4.22 | 0.117 (2) | 4.47 |
| F-F-T | – | – | ✓ | 0.351 (6) | 1.02 | 0.119 (3) | 1.08 |
| T-F-T | ✓ | – | ✓ | 0.367 (7) | 5.33 | 0.143 (7) | 5.19 |
| F-F-F | – | – | – | 0.381 (8) | 1.06 | 0.159 (8) | 1.25 |

violations.

**Real-human data as an implicit robustness test.** Crucially, our real-human experiments (Section 6.2) do not enforce either monotonicity or Hölder continuity by construction; rewards are collected from actual human participants without any artificial smoothing or directional guarantees. Since these real-world responses naturally contain noise and potential violations of both assumptions, the fact that AGCB variants achieve the lowest regret in all settings (Tables 4 and 5) provides empirical evidence that AGCB remains effective under realistic deviations from the ideal theoretical conditions.

## J.8. Ablation Studies: Validating Algorithmic Components

To validate the contribution of each component and verify that our algorithm's empirical behavior aligns with its theoretical underpinnings, we conduct a comprehensive ablation study. We evaluate all $2^3 = 8$ combinations of our three core components under the simple regret and cumulative regret setting ($K = 100, d = 5, T = 1000$) for both binary and continuous feedback. The average final regret and run-time are summarized in Table 12 and Table 13.

**Design of Ablation Variants.** Each variant modifies the corresponding full AGCB algorithm, namely AGCB-SH for simple regret and AGCB-SE for cumulative regret, as follows:

- **Counterfactual Update (CF):** When disabled (False), the algorithm performs standard single-point updates, revising the estimate only for the selected arm.

- **Adaptive Zooming Mechanism (Zoom):** When disabled (False), the theoretically-derived Zooming Mechanism is replaced by a simple heuristic: a new group is created if a new context is farther than a fixed threshold from all existing group centers. This static rule lacks the dynamic bias-variance trade-off central to our design.

- **Cross-Group Sharing (Share):** When disabled (False), groups learn in isolation without information transfer.

## J.9. Analysis of Ablation Results

The ablation results in Table 12 and Table 13 validate the design choices of AGCB under both simple and cumulative regret. Overall, the full configuration (T-T-T) consistently achieves the best regret performance across feedback types and regret metrics, showing that the three components are most effective when used jointly.

**Role of adaptive zooming.** The Zooming Mechanism is a central component of AGCB. Variants with Zoom=T generally occupy the strongest positions across the ablation tables, especially in cumulative regret, where adaptive refinement is crucial for maintaining learning efficiency over time. This supports the theoretical role of zooming: it adaptively balances local approximation bias and statistical uncertainty, refining heterogeneous regions while avoiding unnecessary fragmentation in smoother or data-rich regions. At the same time, the results also show that zooming interacts with the other components rather than acting in isolation; information sharing and counterfactual updates can provide additional gains when combined with a good adaptive partition.

*Table 13.* Ablation study: Average cumulative regret and run-time of AGCB-SE variants under binary and continuous feedback ($K = 100, d = 5, T = 1000$, averaged over 30 runs). The full configuration (T-T-T) achieves the lowest regret, validating the synergistic design. The results show that the Zooming Mechanism is most effective when combined with counterfactual updates and cross-group sharing. Numbers in parentheses indicate performance rank (lower regret is better).

| | Components | | | Binary Feedback | | Continuous Feedback | |
|---|---|---|---|---|---|---|---|
| **Variant** | **CF** | **Zoom** | **Share** | **Regret (Rank)** | **Time(s)** | **Regret (Rank)** | **Time(s)** |
| T-T-T | ✓ | ✓ | ✓ | **279.86 (1)** | 11.18 | **116.03 (1)** | 9.12 |
| T-T-F | ✓ | ✓ | – | 282.41 (2) | 10.37 | 123.94 (2) | 8.87 |
| F-T-F | – | ✓ | – | 287.15 (3) | 9.18 | 129.27 (5) | 7.93 |
| F-T-T | – | ✓ | ✓ | 295.36 (4) | 8.57 | 125.12 (3) | 8.07 |
| F-F-T | – | – | ✓ | 298.94 (5) | 10.11 | 141.86 (8) | 7.82 |
| F-F-F | – | – | – | 301.42 (6) | 14.53 | 127.18 (4) | 9.04 |
| T-F-F | ✓ | – | – | 304.97 (7) | 10.93 | 131.08 (6) | 8.34 |
| T-F-T | ✓ | – | ✓ | 310.28 (8) | 15.41 | 136.21 (7) | 9.48 |

**Synergy of the full configuration.** The complete variant T-T-T achieves the best overall regret performance in both tables. This indicates that counterfactual updates, adaptive zooming, and cross-group information sharing are complementary. Zooming provides the context partition on which the other two mechanisms operate, counterfactual updates improve sample efficiency across the arm space, and sharing transfers information across similar context groups. Variants that remove zooming while keeping other components are less stable, suggesting that counterfactual and sharing mechanisms are most reliable when built on a high-quality adaptive grouping structure.

**Counterfactual updates and information sharing.** The effects of CF and Share are context-dependent. Counterfactual updates are designed to reduce the effective arm-space complexity by propagating information across ordered arms, while information sharing improves data efficiency by borrowing strength across nearby groups. The ablation results show that each component can help in some settings, but neither consistently matches the full configuration alone. In particular, sharing without adaptive zooming can be useful when nearby groups are informative, but it may also be less reliable when the partition is too coarse or poorly aligned with the reward heterogeneity. This reinforces the need for the joint design.

**Computational trade-off.** The run-time varies across ablation variants because different component combinations induce different partition and update dynamics. Although richer variants may require additional computation for adaptive partitions, counterfactual updates, and cross-group pooling, the full configuration remains computationally practical and achieves the best regret. This indicates a favorable trade-off between statistical efficiency and computational cost.

**Conclusion.** The ablation study provides a coherent empirical picture: adaptive zooming is the backbone of AGCB, while counterfactual updates and information sharing serve as complementary mechanisms that improve sample efficiency when coupled with the adaptive partition. The superior performance of T-T-T across both simple and cumulative regret confirms the importance of the co-designed framework.

### J.10. Parameter Sensitivity Analysis

This section demonstrates that AGCB is robust to its core hyperparameters in the Zooming Mechanism (Eq. (3)): the constants $C$ and $\eta$, the smoothness constant $L$, and the zooming exponent $\beta$ (denoted by $\tilde{\beta}$ in this sensitivity analysis). In practice, precise knowledge of these parameters is often unavailable; we show that AGCB remains effective as long as they lie within a reasonable range.

We conduct the parameter sensitivity analysis in both the simple regret and cumulative regret settings with $K = 100$ arms, $d = 5$ context dimension, and horizon $T = 1000$. The analysis is performed under both binary and continuous feedback. We systematically vary each Zooming-Condition parameter while fixing the others at their nominal values used in the main experiments ($C = 0.1$, $\eta = 10.0$, $L = 2.0$, and $\tilde{\beta} = 1.0$). Here, in this sensitivity experiment, $\tilde{\beta}$ denotes the exponent used by the algorithm in the Zooming rule; it is a tuning parameter and need not be interpreted as the true Hölder exponent in Assumption 4.1. For each parameter setting, we run 30 independent trials and report the average regret.

Tables 14 and 15 summarize the parameter sensitivity results for simple and cumulative regret, respectively. Each row varies one parameter over the displayed range while keeping the others fixed at their default values, shown in bold. For each feedback type, the reported *Relative Change* measures the largest regret deviation from the default configuration across the

*Table 14.* Parameter sensitivity analysis: Simple regret under varying Zooming-Condition parameters ($K = 100$, $d = 5$, $T = 1000$). For each row, we test the listed values for one parameter while holding the other parameters at their default values shown in bold. Results (relative change vs. the default) are shown for both binary and continuous reward feedback settings; each cell reports the maximum deviation observed across the tested range. Values are averages over 30 runs.

| Parameter | Tested Values | Relative Change (Binary) | Relative Change (Continuous) |
|---|---|---|---|
| $C$ (constant) | 0.01, 0.05, **0.1**, 0.5, 1.0 | $\leq 0.2\%$ | $\leq 2.8\%$ |
| $\eta$ (balance factor) | 1.0, 5.0, **10.0**, 20.0, 50.0 | $\leq 0.2\%$ | $\leq 2.8\%$ |
| $L$ (smoothness constant) | 0.5, 1.0, **2.0**, 5.0, 10.0 | $\leq 0.0\%$ | $\leq 0.0\%$ |
| $\tilde{\beta}$ (Zooming exponent) | 0.1, 0.5, **1.0**, 1.25, 1.5 | $\leq 1.4\%$ | $\leq 3.5\%$ |

*Table 15.* Parameter sensitivity analysis: Cumulative regret under varying Zooming-Condition parameters ($K = 100$, $d = 5$, $T = 1000$). For each row, we test the listed values for one parameter while holding the others at their default values shown in bold. Each cell reports the maximum relative change in regret compared with the default configuration across the tested range. Values are averages over 30 runs.

| Parameter | Tested Values | Relative Change (Binary) | Relative Change (Continuous) |
|---|---|---|---|
| $C$ (constant) | 0.01, 0.05, **0.1**, 0.5, 1.0 | $\leq 3.6\%$ | $\leq 8.2\%$ |
| $\eta$ (balance factor) | 1.0, 5.0, **10.0**, 20.0, 50.0 | $\leq 3.0\%$ | $\leq 7.5\%$ |
| $L$ (smoothness constant) | 0.5, 1.0, **2.0**, 5.0, 10.0 | $\leq 1.4\%$ | $\leq 2.0\%$ |
| $\tilde{\beta}$ (Zooming exponent) | 0.1, 0.5, **1.0**, 1.25, 1.5 | $\leq 4.2\%$ | $\leq 9.3\%$ |

tested range.

**Robustness of final policy quality.** For simple regret, Table 14 shows that AGCB-SH is highly stable across the tested parameter ranges. Varying a single parameter over a wide range leads to only small relative changes in the final policy regret. This suggests that the final learned policy is not overly sensitive to precise choices of the Zooming-Condition parameters. Empirically, this stability is consistent with the role of the Zooming Mechanism: the split rule adapts to the observed data by balancing local statistical uncertainty and approximation bias, so moderate parameter changes mainly shift the splitting threshold rather than fundamentally changing the learned policy.

**Sensitivity of the online learning process.** Cumulative regret, reported in Table 15, is somewhat more sensitive to parameter variation. This is expected because cumulative regret accumulates exploration costs throughout the learning horizon. Parameters that affect the timing or granularity of splits, such as $C$, $\eta$, and the zooming exponent $\tilde{\beta}$, can influence early exploration decisions, and these choices accumulate over time. Nevertheless, the observed variation remains moderate across the tested ranges, indicating that AGCB-SE retains stable online performance without delicate tuning.

**Practical implications.** These results suggest that AGCB does not require highly precise tuning of $C$, $\eta$, $L$, and the zooming exponent $\tilde{\beta}$ in the tested regimes. Reasonable parameter choices are sufficient to obtain stable empirical performance, especially for simple regret. This empirical robustness is complementary to the theoretical discussion of unknown smoothness in Appendix H and Appendix I: fixed misspecification of the true smoothness exponent can affect asymptotic rates in principle, but in the finite-sample regimes tested here, AGCB remains practically stable across a broad range of Zooming-Condition parameters.

