# OpenReview forum: "Adaptively Grouped Contextual Bandits for Heterogeneous Human-AI Decision Making with Conformal Prediction Sets"
_ICML.cc/2026/Conference — ICML 2026 regular_

### Official Review · Reviewer_nDWR · 2026-02-28

**Soundness:** 3
**Presentation:** 3
**Significance:** 3
**Originality:** 3
**Overall Recommendation:** 4
**Confidence:** 4

**Summary:**

This paper focus on a scenario where an AI system uses Conformal Prediction  to provide a human decision-maker with a set of candidate labels, the size of which is controlled by a significance parameter $\alpha$. The human then makes the final choice from this set. The paper proposes the AGCB framework. Under continuity and monotonicity assumptions, theoretical analysis shows the algorithm achieves minimax-optimal regret rates, and empirical results demonstrate its efficiency and robustness compared to standard baselines.

**Compliance With Llm Reviewing Policy:**

Affirmed.

**Final Justification:**

I maintain my weak accept recommendation. The paper is technically solid and original, with a well-motivated connection between conformal prediction and contextual bandits, strong regret analysis, and a practical lightweight framework. Although I initially had some concerns, the rebuttal adequately addressed my core concerns.

**Key Questions For Authors:**

Questions:

Q1. Could the authors explain why the BOED baseline outperforms AGCB in simple regret in Table 2? Does this imply that when the action space is small enough to allow for exact Bayesian methods, AGCB's zooming/counterfactual mechanisms are inherently suboptimal or too coarse?

Q2. Can the current theoretical framework be extended or relaxed to accommodate like piece-wise continuous property? Or is there any experimental evidence to justify your two assumptions?

Q3. Given the exponential dependence on $d$ in the theoretical bounds, how can the AGCB framework be effectively scaled to high-dimensional tasks (e.g., $d > 50$)?

**Limitations:**

Yes

**Strengths And Weaknesses:**

Strengths:

S1. The paper provides a well-motivated bridge between Conformal Prediction and Contextual Bandits. By treating the CP significance level $\alpha$ as a bandit arm, it offers a principled way to balance AI precision and human cognitive load.

S2.  The authors provide a rigorous regret analysis. The proof is elegant and extends existing metric bandit theory in a non-trivial way.

S3. The AGCB framework is computationally lightweight, which makes it practical for real-time human-AI interaction.

Weaknesses:

W1. Table 2 reveals that the BOED baseline actually achieves strictly lower simple regret than AGCB-SH in both Binary (0.190 vs. 0.193) and Continuous (0.071 vs. 0.086) feedback settings, but there is no discussion of the algorithm's suboptimality in smaller arm spaces.

W2. The upper bound proof heavily benefits from the monotonicity assumption (Assumption 4.2), which allows the algorithm to perform counterfactual updates akin to a "binary search," exponentially reducing the dependence on the action space. However, the theoretical lower bound used to claim "minimax optimality" is directly cited from standard metric space bandits, which do not assume monotonicity. By introducing strong structural information (monotonicity), the lower bound of the problem should naturally be lower. Matching an algorithm empowered by a strong assumption against a lower bound derived for a weak assumption, and claiming it is "optimal," is theoretically imprecise.

W3. The paper relies heavily on Assumption 4.1, which states that the expected reward is Hölder continuous with respect to the context and $\alpha$. In practical conformal prediction, however, threshold effect may trigger non-continuous phase transitions in human cognitive load and response. Applying a smooth continuity assumption here may diverge from the physical reality of the task.

---

> ### Author Rebuttal · Authors · 2026-03-29
>
> Thanks for your insightful feedback.
>
> ### Q1 & W1. Simple Setting Results Interpretation
> In the 10‑arm, 5‑dim setting (Table 2), BOED achieves slightly lower simple regret than AGCB‑SH (binary: 0.190 vs. 0.193; continuous: 0.071 vs. 0.086). This is expected: BOED is a Bayesian pure exploration method that can effectively explore the entire arm space when the number of arms is small. AGCB, by contrast, is designed for large arm spaces and high‑dimensional contexts. Its adaptive grouping and counterfactual mechanisms introduce some overhead in simple regimes but deliver substantial gains in challenging scenarios (100 arms, 5‑10 dims, Tables 1 and 3). Thus, the small gap in the 10‑arm case reflects a natural trade‑off; no single method dominates all settings. We will add a brief discussion to acknowledge this.
>
> ### W2. Theoretical “minimax optimal” Claim
> For nonparametric contextual bandits under Hölder continuity, the minimax lower bounds are:
>
> - Simple regret: $\Omega\left(K^{\frac{\beta}{2\beta+d}} T^{-\frac{\beta}{2\beta+d}}\right)$
> - Cumulative regret: $\Omega\left(K^{\frac{\beta}{2\beta+d}} T^{\frac{\beta+d}{2\beta+d}}\right)$
>
> AGCB framework achieves the following upper bounds:
>
> - Simple regret: $O\bigl((\log K)^{\frac{\beta}{2\beta+d}}\ T^{-\frac{\beta}{2\beta+d}}\bigr)$
> - Cumulative regret: $O\bigl((\log K)^{\frac{\beta}{2\beta+d}}\ T^{\frac{\beta+d}{2\beta+d}}\bigr)$
>
> In terms of the horizon $T$, which is the central parameter in bandit theory, our regret rates are minimax optimal — this is the sense in which we claim optimality in the paper when discussing the rate with respect to $T$. Regarding the number of arms $K$, the standard minimax lower bound scales polynomially in $K$, while under the monotonicity, our algorithm achieves a logarithmic dependence. Thus, the improved $K$-dependence reflects the benefit of monotonicity, other aspects align with the standard rates. We will provide a more detailed clarification in the revised manuscript.
>
> ### W3 & Q2. Assumption and Robustness
> We address it from theoretical, experimental and literature perspectives.
>
> First, continuity in the context $x$ is a standard assumption in contextual bandits. For example, in our additional real‑human experiments (details in Reviewer BQa9), contexts includes image features, decision-maker and decision timing. Small changes in an image or decision time do not cause abrupt jumps in decision quality; the reward varies gradually. Thus, continuity in $x$ is natural and well‑justified.
>
> Second, for $\alpha$, in our problem setting, we state that $\alpha$ values are precisely the empirical quantiles of the calibration set — a finite discrete set $\mathcal{A}$. Therefore, we are not concerned with arbitrarily close discontinuous points; the assumption only requires that the inequality holds at these discrete points, ensuring that the change in the reward function between different $\alpha$ values is controlled. It is a mild requirement and aligns naturally with the structure of conformal prediction for human‑AI decision support.
>
> Third, our method is robust in experiments. In our additional real‑human data experiment, which is built on classification tasks where the prediction sets are discrete sets of labels, the reward comes from actual human behavior—no continuity and monotonicity assumptions are used to generate the data. AGCB still achieves the lowest regret. These results demonstrate that our algorithm does not rely on perfect continuity.
>
> For monotonicity, we also tested robustness by reversing the monotonicity direction with probability $p$; for $p \le 0.5$, AGCB remains best (details in Reviewer BQa9). This robustness stems from soft updates (Eq. 4) and cross‑group information sharing, which limit the impact of misleading observations.
>
> Finally, literature support: the literature provides empirical support: references [1,2] document continuity in human decision‑making, while [3,4] validate monotonicity of decision quality with respect to set size.
>
> [1] Continuous decisions
>
> [2] Naturalistic decision-making: continuous, open-world, and recursive
>
> [3] Designing decision support systems using counterfactual prediction sets
>
> [4] Prune’n predict:Optimizing llm decision-making with conformal prediction
>
> ### Q3. Scaling to High Dimensions
> Theoretically, the regret rates do not suffer from a severe exponential curse of dimensionality (the exponent depends on $d$ either only in the denominator or in both numerator and denominator). Practically, our real‑human data experiment uses 21‑dim contexts. Traditional grouping methods like BSE/ABSE would require at least $2^{21}$ groups in a single split—infeasible. AGCB’s adaptive binary splitting splits one dimension at a time, so groups grow linearly with splits, not exponentially. It runs efficiently and achieves the lowest regret, while BSE/ABSE cannot execute. It demonstrates practical mitigation of the curse of dimensionality.
>
> Thanks again. We hope these address your concerns.

---

> > ### Author Rebuttal · Reviewer_nDWR · 2026-04-03
> >
> > Thank you for the rebuttal. I will maintain my score.

---

> > > ### Author Response · Authors · 2026-04-03
> > >
> > > Thank you for acknowledging that our responses have addressed your main concerns. We are glad to hear that. Please let us know if any further questions remain.

---

### Official Review · Reviewer_1Kqp · 2026-03-08

**Soundness:** 3
**Presentation:** 3
**Significance:** 4
**Originality:** 3
**Overall Recommendation:** 5
**Confidence:** 3

**Summary:**

The paper is motivated by a personalized human-AI decision-support problem. Instead of having AI make the final choice, the AI provides a conformal prediction set, a set of plausible options, and the human chooses from that set. The paper focuses on the question on how to choose the conformal parameter $\alpha$, which controls the size of the set. Smaller sets may improve clarity and reduce cognitive burden, but they also risk excluding the correct option; larger sets are safer but less decisive. The authors argue that the optimal $\alpha$ should vary across users and tasks because humans are heterogeneous in expertise, risk tolerance, and decision behavior. This leads them to study online personalization of conformal set size in a contextual bandit framework.

Their model treats each candidate $\alpha$ as an arm and each human-task instance as a context. Given a context, the system selects $\alpha$, constructs a conformal prediction set, the human makes a decision based on that set, and the system observes reward feedback. The main contribution is a AGCB framework proposed in the paper, which adaptively groups similar contexts, runs simpler local bandit learners within groups, shares information across nearby groups, and uses counterfactual updates across arms by exploiting continuity and monotonicity structure in the human–AI problem. Theoretical results claim regret rates and improved dependence on the number of arms via these counterfactual updates. Empirically, in simulated heterogeneous decision environments with binary and continuous feedback, AGCB outperforms strong baselines, especially when the number of candidate $\alpha$ values is large and contexts are high-dimensional.

**Compliance With Llm Reviewing Policy:**

Affirmed.

**Key Questions For Authors:**

1. I may miss something but the paper notes that under Assumption 4.2, the unconditional reward need not be monotone in $\alpha$, yet the algorithm eliminates half of the arm space at a time, which typically requires a stronger global structure such as monotonicity or unimodality. It is therefore unclear why Assumption 4.2 alone is sufficient to support this elimination step.
2. Are there typos in Eqns (4)-(6) for the notation $\alpha$? It is a bit difficult to understand what are updated here.
3. It is a bit confused to use both $\mu$ and $\mu^*$ as the expected reward function.
4. In Algo 2, when $U=L+1$, the midpoint becomes $M=L$; if the algorithm then sets $L\gets M$ the interval does not shrink, so the procedure may fail to terminate. This seems to require a correction such as $L \gets M+1$ or a different stopping/update rule?
5. In section 3, there is “a pre-trained learner $f$”, but it does not seem to play an explicit role later in Sections. If so, I would suggest not introduce this notation.

**Strengths And Weaknesses:**

The paper studies a timely and important problem, how to personalize conformal prediction sets in human–AI decision support, and the paper proposes a methodologically interesting framework that combines adaptive grouping, information sharing, and counterfactual updates to improve learning efficiency in heterogeneous settings. Its theoretical analysis is comprehensive, covering both simple and cumulative regret, and the simulation results are encouraging, especially in large-arm and high-dimensional environments.  I do not have major concerns about the paper.

---

> ### Author Rebuttal · Authors · 2026-03-28
>
> Thanks for your thorough and valuable review.
>
> ### Q1. Why Assumption 4.2 Supports Eliminating Half the Arm Space
> This insightful question highlights a core challenge of context heterogeneity: while Assumption 4.2 ensures conditional monotonicity for each $x$, the group-averaged reward $\mu_g(\alpha)$ may not be monotone. How can we safely eliminate half the arms?
>
> The answer lies in the Zooming Mechanism. It partitions the context space so that each group is sufficiently homogeneous—the group diameter $d_g$ is balanced such that the approximation bias $L(d_g)^\beta$ is of the same order as the statistical estimation error (Lemma G.4). Under this balance, any deviation from perfect monotonicity caused by group heterogeneity is bounded by $O((d_g)^\beta)$, which is no larger than the statistical uncertainty. Moreover, thanks to the continuity assumption on the reward function (Assumption 4.1), the Zooming Mechanism can refine the partition adaptively. In experiments, our AGCB rapidly splits high-variance groups, ensuring that within each group a single $\alpha$ quickly becomes representative. Importantly, the error during the early, coarse-grouping phase is theoretically controlled: any bias or suboptimal elimination caused by early non-monotonicity is bounded by the group's diameter. This rapid adaptation ensures early-stage errors contribute only a lower-order term to the cumulative regret.
>
> Consequently, while $\mu_g(\alpha)$ may lack strict monotonicity, the risk of eliminating the optimal arm is safely controlled. The Zooming Mechanism ensures per-sample monotonicity holds approximately across the group, absorbing this error into the bias-variance trade-off. In Appendix G.6 we show that this structured bias does not affect the minimax‑optimal regret rate. In short, our AGCB ensures that the elimination step within each group is safe. We will clarify this in the revision.
>
> ### Q2. Notation in Eqs. (4)–(6)
> Using the original formulation:
> - $\alpha_t$: the arm pulled at time $t$.
> - $\alpha'$: any other arm in the group.
> - $\alpha^\dagger = \max_\{\alpha \in \mathcal{A}_ g\} : y \in C_\alpha$: the highest significance level (i.e., the most precise prediction set) that still covers the true label $y$.
>
>  We clarify: these equations update the estimates of other arms $\alpha'$ (not just the pulled arm $\alpha_t$) using the observed reward $r_t$ together with the structural properties of the problem. The updates are defined as:
>
> $$
> \hat{\mu}_g(\alpha') \leftarrow \frac{\hat{\mu}_g(\alpha') \cdot n_g^{\text{eff}}(\alpha') + w(\alpha_t, \alpha') \cdot r_t}{n_g^{\text{eff}}(\alpha') + w(\alpha_t, \alpha')}, \tag{4}
> $$
>
> $$
> \hat{\mu}_g(\alpha') \leftarrow \max\\{\hat{\mu}_g(\alpha'), \hat{\mu}_g(\alpha)\\} \quad \text{for } \alpha^\dagger \geq \alpha' > \alpha_t \tag{5}
> $$
>
> $$
> \hat{\mu}_g(\alpha') \leftarrow \min\\{\hat{\mu}_g(\alpha'), \hat{\mu}_g(\alpha)\\} \quad \text{for } 0 < \alpha' \leq \alpha_t \tag{6}
> $$
>
> Eq. (4) performs a soft update via Gaussian kernel $w(\alpha_t,\alpha')$, incorporating $r_t$ into every arm's estimate weighted by distance to $\alpha_t$, implementing continuity (Assumption 4.1).
>
> Eq. (5) enforces a lower bound for more precise arms ($\alpha^\dagger \geq \alpha' > \alpha_t$) that still cover the true label ($\alpha'\le\alpha^\dagger$): $\hat{\mu}_g(\alpha')\ge\hat{\mu}_g(\alpha_t)$, reflecting that a more precise set cannot lead to worse decision quality under coverage.
>
> Eq. (6) enforces an upper bound for less precise arms ($\alpha'\le\alpha_t$): $\hat{\mu}_g(\alpha')\le\hat{\mu}_g(\alpha_t)$, reflecting that a larger set cannot lead to better decision quality due to higher cognitive load.
>
> When coverage does not occur, the algorithm uses a separate rule (e.g. Algorithm 2, line 11) to set the reward to 0 for arms that cannot cover $y$. This combination allows a single pull to update all arms on one side of the pulled arm, dramatically reducing the dependence on the number of arms $K$. The reward depends on both context $x$ and arm $\alpha$; in our within‑group stochastic bandit, we maintain an estimate $\hat{\mu}_g(\alpha)$ that approximates the conditional expectation over the group’s context distribution. The Zooming Mechanism ensures groups are homogeneous enough for this estimate to be representative. We will clarify it more explicitly.
>
> ### Q3-5. Minor Corrections
>
> Thanks for your detailed suggestion!
>
> $\mu$ vs. $\mu^*$: We will unify the notation by using $\mu(x,\alpha)$ for the true expected reward and $\hat{\mu}_g(\alpha)$ for estimates.
>
> Algorithm 2 termination: We will add a check for $U-L=1$: directly compare the two arms and return the better one, then exit the loop.
>
> Pre‑trained learner $f$: The $f$ is introduced to indicate that our method works with any upstream model (e.g., a large language model) without requiring its internal details. Since it is not used later, we will remove it.
>
> Thanks for your thorough review. We believe our clarifications address your concerns.

---

> > ### Author Rebuttal · Reviewer_1Kqp · 2026-04-03
> >
> > I'd like to keep my current score

---

### Official Review · Reviewer_BQa9 · 2026-03-11

**Soundness:** 3
**Presentation:** 3
**Significance:** 2
**Originality:** 3
**Overall Recommendation:** 4
**Confidence:** 2

**Summary:**

This paper studies personalization of conformal-prediction-based decision support for heterogeneous human decision-makers, and proposes a method called Adaptively Grouped Contextual Bandits (AGCB). AGCB can adaptively partitions the context space while balances estimation variance and approximation bias and performs counterfactual arm updates leveraging a conditional monotonicity assumption together. Experiments show synthetic experiments where AGCB outperforms contextual and non-contextual baselines, especially when the number of arms is large.

**Compliance With Llm Reviewing Policy:**

Affirmed.

**Final Justification:**

My main concerns are addressed in the rebuttal.

**Key Questions For Authors:**

See weakness.

**Limitations:**

yes

**Strengths And Weaknesses:**

**Strengths**
1. Personalizing the informativeness of AI advice for heterogeneous human decision-makers in human–AI collaboration is important.
2. The problem formulation and theoretical guarantee looks good.
3. The split rule that explicitly balances estimation uncertainty and approximation bias is principled and connects cleanly with regret decomposition.

**Weakness**
1. The monotonicity assumption may be too strong in practice. The paper acknowledges the coverage-precision trade-off for unconditional reward but doesn't discuss scenarios where conditional monotonicity itself breaks down. Also, the conditional monotonicity assumption and its use in counterfactual updates appear internally inconsistent with the paper’s stated intuition that smaller sets reduce cognitive load. Please clarify it if my understandings are wrong.
2. Experiments are entirely synthetic that exactly satisfies all the paper's assumptions by construction, there's no real-world human study provided. The heterogeneity claim might be overstated. Does the method work when humans violate monotonicity (e.g., cognitive anchoring effects, preference reversals)?  Although the human studies is costly, simulated experiments on some larger dataset like imageNet16H might improve the illustration [1]. The authors also refer [1], but there's no comparison with that.

[1] Designing Decision Support Systems Using Counterfactual Prediction Sets

---

> ### Author Rebuttal · Authors · 2026-03-28
>
> Thanks for your valuable feedback.
> ### W1. Monotonicity Assumption
> 1.1 The assumption is empirically grounded
>
> Our monotonicity assumption is inspired by the counterfactual monotonicity introduced in [1]. While [1] focused on binary feedback, the core idea is identical: when the prediction set contains the true label, a more precise set leads to no worse decision quality. Their large‑scale human study (2,751 participants) provided strong empirical support that smaller prediction sets lead to higher success probability. Moreover, recent work by [2] confirms that more precise prediction sets improve human decision quality in LLM‑assisted tasks. Thus, far from being an overly strong assumption, our monotonicity condition is well‑grounded. In addition, we have conducted robustness experiments where we deliberately violate the monotonicity direction (detailed in Section 2.2), and AGCB remains the best‑performing algorithm.
>
> [1] Designing decision support systems using counterfactual prediction sets
>
> [2] Prune’n predict: Optimizing llm decision-making with conformal prediction
>
> 1.2 The algorithm is internally consistent
>
> The core of our approach combines two sources of structure:
> - Nested structure of conformal prediction: For $\alpha_1 > \alpha_2$, we have $C_{\alpha_1}(x) \subseteq C_{\alpha_2}(x)$.
> - Monotonicity (Assumption 4.2): When coverage occurs, a smaller set yields higher expected reward.
>
> When pulling the median arm $\alpha_t$ and observing coverage ($y \in C_{\alpha_t}(x)$) with reward $r_t$:
> - For all $\alpha' > \alpha_t$ (more precise sets), nestedness guarantees $y \in C_{\alpha'}(x)$, and monotonicity implies $\mu(\alpha',x) \ge \mu(\alpha_t,x)$. Thus Eq. (5) (max operation) updates $\hat{\mu}_g(\alpha')$, ensuring they are at least as good as $\hat{\mu}_g(\alpha_t)$.
> - For all $\alpha' < \alpha_t$ (larger sets), monotonicity gives $\mu(\alpha',x) \le \mu(\alpha_t,x)$, so Eq. (6) (min operation) updates $\hat{\mu}_g(\alpha')$, ensuring they are no better than $\hat{\mu}_g(\alpha_t)$.
>
> When coverage does not occur, the decision-maker trivially fails (reward 0) and the right side of the interval (smaller sets) is pruned. This logic directly translates the monotonicity intuition into a deterministic pruning mechanism—there is no inconsistency. The updates are conservative directional bounds, not arbitrary adjustments. More details are in our response to Reviewer 1Kqp, Q2.
>
> ### W2. Real‑World Human Studies and Robustness to Assumption Violations
> We fully agree that real‑world validation and robustness analysis are critical. Following your suggestions, we have conducted two new sets of experiments to address both points.
>
> 2.1 Real‑human data experiment
> We used the ImageNet16H‑PS dataset from [1] (194,407 predictions, 2,751 participants). Contexts are 21-dimensional vectors comprising image features (via pre-trained VGG-19), decision-maker and decision timing. Setting: $T=1000$, 30 runs. For continuous feedback, we added Gaussian noise to binary rewards.
>
> *Simple regret.* Our AGCB‑SH achieves the lowest regret.
> | Alg | Binary| Continuous|
> |-|-|-|
> | AGCB‑SH (ours) | 0.103 | 0.101 |
> | SH (vanilla) | 0.110 | 0.205 |
> | Contextual‑Gap | 0.110 | 0.109 |
> | BOED | 0.116 | 0.116 |
> | TTTS‑C | 0.135 | 0.155 |
> | LinUCB‑PE | 0.156 | 0.171 |
>
> *Cumulative regret.* The methods in [1] are UCB (with CF) and SE (with CF). We reproduce their finding that counterfactual updates reduce regret and UCB‑based methods perform best in binary feedback settings for cumulative regret. Our AGCB variants additionally leverage context and achieve even lower regret.
> | Alg | Binary| Continuous|
> |-|-|-|
> | AGCB‑UCB (ours) | 75.93 | 107.24 |
> | AGCB‑SE (ours) | 93.87 | 106.23 |
> | UCB (with CF) | 93.97 | 109.86 |
> | UCB (vanilla) | 150.23 | 179.55 |
> | SE (with CF) | 182.47 | 143.87 |
> | SE (vanilla) | 186.53 | 175.62 |
> | GP‑UCB | 94.50 | 120.52 |
> | LinUCB | 113.33 | 137.53 |
>
> 2.2 Robustness to monotonicity violations.
>
> We deliberately reversed the monotonicity direction with varying probabilities so that larger sets yield higher reward when coverage occurs. The results below (for simple regret) show that for $p \le 0.5$, AGCB‑SH remains the best‑performing algorithm. Compared to the original synthetic setting (binary: 0.298, continuous: 0.111), its regret increases to 0.352 and 0.131 at $p=0.5$, while other methods suffer even larger drops for the complex reward pattern.
> | Alg | Binary| Continuous|
> |-|-|-|
> | AGCB‑SH (ours) | 0.352 | 0.131 |
> | SH (vanilla) | 0.558 | 0.140 |
> | Contextual‑Gap | 0.523 | 0.174 |
> | BOED | 0.394 | 0.180 |
> | TTTS‑C | 0.386 | 0.204 |
> | LinUCB‑PE | 0.532 | 0.146 |
>
> Intuitively, robustness stems from weighted updates (Eq. 4) using a Gaussian kernel that decays with distance, making updates soft, and cross‑group information sharing, which dilutes the impact of any single erroneous update. Together, these ensure graceful degradation.
>
> Thanks again. We hope these address your concerns and welcome any further discussion.

---

> > ### Author Rebuttal · Reviewer_BQa9 · 2026-04-02
> >
> > Thanks for the rebuttals and my main concerns are addressed. I'm not an expert in this area, so I will raise my score if other reviewers are positive.

---

> > > ### Author Response · Authors · 2026-04-03
> > >
> > > Thank you for acknowledging that our responses have addressed your main concerns. We highly appreciate your understanding and would be grateful for your supportive evaluation if other reviewers share a positive view. Please feel free to let us know if you have any further questions.

---

### Official Review · Reviewer_1xK7 · 2026-03-12

**Soundness:** 3
**Presentation:** 2
**Significance:** 3
**Originality:** 3
**Overall Recommendation:** 4
**Confidence:** 4

**Summary:**

The paper proposes a framework that learns a series of coverage guarantees alpha in conformal prediction algorithms given the feature values. The algorithm proposed by the paper adaptively groups the feature instances to improve the data efficiency and uses binary search to find the optimal coverage guarantee given the monotonicity assumption. The theoretical results show that the proposed algorithm can achieve a regret that is logarithmic in the size of search grid of alpha and inversely polynomial in the size of training data. The experiment shows that the proposed algorithm achieves the lowest regret among several baselines.

**Compliance With Llm Reviewing Policy:**

Affirmed.

**Final Justification:**

The authors provide an additional experiment that solves my concern. I also notice that I have a misunderstanding regarding the setup. The revision mentioned by the authors address my other concerns.

**Key Questions For Authors:**

- See the questions mentioned in Weaknesses.
- What is the difference between simple regret and cumulative regret? It seems that they are both based on offline setting (assuming a distribution over X and Y). Is the cumulative just assuming that at each step the algorithm only has access to partial training data? But why the partition $\mathcal{G}_T$ is used as the final one?
- What is base learner $\mathcal{B}$?

**Limitations:**

Already discussed above

**Strengths And Weaknesses:**

Strengths:
- The paper grounds rigorously in theory.
- The structure of the paper is comprehensive, offering clear problem setup, theoretical results for bounds on regret, and experiment comparing against several baselines.

Weaknesses:
- The contribution of the paper seems to be overclaimed. In the introduction, the authors claim that the paper "providing a framework for learning personalized decision support.", but there is no result regarding different humans who have different decision functions. In the right-half column on line 143-144, the framework just unifies the human decision function into a reduced function $\mu$. The heterogeneity talked by the authors seems to just be the difference across different feature $x$.
- The experiment result is different from what claimed by the authors in the introduction. In the introduction, the authors claim that "empirical validation showing that our approach consistently improves human decision outcomes over strong and established baselines." However, the experiment does not involve any human decision-makers and also just based on a synthetic decision task, which simulates the prediction on the coverage guarantee $\alpha$.
- The setup of the paper is rarely seen. We usually want the coverage guarantee to be as high as possible but not to optimizing by varying that. What is the intuition behind that?
- There is no constrains on either the learn $f$ or the conformal prediction algorithm $C_{\alpha}(x)$ discussed by the paper. Would this make the problem trivial? For example, if $C_{\alpha}(x)$ always gives the full set $\mathcal{Y}$, changing $\alpha$ makes no difference to the regret.
- I find section 5 very hard to follow, largely because some notations lack definition or enough intuition.

Miscellaneous:
- $\Gamma_{share}$ on line 333 is undefined.
- Typos I found:
    - "andand" on line 229

---

> ### Author Rebuttal · Authors · 2026-03-28
>
> Thanks for your feedback. To clarify our core contribution: we study human-AI decision support where an AI provides a conformal prediction set (size controlled by $\alpha$) and the human makes the final decision. Traditional conformal prediction focuses on coverage, which alone is insufficient: too large a set imposes high cognitive load; too small a set risks missing the correct label. The optimal $\alpha$ balances coverage and precision, varying with task and human. Our AGCB framework learns this optimal $\alpha$ in a contextual bandit setting using adaptive grouping and counterfactual updates, without explicit modeling of the human decision function.
> ### W1. Personalization
> Our framework achieves personalization through the context vector $x$, which can encode user-specific features (e.g., user experience, demographics) alongside task features. By mapping these contexts to the optimal set size $\alpha$, the system inherently adapts to individual users. Crucially, we treat the human as a black box rather than explicitly modeling internal cognitive processes. Any unobserved human heterogeneity is simply absorbed as stochastic noise. This is a standard, principled online learning approach to optimize outcomes purely from behavioral feedback. Thus, our claim of "personalized decision support" is strictly justified.
> ### W2. Human Experiments
> We conducted additional experiments using real human data from ImageNet16H-PS [1] (194,407 predictions, 2,751 participants), constructing a realistic environment using its real empirical reward data. Contexts are 21-dimensional vectors comprising image features (via pre-trained neural network VGG-19), decision-maker and decision timing. Setting: $T=1000$, 30 runs. Our AGCB-SH achieves the lowest simple regret and cumulative regret, outperforming all baselines. More details are in our responses to Reviewer BQa9 W2.
>
> [1] Designing Decision Support Systems Using Counterfactual Prediction Sets.
> ### W3 & 4. Optimizing $\alpha$ and $f$ / $C_\alpha(x)$ Discussion
> We optimize $\alpha$ to balance coverage and precision for human decision quality, not to maximize coverage alone—a delicate trade‑off. Small $\alpha$ gives large sets, which overload the human with too many options, making effective decision difficult. Large $\alpha$ gives more precise sets that risk missing the true label. This balance is context‑dependent, making $\alpha$ optimization meaningful (More details in our response to Reviewer 1Kqp, Q2)
>
> The degenerate case $C_\alpha(x) \equiv \mathcal{Y}$ occurs only at $\alpha = 0$ in conformal predictors, such a large set does not reduce cognitive load. For $\alpha > 0$, the sets are nested ($\alpha_1 > \alpha_2 \Rightarrow C_{\alpha_1}(x) \subseteq C_{\alpha_2}(x)$). By varying $\alpha$, we adjust the set size to balance coverage and cognitive load, learning the $\alpha$ that best supports human decision‑making. Thus the problem is non‑trivial, and our counterfactual updates rely on this structure.
> ### W5. Clarity of Section 5
> Thanks for your correction! We will fix the typo by replacing $\Gamma_{\text{share}}$ with $\Gamma_{\min}$ and define it as the minimum information-sharing gain across groups in $\mathcal{G}_ T$, i.e., $\Gamma_{\min}=\min_ {g\in\mathcal{G}_ T}s_g \geq 1$, where $s_g=\sum_h w_{gh}$ measures similarity between groups. Intuitively, larger $\Gamma_{\min}$ means more knowledge exchange, improving sample efficiency and tightening regret bounds. With no sharing, $\Gamma_{\min}=1$ recovers the standard independent per-group learning in non-parametric contextual bandits.
> ### Q2. Simple vs Cumulative Regret
> Our setting is online. During the learning process at each round $t$, the algorithm observes a context $x_t$ and maintains a current partition $\mathcal{G}_t$ that evolves adaptively. Specifically, we determine the context's group $g_t \in \mathcal{G}_t$ and use its local statistics to choose $\alpha_t$ (Alg. 1, line 3), subsequently updating both $\mathcal{G}_t$ and the base learner. After $T$ rounds, the partition $\mathcal{G}_T$ is returned as the final output. It represents the ultimate learning outcome intended for future deployment.
>
> To evaluate our approach, we consider two metrics based on the application needs: cumulative regret measures the total loss during learning, crucial for systems that must perform well from the start, while simple regret measures the quality of the final deployed policy after $T$ rounds, relevant for pre-deployment calibration.
> ### Q3. Base Learner $\mathcal{B}$
> $\mathcal{B}$ is the local bandit algorithm run within each group. We instantiate it as Sequential Halving (for simple regret, Alg. 2) or Successive Elimination (for cumulative regret, Alg. 3), both enhanced with counterfactual updates. More generally, our framework is modular and can incorporate other bandit algorithms (e.g., UCB), as we demonstrate in our additional real‑human experiments.
>
> Thanks again. We hope our clarifications address your concerns.

---

> > ### Author Rebuttal · Reviewer_1xK7 · 2026-04-02
> >
> > Thanks for your rebuttal. I have following questions for the authors.
> >
> > W1. Thanks for clarification. I understand the authors' position better. However, I think the current writing of the paper is kind of misleading, which claims "Heterogeneous Human-AI Decision Making" but it's not. To avoid confusion, I would suggest to revise the paper to make that it clear that the focus is only about personal contextual information x but not the human heterogeneous.
> >
> > W2. Thanks for conducting the additional experiment and providing the results, but I couldn't understand it without enough details. Could the authors provide more? Such as how many participants are recruited, how they were recruited, what the decisions is, how the stimuli is generated, what is the preregistration (if any). Also could the authors report the confidence interval of the results?
> >
> > W3 & 4. I'm not sure about this. The choice of $\alpha$, prediction set algorithm $C(\cdot)$, and the predictor $f$ are independent components in this problem. I think there should be some assumptions on $C(\cdot)$ and $f$ that are not included in the paper. If $f$ is arbitrarily bad (say it's totally random), then no matter what $\alpha$ is, the set will always be very large. That's why I was confused when I saw the optimization just by varying alpha. Could the author explain more on how sensitive the proposed algorithm would be if predictor or prediction set algorithm are bad.
> >
> > Q2. My only question was about $\mathcal{G}_T$. If it's an online setting, shouldn't it be $\mathcal{G}_t$ at each step? Actually I just another question, for the simple regret, did the authors do the train/test split in the experiment?

---

> > > ### Author Response · Authors · 2026-04-03
> > >
> > > Thank you for your careful review.
> > > # W1
> > > We agree and will clarify in the revision: heterogeneity is captured via observed contextual features (including user attributes), following standard contextual bandits – different contexts lead to different optimal arms, no explicit human decision model needed.
> > > # W2
> > > Our additional experiment uses the public dataset ImageNet16H-PS from [1] and experimental setup:
> > > |Question|Answer|
> > > |-|-|
> > > |Participants Number|2751 human participants|
> > > |Recruitment method|Recruited via Prolific; compensated at £9/hour; signed informed consent|
> > > |Decision task|16-class image classification. AI provides a prediction set; participant chooses a label from the set|
> > > |Image source and processing|ImageNet16H dataset, 1200 images corrupted with phase noise ($\omega = 110$). Human-alone accuracy ≈ 0.760|
> > > |Prediction set construction|Conformal prediction with varying $\alpha$ yields 715 questionnaires covering all image–set pairs|
> > >
> > > IRB and preregistration: The original experiment [1] received IRB approval from the University of Saarland but did not mention preregistration. As a secondary analysis, we will clarify this in revision.
> > >
> > > [1] Designing decision support systems using counterfactual prediction sets
> > >
> > > The confidence intervals of the experimental results are:
> > >
> > > ### Cumulative Regret
> > > |Alg|Binary(95%CI)|Continuous(95%CI)|
> > > |-|-|-|
> > > |AGCB‑UCB(ours)|75.93[75.74,76.12]|107.24[105.50,108.98]|
> > > |AGCB‑SE(ours)|93.87[93.79,93.95]|106.23[104.99,107.47]|
> > > |UCB(withCF)|93.97[93.93,94.01]|109.86[109.20,110.52]|
> > > |UCB(vanilla)|150.23[146.32,154.14]|179.55[176.04,183.06]|
> > > |SE(withCF)|182.47[178.39,186.55]|143.87[139.27,148.47]|
> > > |SE(vanilla)|186.53[183.36,189.70]|175.62[172.56,178.68]|
> > > |GP‑UCB|94.50[94.24,94.76]|120.52[119.92,121.12]|
> > > |LinUCB|113.33[107.39,119.27]|137.53[132.56,142.50]|
> > >
> > > ### Simple Regret
> > > |Alg|Binary(95%CI)|Continuous(95%CI)|
> > > |-|-|-|
> > > |AGCB‑SH(ours)|0.103[0.099,0.107]|0.101[0.097,0.105]|
> > > |SH(vanilla)|0.110[0.108,0.112]|0.205[0.192,0.218]|
> > > |Contextual‑Gap|0.110[0.108,0.112]|0.109[0.106,0.112]|
> > > |BOED|0.116[0.110,0.122]|0.116[0.107,0.125]|
> > > |TTTS‑C|0.135[0.124,0.146]|0.155[0.145,0.165]|
> > > |LinUCB‑PE|0.156[0.150,0.162]|0.171[0.164,0.178]|
> > >
> > > Our AGCB methods are best across nearly all settings; the marginal CI overlap in binary feedback cumulative regret also reproduces [1], UCB(withCF), as a strong baseline.
> > > # W3 & 4
> > > Conformal prediction does not assume anything about the upstream model $f$. As stated in [2], "the sets are valid in a distribution-free sense: they possess explicit, non-asymptotic guarantees even without distributional assumptions or model assumptions." and "One can use conformal prediction with any pre-trained model, such as a neural network" Similarly, [3] notes in the abstract that conformal prediction solves the problem of forming prediction sets "without any assumptions on the form of the data generating distribution."
> > >
> > > [2] A Gentle Introduction to Conformal Prediction and Distribution-Free Uncertainty Quantification
> > >
> > > [3] Theoretical Foundations of Conformal Prediction
> > >
> > > A simple example makes this concrete. Take the absolute residual score $S(x,y)=|y-\hat{f}(x)|$. For any upstream model $f$, let $Q_{1-\alpha}$ denote the empirical $(1-\alpha)$-quantile of the scores computed on a calibration set. As $\alpha$ increases, $Q_{1-\alpha}$ decreases (or remains the same), so the prediction set $\{y: S(x,y) \le Q_{1-\alpha}\} = \{y: |y-\hat{f}(x)| \le Q_{1-\alpha}\}$ shrinks monotonically. This holds even if $f$ is completely random. Hence $\alpha$ always controls the set size, and optimising $\alpha$ is always meaningful.
> > >
> > > # Q2
> > > **1. $\mathcal{G}_t$ vs $\mathcal{G}_T$**
> > > The algorithm uses the current partition $\mathcal{G}_t$ during online learning (Alg 1, line 3). $\mathcal{G}_T$ denotes the final partition output after $T$ rounds, intended for deployment (Alg 1, line 8); during training we only use $\mathcal{G}_t$.
> > >
> > > **2. Train/test split for simple regret**
> > > We strictly follow a train/test split.
> > >
> > > Training phase: learn the final grouping rule $\mathcal{G}_T$ (mapping context to group) and per-group optimal $\alpha$, forming a mapping from context to recommended $\alpha$.
> > >
> > > Test phase: evaluate on a fresh set of contexts. For each test context, we compute its true optimal $\alpha$ (by exhaustive search over the finite set $\mathcal{A}$) and the utility of the algorithm's recommended $\alpha$. Simple regret is the average difference between the optimal utility and the algorithm's utility across test samples, matching the standard definition.
> > >
> > > Finally, we highlight the contribution and positioning of this paper. We investigate workflow optimization for the emerging field of AI–human collaborative decision-making, with the goal of optimizing AI provision via context-driven adaptive experimentation. To the best of our knowledge, this work is the first of its type reported in the existing literature. We anticipate that it will inspire and advance continued research in this exciting direction.

---

### Decision · Program_Chairs · 2026-04-30

**Decision:**

Accept (regular)

**Comment:**

This paper studies the novel problem of human-AI collaboration, where the AI system provides the human decision-maker with a decision set built by conformal prediction whose size is controlled by a learned significance parameter. This helps humans in decision-making by narrowing the set of alternatives to consider. The idea is interesting and timely, and has been formalized and analyzed in an elegant theoretical framework, which is likely to benefit future developments in human-AI joint decision-making systems.

One major criticism was that the experiments were focused on synthetic decision tasks. In the rebuttal, the authors expanded their experiments using real-world human data and showed that their method outperformed the baselines. Another one was questioning the monotonicity assumption. The authors referenced evidence from real-world studies and also argued that their algorithm is consistent under this assumption.

Overall, this paper is likely to have a moderate-to-high impact within the research community. The clarifications and new experiments that were made during the rebuttal phase were critical in reaching consensus among the reviewers and the final decision. Therefore, these changes should be carefully integrated into the final version of the paper.